# Does Flatness imply Generalization for Logistic Loss in Univariate Two-Layer ReLU Network?

## Abstract

We consider the problem of generalization of arbitrarily overparameterized two-layer ReLU Neural Networks with univariate input. Recent work showed that under square loss, flat solutions (motivated by flat / stable minima and Edge of Stability phenomenon) provably cannot overfit, but it remains unclear whether the same phenomenon holds for logistic loss. This is a puzzling open problem because existing work on logistic loss shows that gradient descent with increasing step size converges to interpolating solutions (at infinity, for the margin-separable cases). In this paper, we prove that the *flatness implied generalization* is more delicate under logistic loss. On the positive side, we show that flat solutions enjoy near-optimal generalization bounds within a region between the left-most and right-most *uncertain* sets determined by each candidate solution. On the negative side, we show that there exist arbitrarily flat yet overfitting solutions at infinity that are (falsely) certain everywhere, thus certifying that flatness alone is insufficient for generalization in general. We demonstrate the effects predicted by our theory in a well-controlled simulation study.

## 1 Introduction

In modern deep learning, the optimization landscape is highly nonconvex and often filled with numerous local minima and saddle points. Therefore, it is natural and important to consider the role of the optimizer in guiding the model toward particular types of solutions among many possible low-loss regions (Zhang et al., 2021). One important phenomenon in this context is the implicit bias of minima stability, which refers to the tendency of optimization algorithms—particularly gradient-based methods—to prefer solutions that are more stable under perturbations (Wu et al., 2018). These stable solutions, often characterized by flatter curvature in the loss landscape, are less sensitive to small changes in input or model parameters. This implicit preference arises not from the explicit objective function, but from the dynamics of the training process itself, and has important implications for generalization performance even in the absence of global optimality (Qiao et al., 2024).

The implicit bias of minima stability has been extensively studied in the context of squared loss, particularly through analysis conducted in function space (Mulayoff et al., 2021; Nacson et al., 2022). These studies have shown that gradient-based optimization algorithms tend to converge to predictors that exhibit greater stability—often corresponding to smoother or simpler functions—even in the absence of explicit regularization (Qiao et al., 2024). This perspective has provided valuable insight into how model generalization arises from the training dynamics themselves. However, many practical classification problems are trained using logistic or cross-entropy loss rather than squared loss. Extending the analysis of implicit bias (in function space) to logistic loss is therefore both natural and necessary, as it enables a deeper understanding of how stability and generalization emerge in classification settings, where the geometry of the loss landscape and the behavior of the optimization algorithm can differ significantly from regression (Chizat & Bach, 2020).

**Our contributions.** In this paper, we study the minima stability and generalization theory for the solutions gradient descent with constant learning rate can stably converge to. More specifically,

1. We show that for 1D nonparametric logistic regression with noisy labels, the solutions that GD can stably converge to must be regular functions with small (weighted) first-order total variation, while the constraint is significant (*i.e.* the weight function is large) only between the left-most and

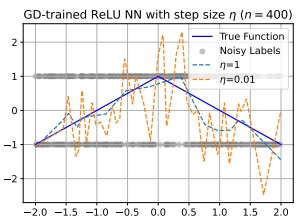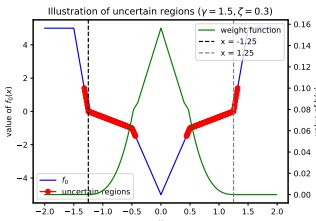

Figure 1: The **left panel** summarizes our findings about flatness, generalization and interpolation in logistic regression. The **middle panel** compares the learned function by GD with large and small learning rates, the stable solution of large learning rate is simpler and smoother. The **right panel** provides an illustration for the "uncertain region" (red part) of the function and the weight function (the $h$ function in equation 5 and equation 7) supported in the interior of uncertain regions. Briefly speaking, a larger weight function poses stronger smoothness guarantee in the corresponding region. Here we plot the asymptotic weight function in Theorem 3.5 with $\gamma = 1.5, \zeta = 0.3$ and $\mathcal{P}_x = \text{Unif}([-2, 2])$. The weight function in Theorem 3.1 can be derived identically by replacing $f_0$ with $f_\theta$.

right-most "uncertain sets" where the prediction of the output function is not certain (Theorem 3.1). Please refer to Figure 1 for an illustration of the relevant concepts. Unfortunately, we construct an example (Theorem 3.2) showing that interpolating solutions (at infinity) can be arbitrarily flat, thus flatness alone does not imply generalization. However, with a mild assumption (Assumption 3.4) that the optimization effectively minimizes the excess risk, we further show that such stable solutions must be smooth in the (predefined) convex hull of "uncertain regions" of the ground-truth function (Theorem 3.5), which excludes those pathological interpolating solutions.

2. We establish concrete generalization bounds for such solutions (stable local minima that GD converges to). We first prove a generalization gap bound that depends on the scale of the learned coefficients (Theorem 3.3), showing that sparse functions provably do not overfit. Moreover, under a mild additional assumption on gradient descent finding "optimized" solutions, we refine the analysis over the interior of the convex hull of the "uncertain regions" and achieve near optimal rates of excess risk for estimating first-order bounded variation functions (Theorem 3.7).

3. We perform comprehensive numerical experiments to support our theoretical predictions, verify key technical assumptions, and visualize both the structure of ReLU neural networks and the basis functions discovered by gradient descent under varying step sizes. These findings offer fresh insights into how gradient descent training effectively captures representations and naturally promotes implicit sparsity under logistic regression.

**Technical Novelty.** We build on top of Qiao et al. (2024) to analyze the generalization ability of trained NNs under logistic regression. However, different from the squared loss where all interpolating solutions are sharp, the relationship between flatness and smoothness is more delicate for logistic loss. To handle the increasing flatness when the prediction approaches infinity, we separate the data points with "certain" and "uncertain" predictions. Moreover, we manage to derive smoothness guarantees based on the part with "uncertain" predictions, where the guarantee becomes weaker as the extent of uncertainty decreases. Meanwhile, based on a diminishing excess risk assumption, we take a novel approach to approximate the "uncertain set" of the learned function by the uncertain regions of the ground-truth, thus achieving a prediction-independent weighted TV1 bound from flatness, which excludes those arbitrarily flat but overfitting solutions. As a result, together with a careful analysis on the metric entropy of bounded variation class, this allows us to amplify the diminishing excess risk to a nearly optimal rate, when restricted to the interior of the convex hull of uncertain regions.

## 1.1 DISCLAIMERS AND LIMITATIONS

It is worth noting that our results hold for the learned functions satisfying several conditions (*e.g.* flatness, optimized), while we do not have guarantees whether GD could actually converge to such solutions. We acknowledge that initialization and dynamics of GD are important in training, while we focus on the global properties of the solutions instead. The (somewhat loose) connection to GD is established via local linear stability theory (see Appendix B.1 for completeness) and via empirical

observations of "Edge of Stability" phenomenon (which appears for logistic loss). It is important to note that our results on generalization do not depend on the optimization being successful, *i.e.*, our generalization gap bound works for every candidate solution with low-loss-curvature. Our findings lie in an intermediate regime between classical learning theory which disregards optimization consideration and modern theoretical frameworks centered on optimization dynamics.

Admittedly, we focus on the univariate case and logistic loss, which may be perceived as restrictive. We remark that the problem is sufficiently interesting and challenging even for the univariate case. The challenges and our technical innovations needed for handling logistic loss are complementary to understanding the multivariate inputs. The restriction to logistic loss (instead of more general family of losses) is a deliberate choice too. Logistic loss is by far the most popular in practice, and it is well understood in various settings, which implies more concrete connections between our results and other theoretical work. Meanwhile, extending our results to more general loss functions is possible and promising (*e.g.* in Appendix F we extend the generalization bound to Lipschitz losses).

## 1.2 RELATED WORK

**Implicit bias of gradient descent.** The implicit bias of gradient descent training of overparameterized NN is well-studied for both squared loss (Arora et al., 2019; Mei et al., 2019; Jin & Montúfar, 2023) and logistic loss. Under logistic regression with (linearly) separable data, linear predictors are shown to converge to the direction of the max-margin solution for small enough learning rates (Soudry et al., 2018). The result is extended to the maximal linearly separable subset of nonseparable data (Ji & Telgarsky, 2019), wide two-layer neural networks (Chizat & Bach, 2020), more general loss functions (Schliserman & Koren, 2022) and multi-class classification (Ravi et al., 2024). However, none of the results implies generalization bounds when the labels are noisy, which is our focus.

**Implicit bias of minima stability.** The most relevant works are the series of papers studying the implicit bias of minima stability (Ma & Ying, 2021; Mulayoff et al., 2021; Nacson et al., 2022; Wu & Su, 2023; Qiao et al., 2024). Among the results above, Mulayoff et al. (2021) interprets the minima stability from the function space, relying on the minima interpolating the data. Later, Qiao et al. (2024) considers noisy data and derives generalization bounds without the assumption of interpolation. However, all these works focus on the squared loss, which is inherently different from the widely applied logistic loss. We consider the implicit bias of minima stability from the view of function space under logistic regression, which raises substantial technical challenges.

**Edge of Stability.** Our problem setup is motivated by the empirical observations of the "edge of stability" phenomenon (Cohen et al., 2020), where large learning rate training of NN finds solutions with Hessian's largest eigenvalue oscillating around $2/\eta$. A growing body of research attempts to understand such behavior of GD training (Kong & Tao, 2020; Arora et al., 2022; Ahn et al., 2022; Wang et al., 2022; Damian et al., 2022; Zhu et al., 2022; Lyu et al., 2022; Ahn et al., 2023; Kreisler et al., 2023; Even et al., 2023; Lu et al., 2023). Our work is complementary in that we provide generalization bounds to the final solution GD stabilizes on no matter how GD gets there.

**Logistic regression with large step size.** Previous works (Wu et al., 2023; 2024; Cai et al., 2024; Zhang et al., 2025) show that GD with large learning rate could accelerate the convergence of logistic regression for linearly separable data. Our results are different in two aspects. First, we consider labels generated randomly, which can be arbitrarily nonseparable. In addition, we do not study optimization dynamics, but on the generalization ability of the final learned function.

**Flat minima and generalization.** Early empirical evidence (Hochreiter & Schmidhuber, 1997; Keskar et al., 2017) supports the hypothesis that "flat solutions generalize better". More recently Wu & Su (2023); Qiao et al. (2024); Ding et al. (2024) established theoretical generalization bounds for flat solutions in various settings. Among them, Ding et al. (2024) focuses on matrix sensing models and neural networks with quadratic activations. Both Wu & Su (2023); Ding et al. (2024) require interpolation. Moreover, they all use square loss, while we consider logistic loss, which reveals surprising new insight on this hypothesis.

## 2 PROBLEM SETUP

In this section, we introduce our problem setup. Throughout the paper, $[n] = \{1, 2, \cdots, n\}$. We use standard notation $O(\cdot), \Omega(\cdot), \widetilde{O}(\cdot)$ to absorb constants or logarithmic terms, respectively.

**Two-layer ReLU neural network.** The function class we consider is two-layer (*i.e.* one-hidden-layer) univariate ReLU networks, defined as

$$\mathcal{F} = \left\{ f : \mathbb{R} \to \mathbb{R} \;\middle|\; f(x) = \sum_{i=1}^{k} w_i^{(2)} \phi\left(w_i^{(1)} x + b_i^{(1)}\right) + b^{(2)} \right\}, \tag{1}$$

where the network consists of $k$ hidden neurons and $\phi(\cdot)$ denotes the ReLU activation function.

**Binary classification and (Nonlinear) logistic regression.** The training dataset is denoted by $\mathcal{D} = \{(x_i, y_i) \in \mathbb{R} \times \{-1, 1\}, i \in [n]\}$. The univariate feature $x_i$ is assumed to be supported by $[-x_{\max}, x_{\max}]$ for some constant $x_{\max} > 0$, while the label $y_i \in \{-1, 1\}$. We consider the logistic regression problem with loss $\ell(f, (x, y)) = \log(1 + e^{-yf(x)})$. Then the training loss is $\mathcal{L}(f) = \frac{1}{n}\sum_{i=1}^{n} \log\left(1 + e^{-y_i f(x_i)}\right)$. We denote the parameters of $f$ by $\theta := [w_{1:k}^{(1)}, b_{1:k}^{(1)}, w_{1:k}^{(2)}, b^{(2)}] \in \mathbb{R}^{3k+1}$. In addition, we let $\ell_i(\theta) := \ell(f_\theta, (x_i, y_i))$ and $\mathcal{L}(\theta) := \frac{1}{n}\sum_{i \in [n]} \ell_i(\theta)$ as a short hand.

**Gradient descent.** Let $\theta_0$ be the initial parameter, we optimize the logistic loss above using Gradient descent (GD). At the $t$-th step, we update $\theta$ as $\theta_t = \theta_{t-1} - \eta \nabla \mathcal{L}(\theta_{t-1})$, where $\eta > 0$ is the learning rate (a.k.a. step size). We leave the calculation of the gradient for logistic loss and two-layer ReLU networks to Appendix C.

**Minima stability.** The stability of gradient descent (GD) is often described by the local curvature of the training loss function, *i.e.*, $\lambda_{\max}(\nabla^2 \mathcal{L}(\theta))$ at $\theta$. Classical optimization theory teaches us to use step size $\eta < 2/\sup_\theta \lambda_{\max}(\nabla^2 \mathcal{L}(\theta))$ such that gradient descent steps do not diverge. However, for overparameterized neural networks, there is no global upper bound of the smoothness. Instead, it has been observed that gradient descent with any fixed $\eta > 0$ only discovers solutions such that $\lambda_{\max}(\nabla^2 \mathcal{L}(\theta)) \approx 2/\eta$. This phenomenon is known as the edge of stability (EoS) regime (Cohen et al., 2020; Damian et al., 2022). Moreover, the linear stability theory (Wu et al., 2018; Mulayoff et al., 2021) also suggests that gradient descent can only stabilize around solutions $\theta$ where the Hessian's largest eigenvalue is smaller than $2/\eta$ in the local neighborhood of $\theta$.

This motivated Qiao et al. (2024) to define and analyze the following set of functions.

$$\mathcal{F}(\eta, \mathcal{D}) := \left\{ f_\theta \;\middle|\; \lambda_{\max}(\nabla^2 \mathcal{L}(\theta)) \le \frac{2}{\eta} \right\}. \tag{2}$$

Unless otherwise specified, a *"stable solution"* or a *"flat solution"* are used interchangeably in the rest of the paper to refer to an element of $\mathcal{F}(\eta, \mathcal{D})$.

**Data generation.** We assume that the features $\{x_i\}_{i=1}^{n}$ are i.i.d. samples from some distribution $\mathcal{P}_x$ supported on $[-x_{\max}, x_{\max}]$. Meanwhile, we assume that there exists a ground-truth function $f_0 : [-x_{\max}, x_{\max}] \to \mathbb{R}$ such that conditional on the feature $x_i$, the label $y_i$ is sampled independently from the following distribution:

$$y = \begin{cases} 1 & \text{with probability } p = \sigma(f_0(x)) \\ -1 & \text{with probability } 1 - p \end{cases} \tag{3}$$

where $\sigma(t) = \frac{1}{1+e^{-t}}$ is the sigmoid function. Therefore, the populational (testing) loss is defined as $\bar{\mathcal{L}}(f) := \mathbb{E}_{x \sim \mathcal{D}} \mathbb{E}_{y \sim \mathcal{B}(x)} \log\left(1 + e^{-yf(x)}\right)$, where $x \sim \mathcal{D}$ is short for uniformly sampling from the dataset (features) and $\mathcal{B}(x)$ is short for the distribution equation 3 throughout the paper. According to direct calculation (details in Lemma B.3), the optimal prediction function is $f^\star = \arg\min_f \bar{\mathcal{L}}(f) = f_0$. Our goal is to find a function $f_\theta$ using the dataset $\mathcal{D}$ to minimize the Excess Risk:

$$\text{ExcessRisk}(f) = \bar{\mathcal{L}}(f) - \bar{\mathcal{L}}(f_0). \tag{4}$$

We consider the nonparametric logistic regression task where we do not require $f_0$ to be represented by a limited number of parameters. Instead, we impose certain regularity conditions. Specifically, our focus is on estimating target functions within the class of first-order bounded variation:

$$f_0 \in \text{BV}^{(1)}(B, C_n) := \left\{ f : [-x_{\max}, x_{\max}] \to \mathbb{R} \;\middle|\; \max_x |f(x)| \le B, \int_{-x_{\max}}^{x_{\max}} |f''(x)| dx \le C_n \right\}.$$

Here, $f''$ is the second-order weak derivative of $f$, and we introduce the shorthand $\text{TV}^{(1)}(f) := \int_{-x_{\max}}^{x_{\max}} |f''(x)| dx$, which we refer to as the $\text{TV}^{(1)}$ norm of $f$ throughout the paper. Some discussions of the historical significance and challenges in estimating BV functions can be found in Section 1.2 of Hu et al. (2022). The complexity of such function class is further explored in Appendix B.3.

## 3 MAIN RESULTS

In this section, we state the main results for stable solutions of GD (functions in $\mathcal{F}(\eta, \mathcal{D})$). Section 3.1 connects the flatness of the function to the smoothness and characterizes the implicit bias of stable solutions in the function space. Section 3.2 provides an example of an arbitrarily flat yet overfitting solution, thus showing that flatness alone could not ensure generalization. Section 3.3 considers the generalization ability through bounding the generalization gap by the coefficient scale. Based on the result, Section 3.4 refines the bound in Section 3.1 and derives a total variation bound independent of the output function. Finally, Section 3.5 derives a concrete Excess Risk bound by leveraging the implicit bias shown in Section 3.4.

### 3.1 IMPLICIT BIAS OF MINIMA STABILITY IN THE FUNCTION SPACE

Theorem 3.1 provides a weighted TV(1) upper bound for the learned stable solution $f = f_\theta$.

**Theorem 3.1.** *For a threshold $\gamma > 0$, define the "uncertain" data points with respect to $f_\theta$ as $\mathcal{A}_\gamma = \left\{ x_i \in \mathcal{D} \;\middle|\; |f_\theta(x_i)| \leq \gamma \right\}$ and let $n_\gamma = |\mathcal{A}_\gamma|$. Define the function $\Gamma(\gamma) = \frac{e^{-\gamma}}{(1+e^{-\gamma})^2} = \Omega(e^{-\gamma})$. Then for any function $f = f_\theta$ such that the training loss $\mathcal{L}$ is twice differentiable at $\theta$ and any $\gamma > 0$[1],*

$$1 + 2 \int_{-x_{\max}}^{x_{\max}} |f''(x)| \, h_\gamma(x) dx \leq \frac{n}{n_\gamma \Gamma(\gamma)} \left( \lambda_{\max}(\nabla^2 \mathcal{L}(\theta)) + 2 x_{\max} \mathcal{L}(f) \right), \tag{5}$$

*where $h_\gamma(x) = \min \left\{ h_\gamma^+(x), h_\gamma^-(x) \right\}$ and*

$$h_\gamma^+(x) = \mathbb{P}^2 \left( X > x \mid X \in \mathcal{A}_\gamma \right) \cdot \sqrt{1 + \left( \mathbb{E} \left[ X \mid X \in \mathcal{A}_\gamma, \, X > x \right] \right)^2} \cdot \mathbb{E} \left[ X - x \;\middle|\; X \in \mathcal{A}_\gamma, \, X > x \right],$$

$$h_\gamma^-(x) = \mathbb{P}^2 \left( X < x \mid X \in \mathcal{A}_\gamma \right) \cdot \sqrt{1 + \left( \mathbb{E} \left[ X \mid X \in \mathcal{A}_\gamma, \, X < x \right] \right)^2} \cdot \mathbb{E} \left[ x - X \;\middle|\; X \in \mathcal{A}_\gamma, \, X < x \right],$$

*where for both functions, $X$ is a random sample from the dataset under the uniform distribution. Moreover, if $f = f_\theta$ is a solution in $\mathcal{F}(\eta, \mathcal{D})$, we can replace $\lambda_{\max}(\nabla^2 \mathcal{L}(\theta))$ in equation 5 by $\frac{2}{\eta}$.*

The proof of Theorem 3.1 is deferred to Appendix D. The theorem connects the flatness (measured by the maximum eigenvalue of the Hessian matrix) in the parameter space to the smoothness of the output function (measured by a weighted TV(1) norm) in the function space. More specifically, the theorem claims that between the left-most and right-most "uncertain sets" where the learned function has uncertain predictions, the learned function must be regular in the sense of first-order total variation. In the following, we first discuss the trade-off between the parameters.

As the threshold $\gamma$ increases, the "uncertain set " $\mathcal{A}_\gamma$ will also become larger. As a result, the term $\frac{n}{n_\gamma}$ on the R.H.S will become smaller, while the interior of $\mathcal{A}_\gamma$ where the TV constraint is strong will also be larger. Both effects tend to make the result more significant. However, the term $\frac{1}{\Gamma(\gamma)} \approx e^\gamma$ on the R.H.S will grow exponentially, and thus balance out the aforementioned effects. Therefore, there is a trade-off in the choice of $\gamma$. Meanwhile, since we do not have any guarantee on the scale of $f_\theta$, the portion or even the existence of the uncertain set $\mathcal{A}_\gamma$ for a fair $\gamma$ is not ensured. Therefore, the question whether flatness could imply generalization still remains.

### 3.2 FLATNESS ALONE DOES NOT ENSURE GENERALIZATION

Unfortunately, we provide a negative answer to the question whether flatness could imply generalization under logistic regression. We construct the following general example and show that for any choice of labels, we can construct a solution that is arbitrarily flat and overconfident at infinity.

**Example setup.** Let $x_{\max} = 1$, then the interval becomes $[-1, 1]$. The design $\{x_i\}_{i=1}^n$ is chosen to be $n$ equally spaced points in $[-1, 1]$. For any label set $\{y_i\}_{i=1}^n \in \{-1, 1\}^n$, let $\gamma_{\max} > 0$ be some

---

[1]W.l.o.g. we assume that $x_{\max} \geq 1$. If this does not hold, we can directly replace $x_{\max}$ in the bounds by 1.

arbitrarily large constant, below we consider the two-layer ReLU NN $f$ where $f(x_i)y_i = \gamma_{\max}$ and $f$ is (almost) linear between any two neighboring design points. We will show that there exists some function $f = f_\theta$ satisfying the conditions above while $\lambda_{\max}(\nabla^2_\theta \mathcal{L}(\theta))$ is small. By saying "almost linear", we mean that we need to perturb the knots from the design points by a small $\epsilon$ to satisfy the twice differentiable assumption.

**Theorem 3.2.** *For the example above, there exists a choice of $\theta$ such that $f_\theta(x_i) = y_i\gamma_{\max}$ for all $i \in [n]$, $\mathcal{L}(\theta)$ is twice differentiable w.r.t. $\theta$ and $\lambda_{\max}\left(\nabla^2_\theta \mathcal{L}(\theta)\right) \leq O\left((n^2\gamma_{\max} + 1)e^{-\gamma_{\max}}\right)$.*

The proof of Theorem 3.2 is deferred to Appendix E. Since $\lim_{\gamma \to \infty} \gamma e^{-\gamma} = 0$, Theorem 3.2 shows that as the learned function tends to interpolate (*i.e.* the training loss gets close to 0), the landscape of the parameter is actually becoming flat. In other words, the solution that predicts $\infty$ for $y = 1$ and $-\infty$ for $y = -1$ is an arbitrarily flat global minimum with 0-training loss, while these solutions are heavily overfitting if the labels are noisy. This is inherently different from nonparametric regression with squared loss (Qiao et al., 2024), where all interpolating solutions must be sharp. However, although our Theorem 3.1 could not exclude these overfitting solutions, in our experiments GD with a constant step size does not actually converge to such interpolating solutions. To close the gap, we will show in the following sections that under some mild assumptions, flatness could imply generalization within certain regions of the data support.

### 3.3 BOUNDING THE GENERALIZATION GAP BY WEIGHT DECAY

Recall that the populational loss is defined as $\bar{\mathcal{L}}(f) = \mathbb{E}_{x\sim\mathcal{D}}\mathbb{E}_{y\sim\mathcal{B}(x)}\log\left(1 + e^{-yf(x)}\right)$. Theorem 3.3 states that the generalization gap $\left|\mathcal{L}(f) - \bar{\mathcal{L}}(f)\right|$ can be bounded by the $\ell_2$ norm of parameters.

**Theorem 3.3.** *For any constant $B > 0$, with probability $1 - \delta$, for any two-layer ReLU NN $f = f_\theta$ supported on $[-x_{\max}, x_{\max}]$ such that $\|f\|_\infty \leq B$, it holds that*

$$\left|\mathcal{L}(f) - \bar{\mathcal{L}}(f)\right| \leq O\left(\left[\frac{\log(1 + e^B)^4 \cdot \max\{\|\theta\|_2^2, 1\} \cdot x_{\max}\log\left(\frac{2\max\{\|\theta\|_2^2, 1\}}{\delta}\right)^2}{n^2}\right]^{\frac{1}{5}}\right), \quad (6)$$

*where the randomness is over the data generation process.*

Theorem 3.3, whose proof is deferred to Appendix F, claims that if the scale of $\theta$ is small, then the generalization gap of $f_\theta$ will also be small. We first point out that the usage of $\|\theta\|_2$ is mainly for the ease of presentation, and the $\|\theta\|_2^2$ in equation 6 can be replaced by the scale of coefficients only $\sum_{i:b_i^{(1)}/w_i^{(1)}\in[-x_{\max},x_{\max}]}\left|w_i^{(1)}w_i^{(2)}\right|$, which upper bounds the TV$^{(1)}$ norm of $f_\theta$ over $[-x_{\max}, x_{\max}]$. The other parts of $\|\theta\|_2$ (*i.e.* the biases) only contribute to a logarithmic term. The rigorous analysis is deferred to Appendix F.3. Therefore, under various cases (*e.g.* training with weight decay (Zhang & Wang, 2022) or large learning rate) where the $\|\theta\|_2$ or $\sum_{i:b_i^{(1)}/w_i^{(1)}\in[-x_{\max},x_{\max}]}\left|w_i^{(1)}w_i^{(2)}\right|$ grows at a rate of order $o(n)$, Theorem 3.3 implies a diminishing generalization gap bound as $n \to \infty$.

Meanwhile, under the case where the $\|\theta\|_2$ or $\sum_{i:b_i^{(1)}/w_i^{(1)}\in[-x_{\max},x_{\max}]}\left|w_i^{(1)}w_i^{(2)}\right|$ grows at a rate of order $o(n)$, if the optimization is effective such that the training loss $\mathcal{L}(f_\theta)$ is smaller than $\mathcal{L}(f_0)$ (this is supported by our experiments), we can further derive a diminishing excess risk bound for $f_\theta$ of the same order as equation 6. The assumptions and excess risk bounds are formalized in Appendix F.2. Finally, Theorem 3.3 focuses on the setting with logistic loss and fixed design (features). The analysis naturally extends to more general loss function that is Lipschitz continuous w.r.t. $f$ (Lemma F.5) or the statistical learning setting where the data $(x_i, y_i)$ are i.i.d. samples from some joint distribution $\mathcal{P}$ with populational risk defined as $\mathcal{R}(f) := \mathbb{E}_{(x,y)\sim\mathcal{P}}\ell(f, (x, y))$ (Corollary F.7).

### 3.4 TOTAL VARIATION BOUND INDEPENDENT OF THE OUTPUT FUNCTION

In this part, we will derive a (weighted) TV(1) bound independent of the output function, based on the assumption that the excess risk is diminishing as the number of data points increases.

**Assumption 3.4.** We assume that the learned function $f = f_\theta$ satisfies $\bar{\mathcal{L}}(f) - \bar{\mathcal{L}}(f_0) \leq \epsilon(n)$, *i.e.*

$$\mathbb{E}_{x\sim\mathcal{D}}\mathbb{E}_{y\sim\mathcal{B}(x)}\log\left(1 + e^{-yf(x)}\right) \leq \mathbb{E}_{x\sim\mathcal{D}}\mathbb{E}_{y\sim\mathcal{B}(x)}\log\left(1 + e^{-yf_0(x)}\right) + \epsilon(n),$$

for some function $\epsilon(n)$ such that $\lim_{n\to\infty}\epsilon(n) = 0$.

The Assumption 3.4 holds by choosing $\epsilon(n)$ to be any diminishing excess risk bound. In addition, $\epsilon(n)$ can be the bound in Appendix F.2 as a special case. Now we are ready to derive a TV(1) bound based on $\epsilon(n)$ and showcase the asymptotic performance of $f_\theta$ as $n \to \infty$.

**Theorem 3.5.** *For a threshold $\gamma > 0$ and probability $p > 0$, define the uncertain set (w.r.t. $f_0$) as $\bar{\mathcal{A}}_\gamma := \left\{ x \in [-x_{\max}, x_{\max}] \mid |f_0(x)| \leq \gamma \right\}$ and the truncated uncertain set as $\bar{\mathcal{A}}_\gamma^p := \left\{ x \in \bar{\mathcal{A}}_\gamma \mid \min \left( \mathbb{P}_{X \sim \mathcal{D}}(X \in \bar{\mathcal{A}}_\gamma,\ X > x),\ \mathbb{P}_{X \sim \mathcal{D}}(X \in \bar{\mathcal{A}}_\gamma,\ X < x) \right) \geq p \right\}$. Define $\Gamma(\gamma) = \frac{e^{-\gamma}}{(1+e^{-\gamma})^2} = \Omega(e^{-\gamma})$. Then for any function $f = f_\theta$ such that the training loss $\mathcal{L}$ is twice differentiable at $\theta$ and any $\gamma > \zeta > 0$, if Assumption 3.4 holds with $\epsilon(n)$, for some function $p(n) = O\left(\epsilon(n) \cdot e^{\gamma+\zeta} \cdot \zeta^{-2}\right)$,*

$$\Gamma(\gamma) \cdot \left(\mathbb{P}_{X \sim \mathcal{D}}\left(X \in \bar{\mathcal{A}}_{\gamma-\zeta}\right) - p(n)\right) \cdot \left(1 + 2 \int_{-x_{\max}}^{x_{\max}} |f''(x)| \bar{h}_{\gamma,\zeta}(x,n) dx\right) \leq \lambda_{\max}(\nabla^2 \mathcal{L}(\theta)) + 2x_{\max} \mathcal{L}(f), \tag{7}$$

*where $\bar{h}_{\gamma,\zeta}(x,n) = \min\left\{\bar{h}_{\gamma,\zeta}^+(x,n), \bar{h}_{\gamma,\zeta}^-(x,n)\right\}$ and*

$$\bar{h}_{\gamma,\zeta}^+(x,n) = \left(\frac{\mathbb{P}_{X \sim \mathcal{D}}(X > x,\ X \in \bar{\mathcal{A}}_{\gamma-\zeta}) - p(n)}{\mathbb{P}_{X \sim \mathcal{D}}(X \in \bar{\mathcal{A}}_{\gamma+\zeta}) + p(n)}\right)^2 \cdot \frac{\mathbb{E}_{X \sim \mathcal{D}}\left[(X - x)\mathbb{1}\left(X > x,\ X \in \bar{\mathcal{A}}_{\gamma-\zeta}^{p(n)}\right)\right]}{\mathbb{P}_{X \sim \mathcal{D}}\left[X > x,\ X \in \bar{\mathcal{A}}_{\gamma+\zeta}\right] + p(n)},$$

$$\bar{h}_{\gamma,\zeta}^-(x,n) = \left(\frac{\mathbb{P}_{X \sim \mathcal{D}}(X < x,\ X \in \bar{\mathcal{A}}_{\gamma-\zeta}) - p(n)}{\mathbb{P}_{X \sim \mathcal{D}}(X \in \bar{\mathcal{A}}_{\gamma+\zeta}) + p(n)}\right)^2 \cdot \frac{\mathbb{E}_{X \sim \mathcal{D}}\left[(x - X)\mathbb{1}\left(X < x,\ X \in \bar{\mathcal{A}}_{\gamma-\zeta}^{p(n)}\right)\right]}{\mathbb{P}_{X \sim \mathcal{D}}\left[X < x,\ X \in \bar{\mathcal{A}}_{\gamma+\zeta}\right] + p(n)}.$$

*Furthermore, since the features $\{x_i\}_{i=1}^n$ are i.i.d. samples from $\mathcal{P}_x$, as $n \to \infty$, the asymptotic total variation guarantee can be derived by plugging in $p(n) = 0$ and replacing $X \sim \mathcal{D}$ with $X \sim \mathcal{P}_x$ in equation 7 and $\bar{h}_{\gamma,\zeta}(x,n)$. Moreover, if $f = f_\theta$ is a stable solution in $\mathcal{F}(\eta, \mathcal{D})$, we can replace $\lambda_{\max}(\nabla^2 \mathcal{L}(\theta))$ in equation 7 and the corresponding asymptotic guarantee by $\frac{2}{\eta}$.*

The complete version and proof of Theorem 3.5 is deferred to Appendix G. The first part of Theorem 3.5 is a relaxation of equation 5 in Theorem 3.1. More specifically, under the assumption (Assumption 3.4) that the excess risk of $f_\theta$ is bounded, we can bound the gap of $f_\theta$ and $f_0$ (Lemma G.2). As a result, we can approximate the uncertain set $\mathcal{A}$ in Theorem 3.1 by $\bar{\mathcal{A}}$ or $\bar{\mathcal{A}}^p$, with a correction term $p(n)$ that is proportional to $\epsilon(n)$. Although the functions satisfying equation 7 is actually a superset of equation 5, note that for a fixed $\gamma > 0$, $\bar{\mathcal{A}}_\gamma$ only depends on $f_0$, while $\bar{\mathcal{A}}_\gamma^p$ only depends on $f_0$ and the feature set $\{x_i\}_{i=1}^n$. This is in striking contrast to Theorem 3.1, where $n_\gamma$ and $h_\gamma(x)$ all depend on the output function $f_\theta$, which can be arbitrary functions. Meanwhile, as $n$ converges to infinity, $p(n) = O(\epsilon(n))$ will converge to 0, while the empirical distribution $x \sim \mathcal{D}$ will converge to $x \sim \mathcal{P}_x$. Therefore the asymptotic smoothness guarantee only depends on $f_0$ and the feature distribution $\mathcal{P}_x$. The asymptotic smoothness guarantee can be found in Theorem G.5, while we leave the discussions about trade-offs between parameters to Appendix G.1.

**Remark 3.6.** Different from the nonparametric regression setting (Qiao et al., 2024) where the smoothness constraint holds over the whole interior of the data support, in logistic regression the smoothness guarantee is restricted to the interior of the convex hull of "uncertain regions". For the boundary part of the interval with small randomness (*i.e.* large $|f_0(x)|$ value), the smoothness constraint is weak since a large $\gamma$ is required to incorporate such region in $\bar{\mathcal{A}}_\gamma$. Such result poses an interesting separation of logistic regression with (nearly) separable data and non-separable data. Previously, Wu et al. (2024) shows that GD with large step size could help with the convergence of logistic regression with (linearly) separable data. As a complement, our results show that large step size also helps with the generalization ability of the trained NN in the non-separable regime.

### 3.5 Minima Stability of GD Leads to Near-optimal Excess Risk

With the weighted TV(1) bound in Theorem 3.5, we are ready to refine the excess risk bound in Section 3.3 over the region where $\widetilde{h}_{\gamma,\zeta}(x,n) := \Gamma(\gamma)\left(\mathbb{P}_{X \sim \mathcal{D}}\left(X \in \bar{\mathcal{A}}_{\gamma-\zeta}\right) - p(n)\right)\bar{h}_{\gamma,\zeta}(x,n)$ in Theorem 3.5 is lower bounded. Recall that the features $\{x_i\}_{i=1}^n$ are i.i.d. samples from some distribution $\mathcal{P}_x$ supported on $[-x_{\max}, x_{\max}]$.

**Theorem 3.7.** *For any interval $\mathcal{I} \subset [-x_{\max}, x_{\max}]$ such that there exists $\gamma > \zeta > 0$ satisfying that with probability at least $1 - \frac{\delta}{2}$, $\widetilde{h}_{\gamma,\zeta}(x, n) \geq c$, $\forall x \in \mathcal{I}$ for some constant $c > 0$, for any stable solution $f = f_\theta$ in $\mathcal{F}(\eta, \mathcal{D})$ such that the training loss $\mathcal{L}$ is twice differentiable at $\theta$ and $\|f\|_\infty \leq B$, if $f$ is "optimized" over $\mathcal{I}$, i.e., $\sum_{x_i \in \mathcal{I}} \ell(f, (x_i, y_i)) \leq \sum_{x_i \in \mathcal{I}} \ell(f_0, (x_i, y_i))$ and $f_0 \in \mathrm{BV}^{(1)}\left(B, \frac{1}{c}\left(\frac{1}{\eta} + x_{\max}(1+B)\right)\right)$, then with probability $1 - \delta$,*

$$\mathrm{ExcessRisk}_{\mathcal{I}}(f) = \bar{\mathcal{L}}_{\mathcal{I}}(f) - \bar{\mathcal{L}}_{\mathcal{I}}(f_0) \leq O\left(\left[\frac{(B+1)^4\left(\frac{x_{\max}}{\eta} + x_{\max}^2(1+B)\right)\log(1/\delta)^2}{n_{\mathcal{I}}^2}\right]^{\frac{1}{5}}\right),$$

(8)

*where $\bar{\mathcal{L}}_{\mathcal{I}}(f) = \frac{1}{n_{\mathcal{I}}}\sum_{x_i \in \mathcal{I}} \mathbb{E}_{y_i' \sim \mathcal{B}(x_i)}\ell(f, (x_i, y_i'))$, $n_{\mathcal{I}}$ is the number of features $x_i$ in $\mathcal{I}$ and the $O(\cdot)$ also absorbs the constant $\frac{1}{c}$ and $\frac{1}{|\mathcal{I}|}$.*

The proof of Theorem 3.7 is deferred to Appendix H. Theorem 3.7 focuses on an interval $\mathcal{I}$ where $\widetilde{h}$ can be lower bounded. In this way, we ignore the extreme data points and derive an Excess Risk upper bound (restricted to $\mathcal{I}$) of order $\widetilde{O}(n_{\mathcal{I}}^{-2/5})$, which matches the minimax optimal rate for estimating $\mathrm{BV}^{(1)}$ functions (Zhang et al., 2024). In addition, Theorem 3.7 does not have explicit dependence on the width of NN $k$, and therefore holds for arbitrary k, even if the neural network is heavily over-parameterized ($k \gg n$). For the choice of $\mathcal{I}$, similar to the analysis in Qiao et al. (2024, App. G), $\mathcal{I}$ can be chosen to incorporate most of the data points in the convex hull of "uncertain regions".

Meanwhile, note that minima stability (Theorem 3.1 & Theorem 3.5) poses stronger smoothness constraint in the middle of the interval, while the constraint becomes weaker towards the boundary. Therefore, compared to Theorem 3.3, Theorem 3.7 provides a refined analysis focusing on the interior of the convex hull of "uncertain regions" instead of the whole interval. The superiority of Theorem 3.7 over Theorem 3.3 will be significant when the learned function $f$ has huge volatility at the boundary while being smoother in the middle, where the $\mathrm{TV}^{(1)}$ norm can be much smaller when restricted to the interval $\mathcal{I}$.

Lastly, the Excess Risk bound depends on the learning rate $\eta$ as $\eta^{-1/5}$. This arises because a larger learning rate leads to a smaller bound on $\mathrm{TV}^{(1)}$ of the learned function $f$. As a result, the hypothesis space excludes many non-smooth functions, which tightens the Excess Risk bound. However, a larger learning rate is not always beneficial. If $\eta$ is too large, GD may diverge. Even in cases where convergence is maintained, the resulting class of stable solutions may be insufficiently expressive to approximate the ground truth function $f_0$, particularly when $\int_{-x_{\max}}^{x_{\max}} |f_0''(x)|dx \gg \frac{1}{c\eta} + \frac{x_{\max}(1+B)}{c}$, thereby invalidating the assumption of an "optimized" solution. In our experiments, we numerically verify this assumption across a wide range of $\eta$ values, and show that by tuning $\eta$, we effectively adapt to the unknown value of $\mathrm{TV}^{(1)}(f_0)$.

## 4 EXPERIMENTS

### 4.1 EXPERIMENTAL SETUP

In this section, we empirically justify our results by training a (mildly overparameterized) two-layer ReLU neural network to minimize logistic loss using gradient descent with varying step sizes. The input dataset comprises of $n$ equally spaced fixed design points $\{x_i\}_{i=1}^n$ located in $[-2, 2]$, where the number of data $n$ is chosen to be 80, 160, 400 in different trials. The noisy labels $y_i$ follow the distribution equation 3 with the ground-truth function $f_0(x) = (x+1)\mathbf{1}(x \leq 0) + (-x+1)\mathbf{1}(x > 0)$. The two-layer ReLU NN is parameterized by $\theta$ as in Section 2 with the number of neurons $k = 400$. The network uses standard parameterization (scale factor of 1) and the parameters are initialized randomly (see Figure 14 for the initial basis functions).

### 4.2 EXPERIMENTAL RESULTS

Figure 2 illustrates the relationship between the learned ReLU NN that GD-training stabilizes on and the choice of step size. The main take-aways are (a) GD with large learning rate stably converges to flatter minima which represent more regular functions (in $\mathrm{TV}^{(1)}$); (b) Training with smaller learning

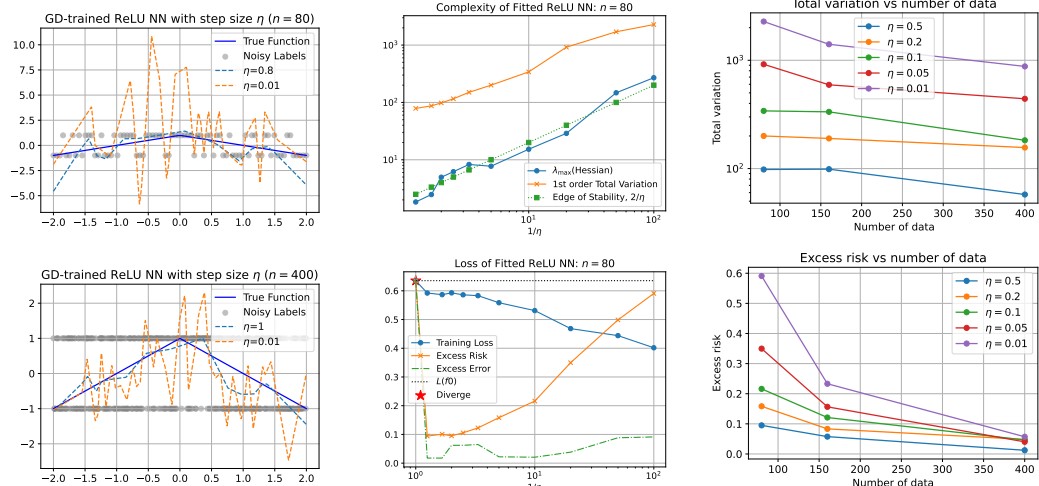

Figure 2: Highlight of our empirical results. The **left panel** illustrates the learned functions for GD with large (0.8 or 1) and small (0.01) learning rates using different numbers of samples. The **middle panel** plots the impact of varying learning rate on the complexity and performance of the learned function. The **right panel** showcases the following relationships: (1) $TV^{(1)}$ norm vs number of data and (2) excess risk vs number of data under several fixed choices of learning rate.

rate will lead to smaller training loss, while the learned function tends to overfit and showcase the standard U-shape for excess risk and excess error[2]; (c) When we fix the learning rate $\eta$ and increase the number of data $n$, overfitting is relieved and the excess risk converges to 0 quickly, thus justifying Assumption 3.4; (d) Meanwhile, the $TV^{(1)}$ norm of the learned function does not increase with $n$, which implies the optimality of the refined version of Theorem 3.3 (in the order of $n$).

We leave more experimental details to Appendix A. The learned functions and learning curves with more choices of $\eta$ and $n$ are shown in Appendix A.1, which further supports the claims in Theorem 3.1 and 3.5. Meanwhile, Appendix A.2 provides more illustrations of the relationship between complexity & performance & sparsity of the learned function and the learning rate, where the results are generally consistent with the theorems. Moreover, we highlight that all results satisfy the "optimized" assumption in Theorem 3.7. Lastly, Appendix A.3 visualizes the learned basis functions under different learning rates. The learned representations are very different from initialization, thus our experiments are clearly describing phenomena not covered by the "kernel" regime.

## 5 CONCLUSION

In this paper, we consider the well-motivated generalization ability under logistic loss from a lens of minima stability. The key take-aways are: (1) Flatness alone could not ensure generalization due to the existence of arbitrarily flat yet interpolating solutions; (2) Flatness + low-confidence predictions together could render generalization, while the guarantee is tighter within the region with less confidence (more uncertainty); (3) GD with large learning rate could converge to a sharpness of $2/\eta$ by either making the function smoother or more confident, which could depend on the initialization (and other factors). Therefore, an interesting direction is to study the relationship between smoothness and confidence by incorporating training dynamics, which we leave to future work.

---

[2]Excess error for $f$ is $\widetilde{\mathcal{L}}(f) - \widetilde{\mathcal{L}}(f_0)$, where $\widetilde{\mathcal{L}}(f) := \mathbb{E}_{x \sim \mathcal{D}}[\mathbb{1}(f(x) \geq 0)(1 - \sigma(f_0(x))) + \mathbb{1}(f(x) < 0)\sigma(f_0(x))]$, which measures the expected 0-1 loss if we use the sign of $f$ as the prediction.

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

# A FULL EXPERIMENTAL RESULTS

## A.1 STABLE MINIMA GD CONVERGES TO AND LEARNING CURVES

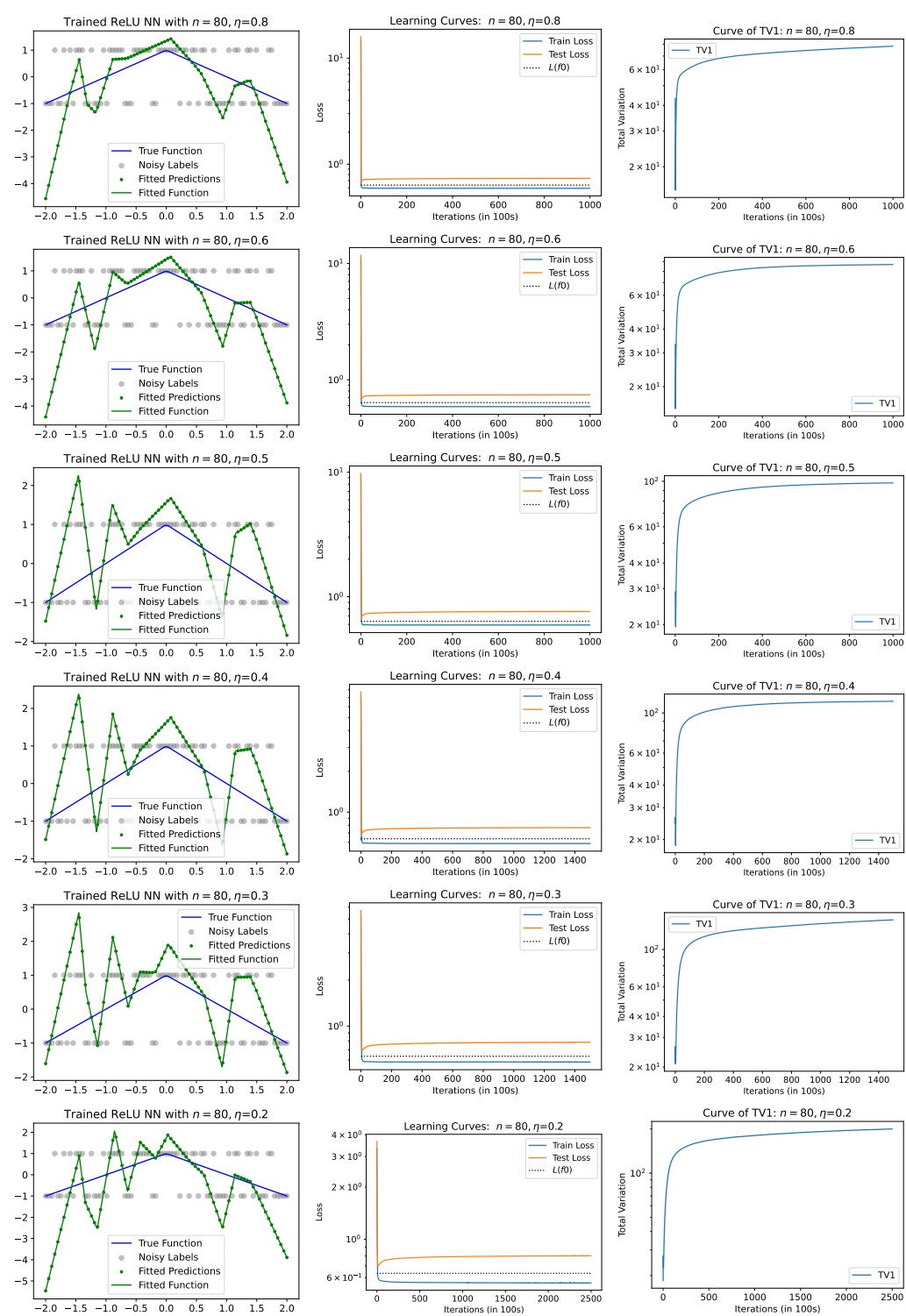

Figure 3: Illustration of the solutions gradient descent with learning rate $\eta$ converges to ($n = 80$: Part I). As $\eta$ decreases, the fitted function goes from simple to complex. Any line below the $\mathcal{L}(f_0)$ line satisfies the "optimized" assumption from Theorem 3.7 and Lemma F.10. Test loss denotes $\mathcal{L}(f)$.

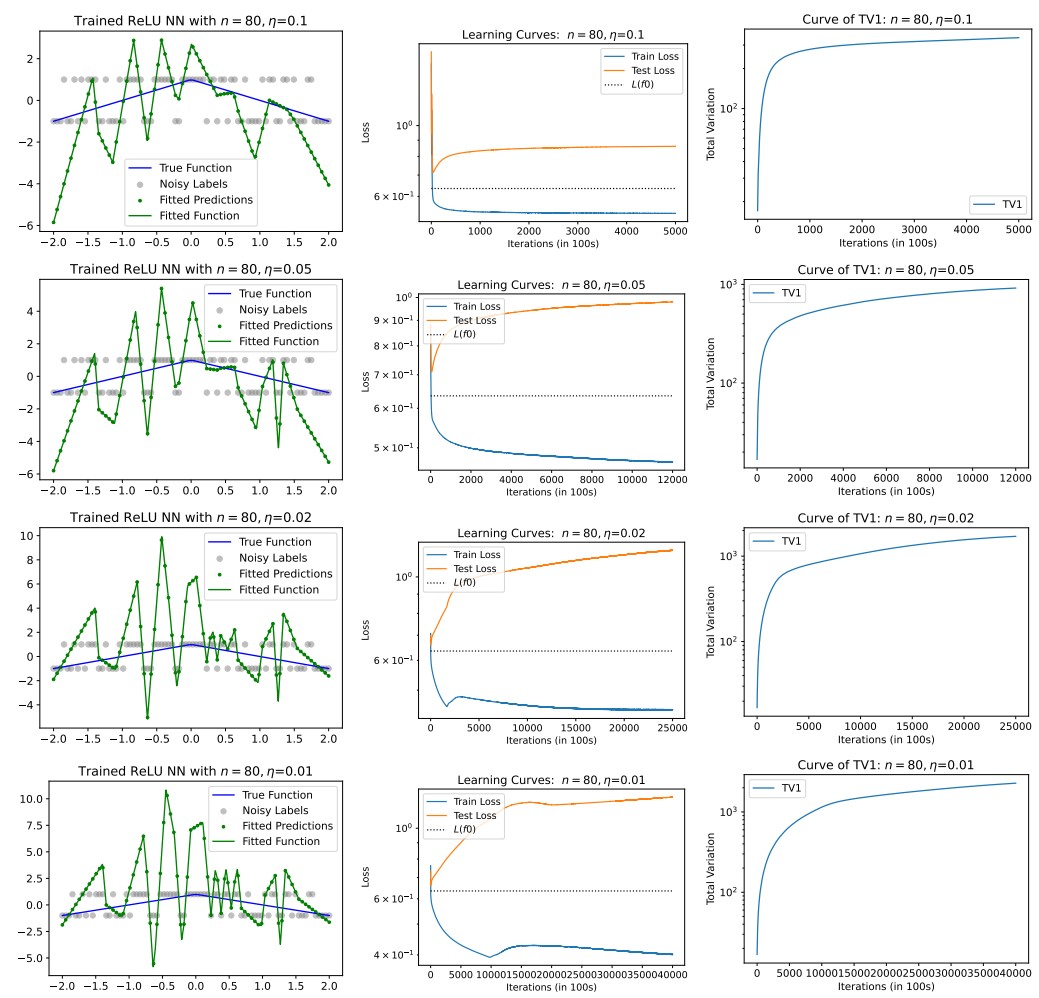

Figure 4: Illustrations ($n = 80$: Part II). As $\eta$ decreases further, the fitted function starts to overfit.

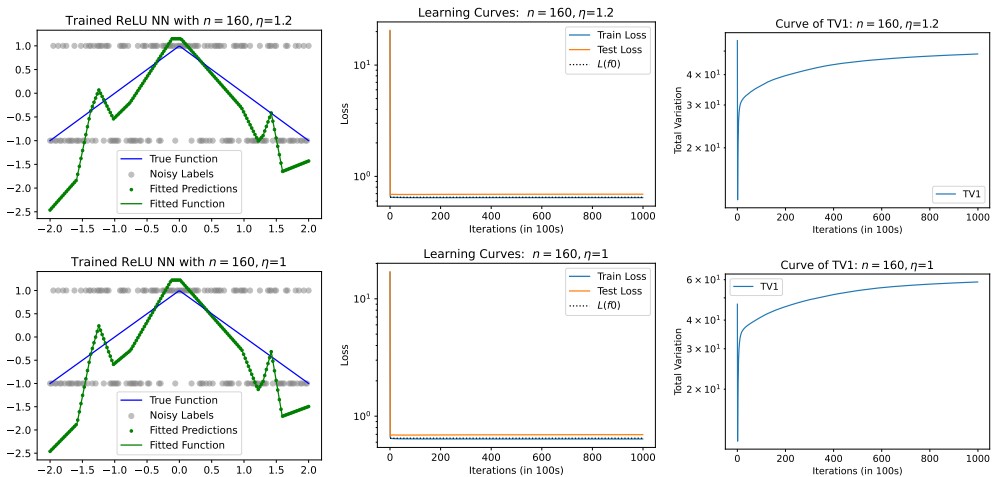

Figure 5: Illustration of the solutions GD with learning rate $\eta$ converges to ($n = 160$: Part I).

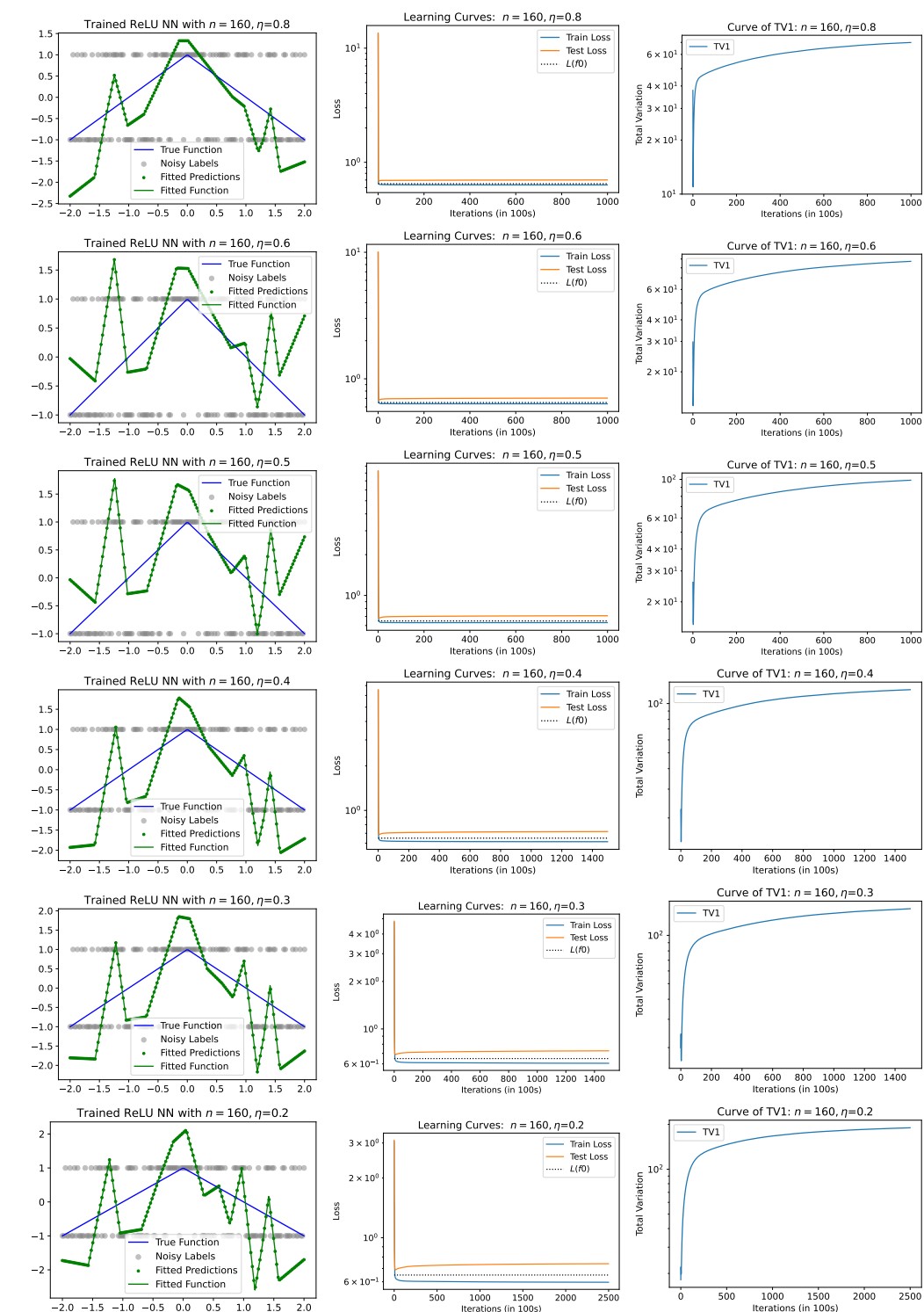

Figure 6: Illustration of the solutions gradient descent with learning rate $\eta$ converges to ($n = 160$: Part II). As $\eta$ decreases, the fitted function goes from simple to complex. Compared to the case where $n = 80$ in Figure 3, for the same $\eta$, the learned function approximates the ground-truth better.

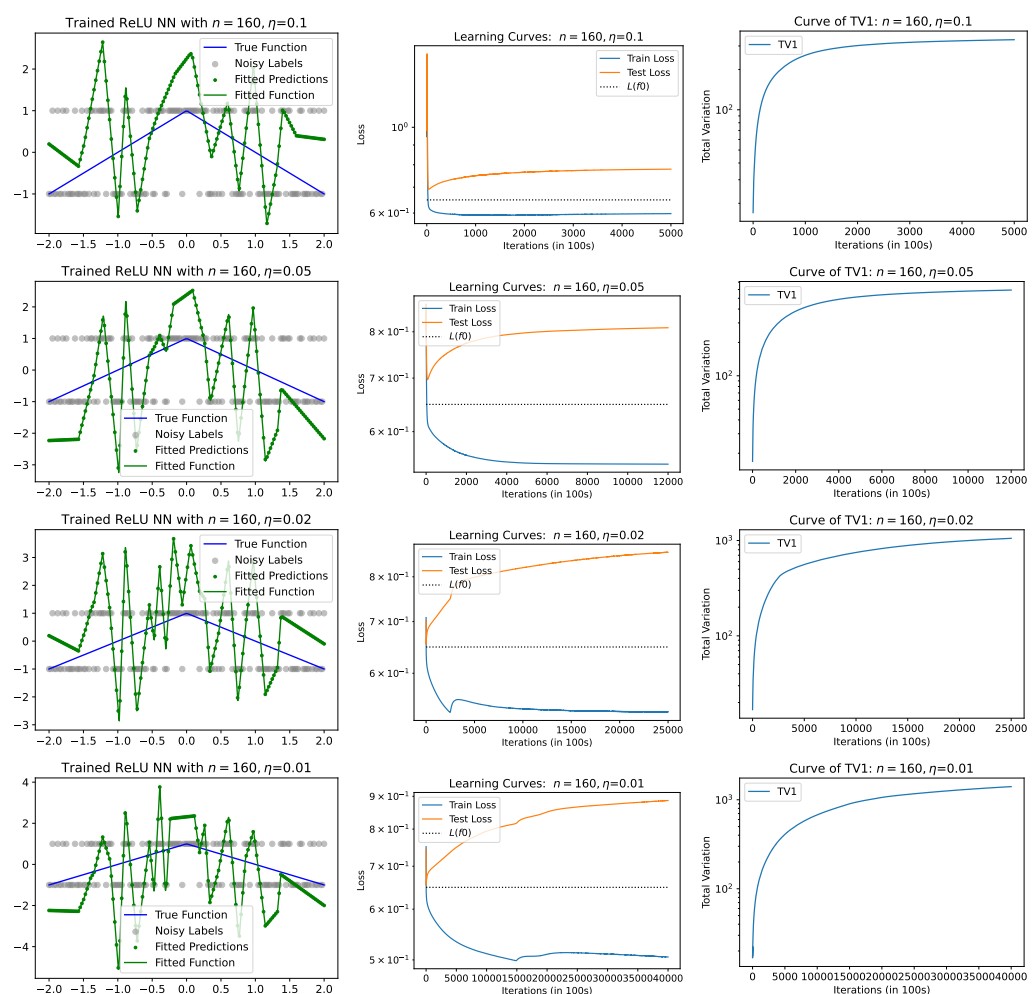

Figure 7: Illustration of the solutions ($n = 160$: Part III). As $\eta$ decreases further, the fitted function starts to overfit, while the overfitting is not as catastrophic as the case with fewer samples (Figure 4).

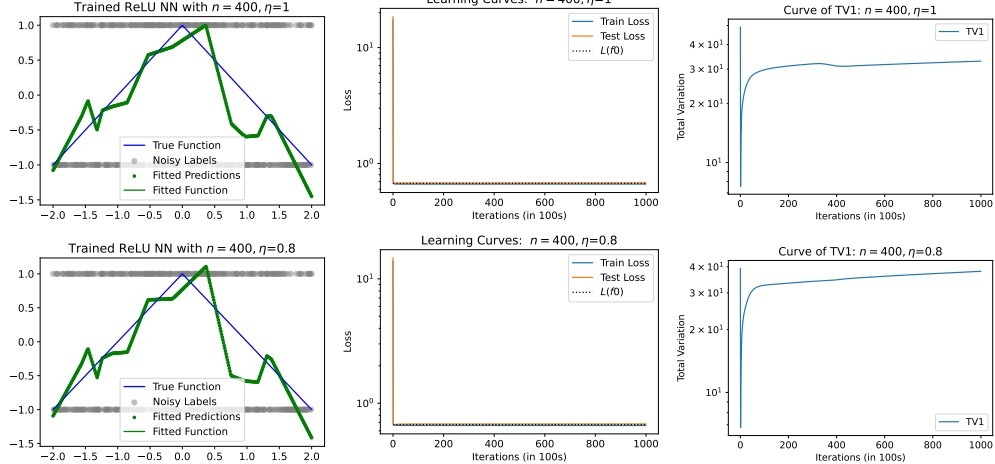

Figure 8: Illustrations ($n = 400$: Part I). For large learning rate, the learned function is close to $f_0$.

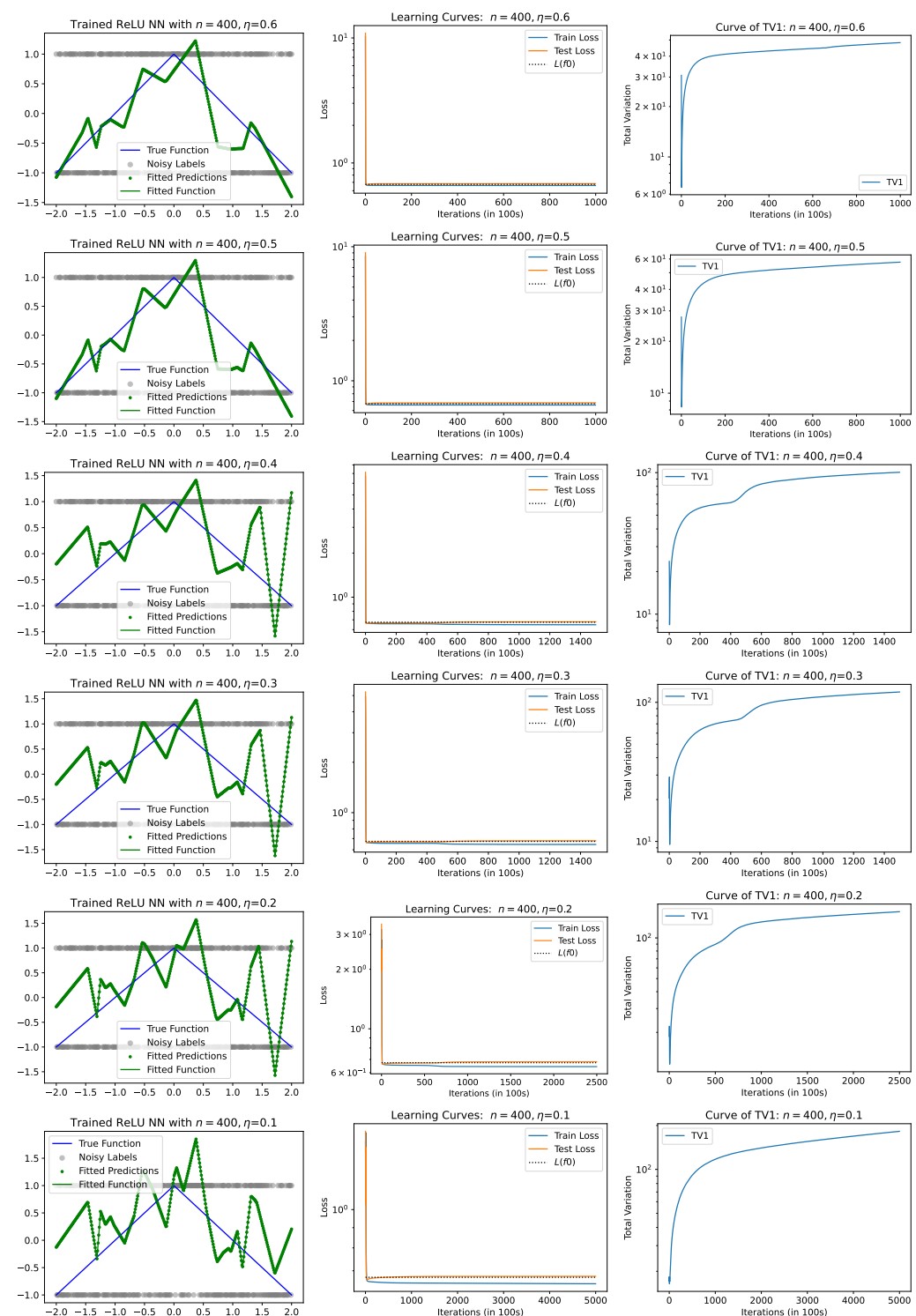

Figure 9: Illustration of the solutions gradient descent with learning rate $\eta$ converges to ($n = 400$: Part II). As $\eta$ decreases, the fitted function goes from simple to complex. Compared to the cases where $n = 80$ in Figure 3 or $n = 160$ in Figure 6, for the same $\eta$, the learned function approximates the ground-truth much better.

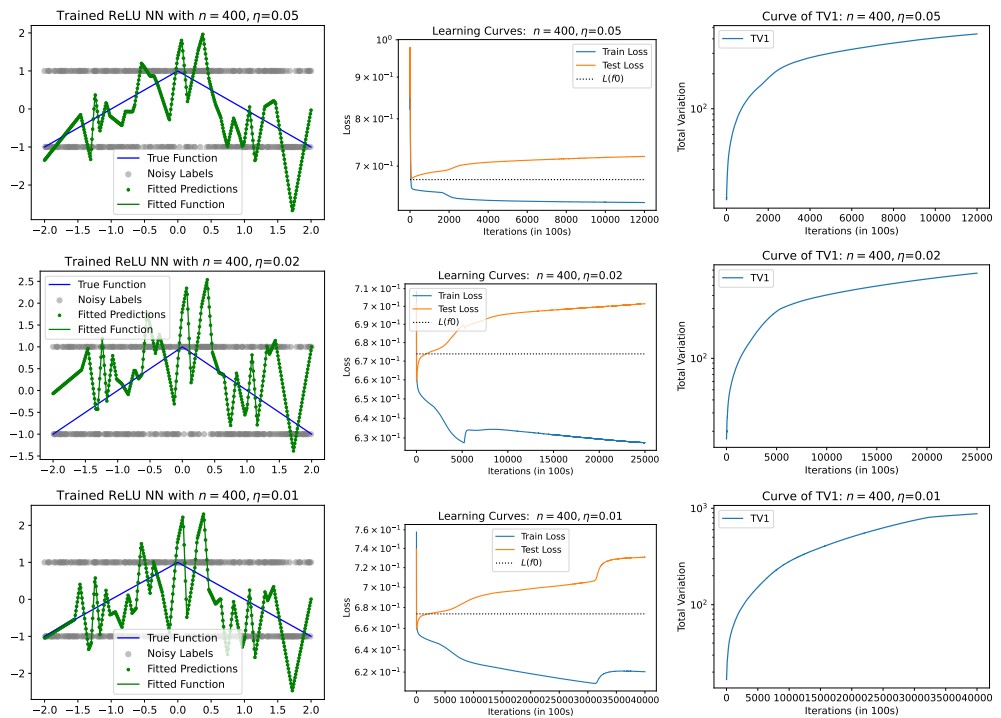

Figure 10: Illustration of the solutions gradient descent with learning rate $\eta$ converges to ($n = 400$: Part III). As $\eta$ decreases further, the fitted function starts to overfit, while the overfitting is mild compared to the cases with fewer samples (Figure 4 and Figure 7).

In conclusion, for datasets with different numbers of data, as $\eta$ decreases, the fitted function goes from simple to complex. Moreover, as $\eta$ decreases further, the fitted function starts to overfit. Among different cases, for the case with larger number of data, the overfitting is less catastrophic.

## A.2 LOSS & COMPLEXITY OF THE LEARNED FUNCTIONS VS STEP SIZE & NUMBER OF DATA

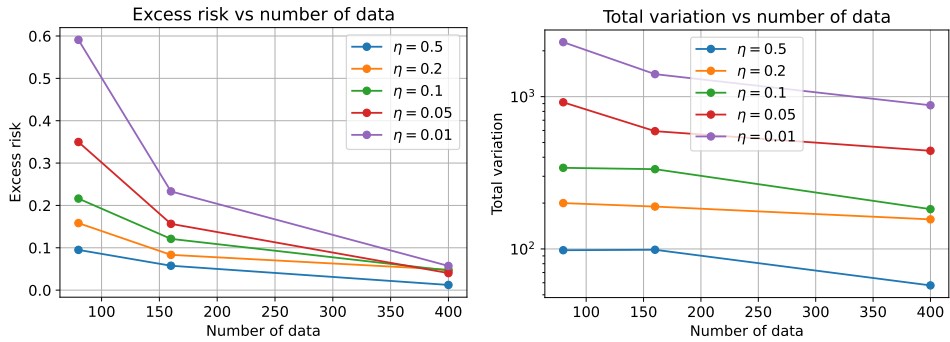

Figure 11: Loss & complexity of the learned function vs number of data. For a fixed $\eta$, as $n$ goes larger, the 1st order total variation does not increase while the excess risk converges to 0 quickly.

In Figure 12, for datasets with different numbers of data, as $\eta$ decreases, the training loss becomes smaller. Meanwhile, tuning $\eta$ gives the classical U-shape for excess risk and excess error, which is consistent with Theorem 3.7. At the same time, as $\eta$ becomes small, $\lambda_{\max}(Hessian)$ approximates $2/\eta$, thus justifying our consideration of "edge of stability" and stable solutions defined as equation 2. Lastly, the 1st order total variation ($\mathrm{TV}^{(1)}$) of the learned function increases monotonically as $n \to 0$, which supports Theorem 3.1 and 3.5. Figure 13 indicates that the weighted TV1 constraint is indeed making the learned function sparse (in the coefficient vector).

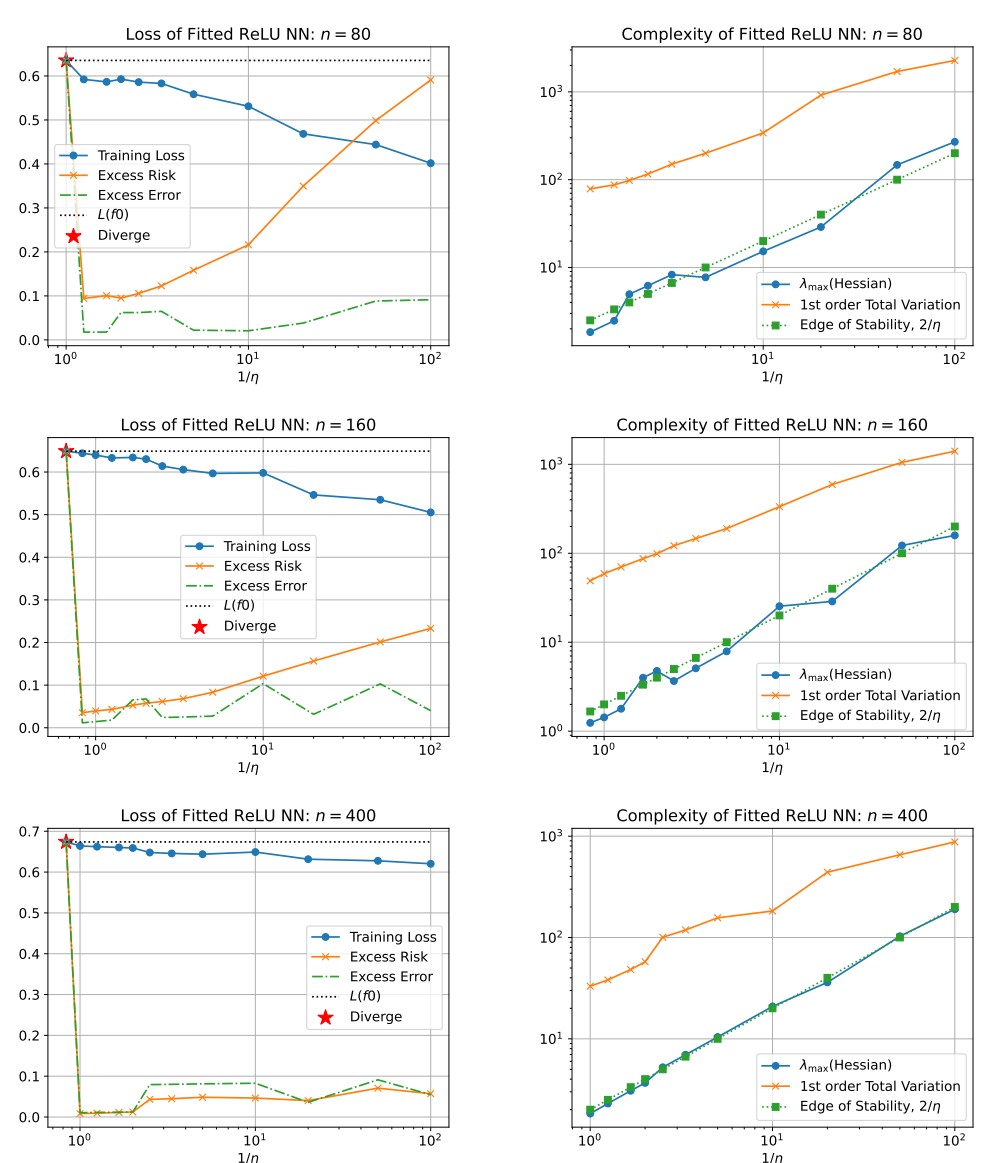

Figure 12: The **left panel** plots the performance (training loss, excess risk, excess error) of the learned functions under different learning rates. The **right panel** illustrates the impact of learning rate on the flatness ($\lambda_{\max}(Hessian)$) and complexity ($\text{TV}^{(1)}$) of the learned function.

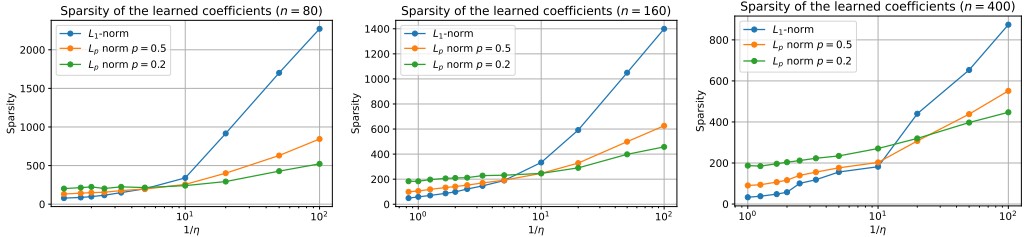

Figure 13: Sparsity of the learned coefficients in sparse $L_1$ and $L_p$ norm as the function of $1/\eta$.

### A.3 REPRESENTATION LEARNING: VISUALIZATION OF LEARNED BASIS FUNCTIONS

In this part, we visualize the basis functions at initialization and after training with different step sizes.

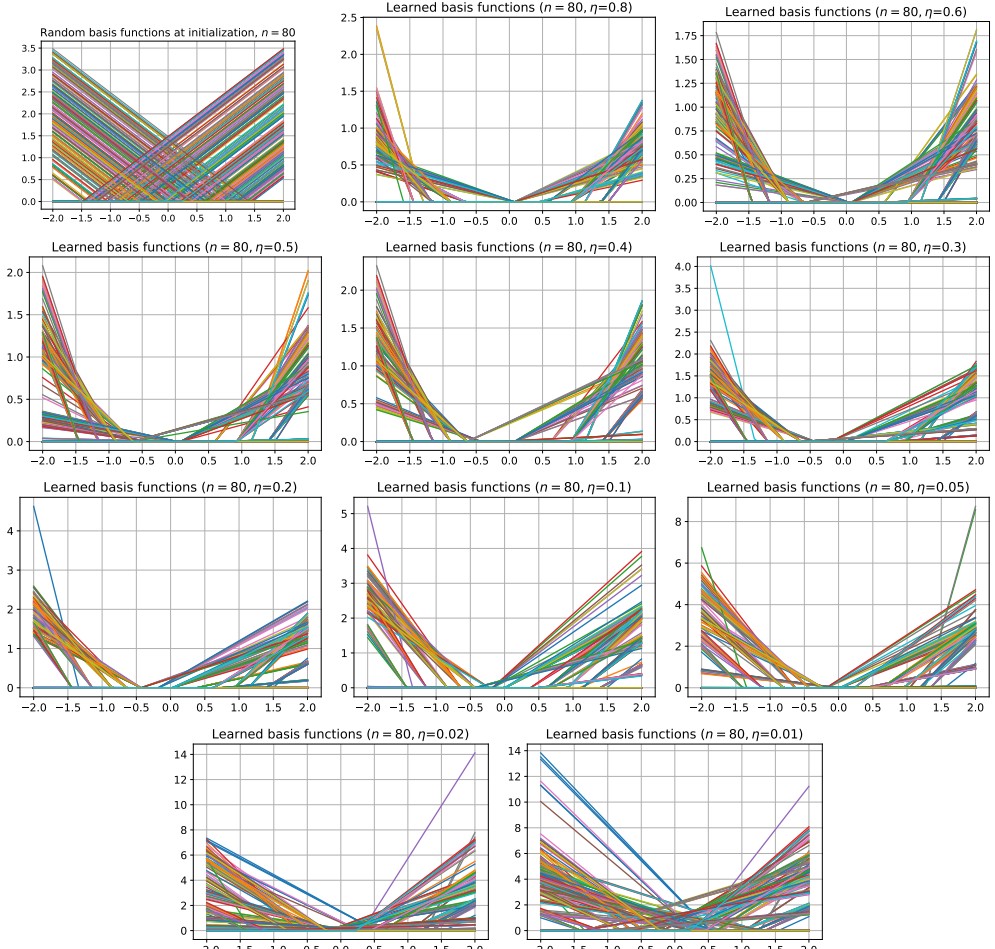

Figure 14: Illustration of the learned basis functions with learning rate $\eta$. It is clear from the figures that there are substantial representation learning, where the learned basis functions are very different from initialization. Also, the structure of active basis functions gets simpler as the learning rate $\eta$ gets bigger.

There are several interesting insights from Figure 14. First, the learned basis functions are very different from the initialization, so a lot of representation learning is happening, in comparison to the "kernel" regime in which nearly no representation learning is happening. Second, as $\eta$ gets smaller, the complexity of the learned basis functions and the number of knots in the fitted function increase. Third, even with large learning rate, the number of active basis functions is still large, which is quite different from the representation leaning for training a two-layer ReLU NN to minimize the MSE loss (Qiao et al., 2024, Figure 7), where large learning rate leads to very few active basis functions. Lastly, the learned basis function displays a strong "clustering" effect in the sense that despite overparameterization, many learned basis functions end up having the same activation threshold on the data support.

### A.4 EXAMPLE OF CONVERGENCE TO INTERPOLATING SOLUTIONS

In this part, we construct an example to show that when the number of data points $n$ is small, GD with large learning rate can actually converge to interpolating solutions at infinity, which supports the negative result in Appendix E.

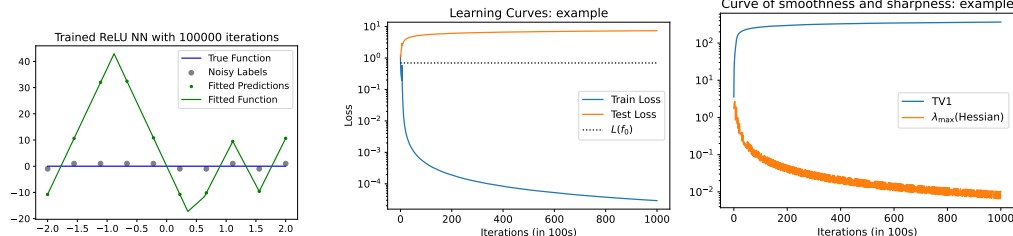

Figure 15: Illustration of the output function, learning curves and the curve of smoothness & sharpness. Interestingly, "Edge of Stability" does not happen and $\lambda_{\max}$(Hessian) decreases to 0.

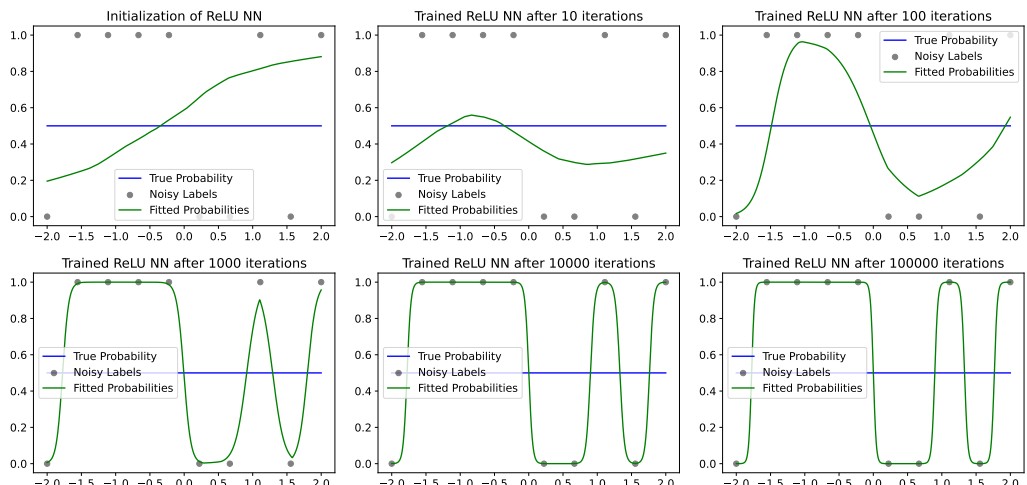

Figure 16: Illustration of the training dynamics. We plot true probability (*i.e.* $\sigma(f_0)$) vs fitted probability (*i.e.* $\sigma(f_\theta)$). Also, we shift the labels from $\{-1, 1\}$ to $\{0, 1\}$ for a better illustration. The learned function tends to interpolate the data after a small number of iterations.

In the example, the design set $\{x_i\}_{i=1}^n$ is chosen to be $n$ equally spaced points in $[-2, 2]$ with $n = 10$. The ground-truth function $f_0(x) = 0$ for any $x \in [-2, 2]$. The two-layer ReLU NN is mildly over-parameterized with $k = 20$ while initialized randomly. During the training, we minimize the logistic loss via GD with learning rate $\eta = 1$, and we train for 100000 iterations.

In the **left panel** of Figure 15, the final output function (after 100000 iterations) tends to be interpolating and heavily overfitting. As a result, the training loss converges to 0, as shown in the **middle panel**. Most interestingly, as supported by the **right panel** of Figure 15, GD actually *bypasses* the "Edge of Stability" regime. Instead of oscillating around $2/\eta = 2$, the largest eigenvalue of the Hessian matrix tends to converge to 0, although in a non-monotonic way. The performance is consistent with our Theorem 3.2, which claims that interpolating solutions can be arbitrarily flat. Together with Figure 16, the training dynamics also corroborate the implication of Theorem 3.2: the more confident the correct prediction is, the flatter the landscape.

**The case with large $n$.** When $n$ becomes large, GD with a constant learning rate will probably converge to some finite function if initialized randomly, as shown in Figure 2. However, flat yet interpolating solutions still exist. Moreover, as shown in Figure 17, when the ReLU NN is initialized to be *correct and confident*, GD with constant learning rate will further push the predictions to be more confident, although very slowly. The main observations are: (1) the prediction for the part with consecutive 1's or -1's as labels will become more confident at a faster rate compared to the part with joint 1 and -1; (2) as we initialize the NN to be more confident, the scale of the predictions will grow at a slower speed.

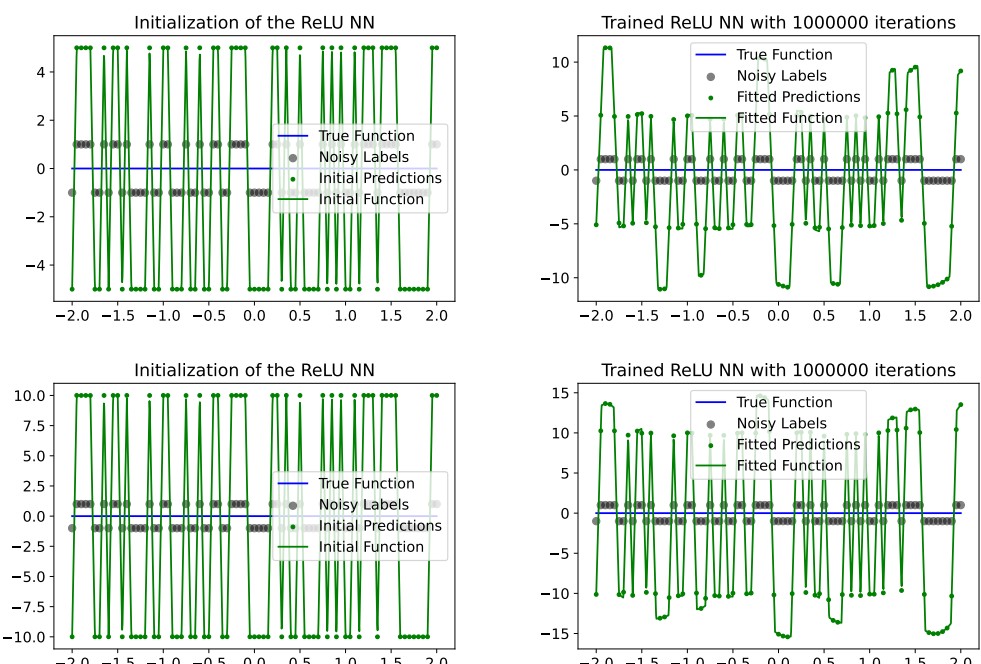

Figure 17: Illustration of the confidently initialized NN and the trained NN after 1000000 iterations. For both rows, $n$ is chosen to be 80 while $k$ is slightly larger than $n$. The learning rate $\eta$ is tuned to be as large as possible such that GD does not diverge. The **first row** initializes the NN such that $f_\theta(x_i)y_i \approx 5$ for all $i \in [n]$, and trains via GD with $\eta = 0.005$. The **second row** initializes the NN such that $f_\theta(x_i)y_i \approx 10$ for all $i \in [n]$, and trains via GD with $\eta = 0.5$.

# B   TECHNICAL LEMMAS

## B.1   CONNECTION BETWEEN STABILITY AND FLATNESS

For a twice differentiable minimum $\theta^\star$, direct application of Taylor's expansion implies

$$\mathcal{L}(\theta) \approx \mathcal{L}(\theta^\star) + (\theta - \theta^\star)^T \nabla \mathcal{L}(\theta^\star) + \frac{1}{2}(\theta - \theta^\star)^T \nabla^2 \mathcal{L}(\theta^\star)(\theta - \theta^\star), \qquad (9)$$

where $\nabla^2 \mathcal{L}$ is the Hessian matrix. Note that $\nabla \mathcal{L}(\theta^\star) = 0$. Therefore, as $\theta_t$ converges to $\theta^\star$, we can approximate the formula for GD as $\theta_{t+1} \approx \theta_t - \eta \left( \nabla^2 \mathcal{L}(\theta^\star)(\theta_t - \theta^\star) \right)$. Motivated by the approximation, Wu et al. (2018) first brought up the notion of linear stability.

**Definition B.1** (Linear stability). With the update rule $\theta_{t+1} = \theta_t - \eta \left( \nabla^2 \mathcal{L}(\theta^\star)(\theta_t - \theta^\star) \right)$, a twice differentiable local minimum $\theta^\star$ of $\mathcal{L}$ is said to be $\epsilon$ linearly stable if for any $\theta_0$ in the $\epsilon$-ball $\mathcal{B}_\epsilon(\theta^\star)$, it holds that $\limsup_{t \to \infty} \|\theta_t - \theta^\star\| \leq \epsilon$.

We remark that since the update of GD does not have any randomness, we remove the expectation before $\|\theta_t - \theta^\star\|$ (which appears in the definition in Wu et al. (2018); Mulayoff et al. (2021)). The definition of "linear stability" ensures that the optimization could stabilize around the local minimum $\theta^\star$ once it gets close enough to $\theta^\star$. The following lemma connects linear stability to the flatness of the local minima by showing that the set of stable minima is equivalent to the set of flat local minima whose largest eigenvalue of Hessian matrix is bounded by $2/\eta$.

**Lemma B.2.** *Consider the update rule in Definition B.1, for any $\epsilon > 0$, a local minimum $\theta^\star$ is an $\epsilon$ linearly stable minimum of $\mathcal{L}$ if and only if $\lambda_{\max}(\nabla^2 \mathcal{L}(\theta^\star)) \leq \frac{2}{\eta}$.*

*Proof of Lemma B.2.* It holds that

$$\begin{aligned}
\theta_{t+1} - \theta^\star &= \theta_t - \theta^\star - \eta \nabla^2 \mathcal{L}(\theta^\star)(\theta_t - \theta^\star) \\
&= \left( I - \eta \nabla^2 \mathcal{L}(\theta^\star) \right)(\theta_t - \theta^\star),
\end{aligned} \qquad (10)$$

where the first equation is from the update rule in Definition B.1. As a result,

$$\theta_t - \theta^\star = \left(I - \eta\nabla^2\mathcal{L}(\theta^\star)\right)^t (\theta_0 - \theta^\star). \tag{11}$$

On one hand, if $\lambda_{\max}(\nabla^2\mathcal{L}(\theta^\star)) \leq \frac{2}{\eta}$, it holds that

$$\|\theta_t - \theta^\star\| \leq \left\|I - \eta\nabla^2\mathcal{L}(\theta^\star)\right\|_2^t \cdot \|\theta_0 - \theta^\star\| \leq \|\theta_0 - \theta^\star\|, \tag{12}$$

where the second inequality is because all the eigenvalues of $I - \eta\nabla^2\mathcal{L}(\theta^\star)$ is bounded between $[-1, 1]$. Therefore, $\theta^\star$ is $\epsilon$ linearly stable for any $\epsilon$.

On the other hand, if $\theta^\star$ is $\epsilon$ linearly stable, we choose $\theta_0$ such that $\frac{\theta_0 - \theta^\star}{\|\theta_0 - \theta^\star\|}$ is the top eigenvector of $\nabla^2\mathcal{L}(\theta^\star)$ and $\|\theta_0 - \theta^\star\| = \epsilon$. Then we have

$$\|\theta_t - \theta^\star\| = \left|1 - \eta\lambda_{\max}\left(\nabla^2\mathcal{L}(\theta^\star)\right)\right|^t \cdot \epsilon, \tag{13}$$

which implies that $\limsup_{t\to\infty}\left|1 - \eta\lambda_{\max}\left(\nabla^2\mathcal{L}(\theta^\star)\right)\right|^t \leq 1$, and therefore $\lambda_{\max}(\nabla^2\mathcal{L}(\theta^\star)) \leq \frac{2}{\eta}$, which finishes the proof. $\qquad\square$

### B.2 PROPERTIES OF LOGISTIC LOSS

Recall that the distribution $\mathcal{B}(x)$ is our data generation assumption equation 3. The first property is that the optimal prediction function $f^\star$ is the ground-truth function $f_0$.

**Lemma B.3.** *For any feature $x$ in the dataset, the optimal prediction function $f^\star$ satisfies that* $f^\star(x) = \operatorname{argmin}_f \mathbb{E}_{y\sim\mathcal{B}(x)}\log\left(1 + e^{-yf}\right) = f_0(x)$.

*Proof of Lemma B.3.* Let $p = \sigma(f_0(x)) = \frac{e^{f_0(x)}}{1 + e^{f_0(x)}}$, $f^\star(x)$ should be

$$\begin{aligned}
f^\star(x) &= \operatorname{argmin}_f p \cdot \log\left(1 + e^{-f}\right) + (1 - p) \cdot \log\left(1 + e^f\right) \\
&= \log\left(\frac{p}{1 - p}\right) = f_0(x),
\end{aligned} \tag{14}$$

where the second equation is derived by taking derivative of the loss. $\qquad\square$

**Remark B.4.** For the fixed design (feature) setting, $f^\star$ must be the same as $f_0$ for the features $x$ from the dataset, while $f^\star$ can take any values for other features.

Meanwhile, the $\log$ function has the following property.

**Lemma B.5.** *For $x > 0$, the following inequality holds:*

$$\frac{x}{1 + x} \leq \log(1 + x) \leq x. \tag{15}$$

*Proof of Lemma B.5.* The inequality directly results from the fact $(\log(1 + t))' = \frac{1}{1+t}$. $\qquad\square$

### B.3 METRIC ENTROPY OF THE BV FUNCTION CLASS

Recall that we assume the target function belongs to the first order bounded variation class

$$f_0 \in \mathrm{BV}^{(1)}(B, C_n) := \left\{ f : [-x_{\max}, x_{\max}] \to \mathbb{R} \;\middle|\; \max_x |f(x)| \leq B, \int_{-x_{\max}}^{x_{\max}} |f''(x)|dx \leq C_n \right\}. \tag{16}$$

In this part, we characterize the complexity of such function class via the notion of metric entropy. We first state the definition of metric entropy (Wainwright, 2019) below.

**Definition B.6.** For a set $\mathbb{T}$ with a corresponding metric $\rho(\cdot, \cdot)$, let $N(\epsilon, \mathbb{T}, \rho)$ denote the $\epsilon$-covering number of $\mathbb{T}$ under metric $\rho$. Then the metric entropy of $\mathbb{T}$ with respect to $\rho$ is $\log N(\epsilon, \mathbb{T}, \rho)$.

For examples of metric entropy, we refer the readers to Chapter 5 of Wainwright (2019). Next we bound the metric entropy of $\mathrm{BV}^{(1)}(1,1)$, which is helpful for bounding the metric entropy of $\mathrm{BV}(1)$ function class with other (larger) parameters. Regarding the metric, we consider the $\ell_\infty$ metric defined as $\rho_\infty(f,g) = \sup_{x \in \Omega} |f(x) - g(x)|$ over the domain $\Omega$. As a short hand, we denote $\rho_\infty$ by $\|\cdot\|_\infty$.

**Lemma B.7.** *[Lemma C.4 of Qiao et al. (2024)] Assume the set* $\mathbb{T}_1 = \left\{ f : [-1,1] \to \mathbb{R} \ \middle| \ \int_{-1}^{1} |f''(x)| dx \leq 1, \ |f(x)| \leq 1 \right\}$ *and the metric is the $\ell_\infty$ distance $\|\cdot\|_\infty$, then there exists a universal constant $C_1 > 0$ such that for any $\epsilon > 0$, the metric entropy of $(\mathbb{T}_1, \|\cdot\|_\infty)$ satisfies*

$$\log N(\epsilon, \mathbb{T}_1, \|\cdot\|_\infty) \leq C_1 \epsilon^{-\frac{1}{2}}. \tag{17}$$

**Remark B.8.** The idea behind the proof is that the set $\mathbb{T}_1$ is a bounded subset of the Besov space $B_{1,\infty}^2$, and the metric entropy of such bounded subset is summarized in Corollary 2 of Nickl & Pötscher (2007). A rigorous proof of Lemma B.7 is stated in Qiao et al. (2024). We refer interested readers to Edmunds & Triebel (1996) for a detailed introduction of Besov space, DeVore & Lorentz (1993) for the connection between bounded total variation class and Besov space, and Nickl & Pötscher (2007) for more discussions about the metric entropy of Besov space.

## C CALCULATION OF GRADIENT AND HESSIAN MATRIX

In this section, we calculate the gradient and Hessian matrix of $\mathcal{L}(\theta)$ with respect to $\theta$. Recall that $\mathcal{L}(\theta) = \frac{1}{n} \sum_{i=1}^{n} \log\left(1 + e^{-y_i f_\theta(x_i)}\right)$. Then the gradient of $\mathcal{L}(\theta)$ is calculated as follows:

$$\nabla_\theta \mathcal{L}(\theta) = \frac{1}{n} \sum_{i=1}^{n} \frac{-y_i e^{-y_i f(x_i)}}{1 + e^{-y_i f(x_i)}} \nabla_\theta f(x_i), \tag{18}$$

where $f = f_\theta$ for the ease of presentation. Moreover, we further calculate the Hessian matrix:

$$\begin{aligned} \nabla_\theta^2 \mathcal{L}(\theta) &= \frac{1}{n} \sum_{i=1}^{n} \frac{-y_i e^{-y_i f(x_i)}}{1 + e^{-y_i f(x_i)}} \nabla_\theta^2 f(x_i) + \frac{1}{n} \sum_{i=1}^{n} \frac{y_i^2 e^{-y_i f(x_i)}}{\left(1 + e^{-y_i f(x_i)}\right)^2} \nabla_\theta f(x_i) \nabla_\theta f(x_i)^T \\ &= \frac{1}{n} \sum_{i=1}^{n} \frac{-y_i e^{-y_i f(x_i)}}{1 + e^{-y_i f(x_i)}} \nabla_\theta^2 f(x_i) + \frac{1}{n} \sum_{i=1}^{n} \frac{e^{-y_i f(x_i)}}{\left(1 + e^{-y_i f(x_i)}\right)^2} \nabla_\theta f(x_i) \nabla_\theta f(x_i)^T, \end{aligned} \tag{19}$$

where the second equation holds because $y_i^2 = 1$.

Then what remains is to calculate $\nabla_\theta f_\theta(x)$ and $\nabla_\theta^2 f_\theta(x)$. Recall that $f_\theta(x) = \sum_{i=1}^{k} w_i^{(2)} \phi\left(w_i^{(1)} x + b_i^{(1)}\right) + b^{(2)}$ where $\phi(x) = \max\{x, 0\}$. Also, we denote $\theta = (w_1^{(1)}, \cdots, w_k^{(1)}, b_1^{(1)}, \cdots, b_k^{(1)}, w_1^{(2)}, \cdots, w_k^{(2)}, b^{(2)})^T$.

### C.1 CALCULATION OF $\nabla_\theta f_\theta(x)$

Resulting from direct calculation, for a given $x \in [-x_{\max}, x_{\max}]$ we have

$$\begin{cases} \nabla_{w_i^{(1)}} f_\theta(x) = x w_i^{(2)} \mathbb{1}\left(w_i^{(1)} x + b_i^{(1)} > 0\right), \ \forall i \in [k] \\ \nabla_{b_i^{(1)}} f_\theta(x) = w_i^{(2)} \mathbb{1}\left(w_i^{(1)} x + b_i^{(1)} > 0\right), \ \forall i \in [k] \\ \nabla_{w_i^{(2)}} f_\theta(x) = \phi\left(w_i^{(1)} x + b_i^{(1)}\right) = \left(w_i^{(1)} x + b_i^{(1)}\right) \mathbb{1}\left(w_i^{(1)} x + b_i^{(1)} > 0\right), \ \forall i \in [k] \\ \nabla_{b^{(2)}} f_\theta(x) = 1 \end{cases} \tag{20}$$

### C.2 CALCULATION OF $\nabla_\theta^2 f_\theta(x)$

In this part, we calculate $\nabla_\theta^2 f_\theta(x)$ for a given $x \in [-x_{\max}, x_{\max}]$. Below we calculate $\frac{\partial^2 f_\theta(x)}{\partial \theta_i \partial \theta_j}$.

First of all, if $\theta_i = b^{(2)}$ or $\theta_j = b^{(2)}$, $\frac{\partial^2 f_\theta(x)}{\partial \theta_i \partial \theta_j} = 0$. Then it remains to calculate $\frac{\partial^2 f_\theta(x)}{\partial \theta_i \partial \theta'_j}$ where $i, j \in [k]$ and $\theta, \theta' \in \{w^{(1)}, b^{(1)}, w^{(2)}\}$. It is obvious that if $i \neq j$, $\frac{\partial^2 f_\theta(x)}{\partial \theta_i \partial \theta'_j} = 0$. Therefore, we only calculate the case when $j = i$. Let $\delta$ denote the Dirac function, it holds that:

$$
\begin{cases}
\frac{\partial^2 f_\theta(x)}{\partial w_i^{(1)} \partial w_i^{(1)}} = w_i^{(2)} x^2 \delta\left(w_i^{(1)} x + b_i^{(1)}\right), & \forall\, i \in [k] \\
\frac{\partial^2 f_\theta(x)}{\partial w_i^{(1)} \partial b_i^{(1)}} = \frac{\partial^2 f_\theta(x)}{\partial b_i^{(1)} \partial w_i^{(1)}} = x w_i^{(2)} \delta\left(w_i^{(1)} x + b_i^{(1)}\right), & \forall\, i \in [k] \\
\frac{\partial^2 f_\theta(x)}{\partial b_i^{(1)} \partial b_i^{(1)}} = w_i^{(2)} \delta\left(w_i^{(1)} x + b_i^{(1)}\right), & \forall\, i \in [k] \\
\frac{\partial^2 f_\theta(x)}{\partial w_i^{(2)} \partial w_i^{(2)}} = 0, & \forall\, i \in [k] \\
\frac{\partial^2 f_\theta(x)}{\partial w_i^{(1)} \partial w_i^{(2)}} = \frac{\partial^2 f_\theta(x)}{\partial w_i^{(2)} \partial w_i^{(1)}} = x \mathbb{1}\left(w_i^{(1)} x + b_i^{(1)} > 0\right), & \forall\, i \in [k] \\
\frac{\partial^2 f_\theta(x)}{\partial b_i^{(1)} \partial w_i^{(2)}} = \frac{\partial^2 f_\theta(x)}{\partial w_i^{(2)} \partial b_i^{(1)}} = \mathbb{1}\left(w_i^{(1)} x + b_i^{(1)} > 0\right), & \forall\, i \in [k]
\end{cases}
\tag{21}
$$

Due to the existence of the Dirac function, the Hessian matrix is not well-defined in general. However, we only consider the function $f_\theta$ that is twice differentiable with respect to $\theta$ (*i.e.* the knots of $f$ do not coincide with $x$), which implies that all the Dirac functions take the value 0. In this case,

$$
\begin{cases}
\frac{\partial^2 f_\theta(x)}{\partial w_i^{(1)} \partial w_i^{(1)}} = 0, & \forall\, i \in [k] \\
\frac{\partial^2 f_\theta(x)}{\partial w_i^{(1)} \partial b_i^{(1)}} = \frac{\partial^2 f_\theta(x)}{\partial b_i^{(1)} \partial w_i^{(1)}} = 0, & \forall\, i \in [k] \\
\frac{\partial^2 f_\theta(x)}{\partial b_i^{(1)} \partial b_i^{(1)}} = 0, & \forall\, i \in [k]
\end{cases}
\tag{22}
$$

## C.3 Upper Bounding the Operator Norm

In this part, we upper bound the operator norm of the Hessian matrix calculated above. The following lemma provides an upper bound for $\left|v^T \nabla^2 f_\theta(x) v\right|$ under the constraint that $\|v\|_2 = 1$.

**Lemma C.1.** *Assume that $f_\theta(x)$ is twice differentiable with respect to $\theta$ and $x \in [-x_{\max}, x_{\max}]$, for any $v$ such that $\|v\|_2 = 1$, it holds that*

$$
\left|v^T \nabla^2 f_\theta(x) v\right| \leq 2 \max\{x_{\max}, 1\}.
\tag{23}
$$

*Proof of Lemma C.1.* Assume $v = (\alpha_1, \cdots, \alpha_k, \beta_1, \cdots, \beta_k, \gamma_1, \cdots, \gamma_k, \iota)^T \in \mathbb{R}^{3k+1}$ such that $\sum_{i=1}^k (\alpha_i^2 + \beta_i^2 + \gamma_i^2) + \iota^2 = 1$. Note that the Hessian matrix $\nabla_\theta^2 f_\theta(x)$ follows the structure:

$$
\nabla_\theta^2 f_\theta(x) = \begin{pmatrix}
A_{w^{(1)} w^{(1)}} & A_{w^{(1)} b^{(1)}} & A_{w^{(1)} w^{(2)}} & A_{w^{(1)} b^{(2)}} \\
A_{b^{(1)} w^{(1)}} & A_{b^{(1)} b^{(1)}} & A_{b^{(1)} w^{(2)}} & A_{b^{(1)} b^{(2)}} \\
A_{w^{(2)} w^{(1)}} & A_{w^{(2)} b^{(1)}} & A_{w^{(2)} w^{(2)}} & A_{w^{(2)} b^{(2)}} \\
A_{b^{(2)} w^{(1)}} & A_{b^{(2)} b^{(1)}} & A_{b^{(2)} w^{(2)}} & A_{b^{(2)} b^{(2)}}
\end{pmatrix}
\tag{24}
$$

where $A_{w^{(1)} w^{(1)}}, A_{w^{(1)} b^{(1)}}, A_{b^{(1)} w^{(1)}}, A_{b^{(1)} b^{(1)}}, A_{w^{(2)} w^{(2)}} \in \mathbb{R}^{k \times k}$, $A_{w^{(1)} b^{(2)}}, A_{b^{(1)} b^{(2)}}, A_{w^{(2)} b^{(2)}} \in \mathbb{R}^{k \times 1}$, $A_{b^{(2)} w^{(1)}}, A_{b^{(2)} b^{(1)}}, A_{b^{(2)} w^{(2)}} \in \mathbb{R}^{1 \times k}$ and $A_{b^{(2)} b^{(2)}} \in \mathbb{R}$ are all zero matrices. Meanwhile, $A_{w^{(1)} w^{(2)}}, A_{b^{(1)} w^{(2)}}, A_{w^{(2)} w^{(1)}}, A_{w^{(2)} b^{(1)}} \in \mathbb{R}^{k \times k}$ are all diagonal matrices whose non-zero elements are between $[-\max\{x_{\max}, 1\}, \max\{x_{\max}, 1\}]$. Therefore, it holds that:

$$
\left|v^T \nabla^2 f_\theta(x) v\right| \leq 2 \max\{x_{\max}, 1\} \sum_{i=1}^k \left(|\alpha_i \gamma_i| + |\beta_i \gamma_i|\right)
$$

$$
\leq 2 \max\{x_{\max}, 1\} \left(\sqrt{\sum_{i=1}^k \alpha_i^2 \cdot \sum_{i=1}^k \gamma_i^2} + \sqrt{\sum_{i=1}^k \beta_i^2 \cdot \sum_{i=1}^k \gamma_i^2}\right)
\tag{25}
$$

$$
\leq 2 \max\{x_{\max}, 1\},
$$

where the second inequality holds because of Cauchy-Schwarz inequality. The last inequality results from $x(1-x) \leq \frac{1}{4}$. $\qquad \square$

## C.4 DERIVATIVES OF POPULATION-LEVEL LOSS

Previously, we show that the minimizer of the population-level loss is the ground-truth function. In this part, we calculate the first and second order derivatives of the population-level loss. Recall that for a fixed feature $x$, the label $y$ is sampled from the following distribution:

$$y = \begin{cases} 1 & \text{with probability } p = \sigma(f_0(x)) \\ -1 & \text{with probability } 1 - p \end{cases} \tag{26}$$

Therefore, for a fixed feature $x$ (with $p = \sigma(f_0(x)) = \frac{e^{f_0(x)}}{1+e^{f_0(x)}}$) and the prediction value $f$ at $x$, the population-level loss at $x$ is defined as

$$\bar{l}_x(f) := p\log(1 + e^{-f}) + (1 - p)\log(1 + e^f). \tag{27}$$

Then the first order derivative of $\bar{l}_x(f)$ with respect to $f$ can be calculated as below.

$$\begin{aligned} \bar{l}'_x(f) =& p \cdot \frac{-1}{e^f + 1} + (1 - p) \cdot \frac{e^f}{1 + e^f} \\ =& \frac{e^f - e^{f_0(x)}}{(1 + e^f)(1 + e^{f_0(x)})} \\ =& \frac{1}{1 + e^{f_0(x)}} - \frac{1}{1 + e^f}, \end{aligned} \tag{28}$$

where the second equation holds because $p = \sigma(f_0(x)) = \frac{e^{f_0(x)}}{1+e^{f_0(x)}}$. Meanwhile, we have

$$\bar{l}''_x(f) = \frac{e^f}{(1 + e^f)^2} > 0. \tag{29}$$

We highlight that the second order derivative is always positive and independent of $f_0(x)$. As $|f|$ becomes smaller, the value of $\bar{l}''_x(f)$ will become larger.

Finally, due to direct calculation, the relation between $\bar{l}_x$ and $\bar{\mathcal{L}}$ is

$$\bar{\mathcal{L}}(f) = \mathbb{E}_{x \sim \mathcal{D}} \mathbb{E}_{y \sim \mathcal{B}(x)} \log\left(1 + e^{-yf(x)}\right) = \mathbb{E}_{x \sim \mathcal{D}} \bar{l}_x(f(x)). \tag{30}$$

# D PROOF OF THEOREM 3.1

In this section, we prove the implicit bias of minima stability. We begin with a decomposition of the Hessian matrix.

## D.1 DECOMPOSITION OF THE HESSIAN MATRIX

Suppose the logistic loss $\mathcal{L}(\theta) = \mathcal{L}(f_\theta)$ is twice differentiable at $\theta$, it holds that

$$\lambda_{\max}\left(\nabla_\theta^2 \mathcal{L}(\theta)\right) \geq v^T \nabla_\theta^2 \mathcal{L}(\theta) v$$

$$= \lambda_{\max}\underbrace{\left(\frac{1}{n}\sum_{i=1}^n \frac{e^{-y_i f(x_i)}}{\left(1 + e^{-y_i f(x_i)}\right)^2}\nabla_\theta f(x_i)\nabla_\theta f(x_i)^T\right)}_{(i)} + \underbrace{\frac{1}{n}\sum_{i=1}^n \frac{-y_i e^{-y_i f(x_i)}}{1 + e^{-y_i f(x_i)}} v^T \nabla_\theta^2 f(x_i) v}_{(ii)}, \tag{31}$$

where $v$ is the unit eigenvector of the largest eigenvalue of $\frac{1}{n}\sum_{i=1}^n \frac{e^{-y_i f(x_i)}}{\left(1+e^{-y_i f(x_i)}\right)^2}\nabla_\theta f(x_i)\nabla_\theta f(x_i)^T$.

Then it remains to handle the terms (i) and (ii).

## D.2 UPPER BOUNDING THE TERM (II)

We upper bound the absolute value of (ii) by the empirical loss. It holds that

$$\begin{aligned} |(ii)| \leq& \frac{1}{n}\sum_{i=1}^n \frac{e^{-y_i f(x_i)}}{1 + e^{-y_i f(x_i)}} \left|v^T \nabla_\theta^2 f(x_i) v\right| \\ \leq& 2\max\{x_{\max}, 1\} \cdot \frac{1}{n}\sum_{i=1}^n \log\left(1 + e^{-y_i f(x_i)}\right) \\ =& 2\max\{x_{\max}, 1\} \cdot \mathcal{L}(f), \end{aligned} \tag{32}$$

where the second inequality results from the uniform upper bound of $\left|v^T \nabla_\theta^2 f(x)v\right|$ (Lemma C.1) and Lemma B.5. As the optimization converges and the empirical loss $\mathcal{L}(f)$ becomes small, the term $|(ii)|$ will also be small.

### D.3 HANDLING THE TERM (I)

Let $\Phi = [\nabla_\theta f_\theta(x_1), \cdots, \nabla_\theta f_\theta(x_n)] \in \mathbb{R}^{(3k+1)\times n}$ and $D = \text{Diag} \left\{ \frac{e^{-y_i f(x_i)}}{\left(1+e^{-y_i f(x_i)}\right)^2} \right\}_{i=1}^n \in \mathbb{R}^{n\times n}$, a diagonal matrix whose $i$-th diagonal element is $\frac{e^{-y_i f(x_i)}}{\left(1+e^{-y_i f(x_i)}\right)^2}$. Then we have $(i) = \lambda_{\max}\left(\frac{1}{n}\Phi D\Phi^T\right)$, and we want to connect it to $\int_{-x_{\max}}^{x_{\max}} |f''(x)|\, dx$. Lemma 4 in Mulayoff et al. (2021) states that $\lambda_{\max}\left(\frac{1}{n}\Phi\Phi^T\right) \geq 1 + 2\int_{-x_{\max}}^{x_{\max}} |f''(x)|\, g(x)dx$ for some weight function $g$. However, in our case, the diagonal components of $D$ can be arbitrarily close to 0, and therefore $\lambda_{\max}\left(\frac{1}{n}\Phi D\Phi^T\right)$ can be much smaller than $\lambda_{\max}\left(\frac{1}{n}\Phi\Phi^T\right)$, raising some technical challenges.

Note that the value of $\frac{e^{-y_i f(x_i)}}{\left(1+e^{-y_i f(x_i)}\right)^2}$ being large is equivalent to the value of $|f(x_i)|$ being small. Therefore, we only consider the data points (*i.e.* the corresponding diagonal components in $D$) where $|f_\theta(x)|$ is small (*i.e.* the prediction is not very certain). For the other components in $D$, we simply lower bound them by 0. More specifically, we choose a threshold $\gamma > 0$ which is a small constant, and define the "uncertain set" $\mathcal{A}_\gamma$ as:

$$\mathcal{A}_\gamma = \left\{ x_i \in \mathcal{D} \;\middle|\; |f_\theta(x_i)| \leq \gamma \right\} = \left\{ x_i \in \mathcal{D} \;\middle|\; \frac{e^{-y_i f(x_i)}}{\left(1+e^{-y_i f(x_i)}\right)^2} \geq \frac{e^{-\gamma}}{\left(1+e^{-\gamma}\right)^2} := \Gamma(\gamma) \right\}. \tag{33}$$

According to direct calculation, we have

$$\lambda_{\max}\left(\frac{1}{n}\Phi D\Phi^T\right) = \max_{v\in S^{3k}} \frac{1}{n} v^T \Phi D\Phi^T v = \frac{1}{n}\max_{v\in S^{3k}} \left\| \left(\Phi D^{\frac{1}{2}}\right)^T v\right\|_2^2 = \frac{1}{n}\max_{u\in S^{n-1}} \left\|\Phi D^{\frac{1}{2}} u\right\|_2^2. \tag{34}$$

We can choose $u = [u_1, \cdots, u_n]^T \in \mathbb{R}^n$ to be (for some constant $\alpha > 0$):

$$u_j = \begin{cases} \alpha D_{j,j}^{-\frac{1}{2}}, & x_j \in \mathcal{A}_\gamma, \\ 0, & \text{otherwise,} \end{cases} \tag{35}$$

such that $\|u\|_2 = 1$. Defining $n_\gamma = |\mathcal{A}_\gamma|$ (the number of uncertain data points), it is clear that $\alpha \geq \sqrt{\frac{\Gamma(\gamma)}{n_\gamma}}$.

Let $I_{j,i} = \mathbb{1}\left(w_i^{(1)} x_j + b_i^{(1)} > 0\right)$, according to a similar analysis as Mulayoff et al. (2021), we have

$$\lambda_{\max}\left(\frac{1}{n}\Phi D\Phi^T\right) = \frac{1}{n}\max_{u\in S^{n-1}} \left\|\Phi D^{\frac{1}{2}} u\right\|_2^2$$

$$\geq \frac{n_\gamma \Gamma(\gamma)}{n} + \frac{\alpha^2}{n}\cdot \sum_{i=1}^k \left[ \left(\sum_{x_j\in\mathcal{A}_\gamma} x_j I_{j,i} w_i^{(2)}\right)^2 + \left(\sum_{x_j\in\mathcal{A}_\gamma} I_{j,i} w_i^{(2)}\right)^2 + \left(\sum_{x_j\in\mathcal{A}_\gamma} \phi\left(w_i^{(1)} x_j + b_i^{(1)}\right)\right)^2 \right]$$

$$\geq \frac{n_\gamma \Gamma(\gamma)}{n} + \frac{2\Gamma(\gamma)}{n\cdot n_\gamma}\cdot \sum_{i=1}^k \left|w_i^{(2)}\right| \sqrt{\left(\sum_{x_j\in\mathcal{A}_\gamma} x_j I_{j,i}\right)^2 + \left(\sum_{x_j\in\mathcal{A}_\gamma} I_{j,i}\right)^2} \cdot \left|\sum_{x_j\in\mathcal{A}_\gamma} \phi\left(w_i^{(1)} x_j + b_i^{(1)}\right)\right|, \tag{36}$$

where the first inequality holds from the choice of $u$ equation 35. The second inequality holds because of the fact $\alpha \geq \sqrt{\frac{\Gamma(\gamma)}{n_\gamma}}$ and AM-GM inequality.

**Remark D.1.** The case in Mulayoff et al. (2021) (and also Qiao et al. (2024)) can be considered as a special case of the analysis above. Plugging in $D_{j,j} = 1$, $\Gamma(\gamma) = 1$, $\mathcal{A}_\gamma = \{x_i, \; i \in [n]\}$ and $n_\gamma = n$, our analysis will recover the analysis in Mulayoff et al. (2021).

For $i \in [k]$, define $\mathcal{C}_i = \left\{ x_j \in \mathcal{A}_\gamma \mid I_{j,i} = 1 \right\}$, the data points in $\mathcal{A}_\gamma$ such that the $i$-th neuron is active. Let $n_i = \sum_{x_j \in \mathcal{A}_\gamma} I_{j,i} = |\mathcal{C}_i|$ for $i \in [k]$. According to a reformulation, we have

$$\frac{2\Gamma(\gamma)}{n \cdot n_\gamma} \cdot \sum_{i=1}^k \left| w_i^{(2)} \right| \sqrt{\left( \sum_{x_j \in \mathcal{A}_\gamma} x_j I_{j,i} \right)^2 + \left( \sum_{x_j \in \mathcal{A}_\gamma} I_{j,i} \right)^2} \cdot \left| \sum_{x_j \in \mathcal{A}_\gamma} \phi \left( w_i^{(1)} x_j + b_i^{(1)} \right) \right|$$

$$= 2\Gamma(\gamma) \cdot \frac{n_\gamma}{n} \cdot \sum_{i=1}^k \left| w_i^{(2)} \right| \cdot \mathbb{P}^2 \left( X \in \mathcal{C}_i \mid X \in \mathcal{A}_\gamma \right) \cdot \sqrt{1 + \left( \mathbb{E} \left[ X \mid X \in \mathcal{C}_i \right] \right)^2} \cdot \mathbb{E} \left[ w_i^{(1)} X + b_i^{(1)} \mid X \in \mathcal{C}_i \right], \tag{37}$$

where $X$ is a random sample from the dataset under the uniform distribution. We define the threshold of the $i$-th neuron as: for $i \in [k]$,

$$\tau_i = \begin{cases} -\frac{b_i^{(1)}}{w_i^{(1)}}, & w_i^{(1)} \neq 0, \\ 0, & w_i^{(1)} = 0. \end{cases} \tag{38}$$

Then it holds that

$$2\Gamma(\gamma) \cdot \frac{n_\gamma}{n} \cdot \sum_{i=1}^k \left| w_i^{(2)} \right| \cdot \mathbb{P}^2 \left( X \in \mathcal{C}_i \mid X \in \mathcal{A}_\gamma \right) \cdot \sqrt{1 + \left( \mathbb{E} \left[ X \mid X \in \mathcal{C}_i \right] \right)^2} \cdot \mathbb{E} \left[ w_i^{(1)} X + b_i^{(1)} \mid X \in \mathcal{C}_i \right]$$

$$\geq 2\Gamma(\gamma) \cdot \frac{n_\gamma}{n} \cdot \sum_{i=1}^k \left| w_i^{(2)} w_i^{(1)} \right| \cdot \mathbb{P}^2 \left( X \in \mathcal{C}_i \mid X \in \mathcal{A}_\gamma \right) \cdot \sqrt{1 + \left( \mathbb{E} \left[ X \mid X \in \mathcal{C}_i \right] \right)^2} \cdot \left| \mathbb{E} \left[ X - \tau_i \mid X \in \mathcal{C}_i \right] \right|, \tag{39}$$

where the inequality is because for some neurons, $w_i^{(1)}$ may be 0, while $\mathbb{E} \left[ w_i^{(1)} X + b_i^{(1)} \mid X \in \mathcal{C}_i \right]$ can still be positive. Note that when $w_i^{(1)} \neq 0$, $\mathcal{C}_i$ can be either $\{ X > \tau_i, \ X \in \mathcal{A}_\gamma \}$ or $\{ X < \tau_i, \ X \in \mathcal{A}_\gamma \}$, and we choose the minimum among these two to lower bound the R.H.S of equation 39. More specifically, we define the following two functions with respect to the two choices of $\mathcal{C}_i$:

$$h_\gamma^+(x) = \mathbb{P}^2 \left( X > x \mid X \in \mathcal{A}_\gamma \right) \cdot \sqrt{1 + \left( \mathbb{E} \left[ X \mid X \in \mathcal{A}_\gamma, \ X > x \right] \right)^2} \cdot \mathbb{E} \left[ X - x \mid X \in \mathcal{A}_\gamma, \ X > x \right], \tag{40}$$

$$h_\gamma^-(x) = \mathbb{P}^2 \left( X < x \mid X \in \mathcal{A}_\gamma \right) \cdot \sqrt{1 + \left( \mathbb{E} \left[ X \mid X \in \mathcal{A}_\gamma, \ X < x \right] \right)^2} \cdot \mathbb{E} \left[ x - X \mid X \in \mathcal{A}_\gamma, \ X < x \right], \tag{41}$$

where for both functions, $X$ is a random sample from the dataset under the uniform distribution. Moreover, let $h_\gamma(x) = \min \left\{ h_\gamma^+(x), h_\gamma^-(x) \right\}$, we have

$$\mathbb{P}^2 \left( X \in \mathcal{C}_i \mid X \in \mathcal{A}_\gamma \right) \cdot \sqrt{1 + \left( \mathbb{E} \left[ X \mid X \in \mathcal{C}_i \right] \right)^2} \cdot \left| \mathbb{E} \left[ X - \tau_i \mid X \in \mathcal{C}_i \right] \right| \geq h_\gamma(\tau_i). \tag{42}$$

Plugging this into equation 39, it holds that

$$2\Gamma(\gamma) \cdot \frac{n_\gamma}{n} \cdot \sum_{i=1}^k \left| w_i^{(2)} \right| \cdot \mathbb{P}^2 \left( X \in \mathcal{C}_i \mid X \in \mathcal{A}_\gamma \right) \cdot \sqrt{1 + \left( \mathbb{E} \left[ X \mid X \in \mathcal{C}_i \right] \right)^2} \cdot \mathbb{E} \left[ w_i^{(1)} X + b_i^{(1)} \mid X \in \mathcal{C}_i \right]$$

$$\geq 2\Gamma(\gamma) \cdot \frac{n_\gamma}{n} \cdot \sum_{i=1}^k \left| w_i^{(1)} w_i^{(2)} \right| \cdot h_\gamma(\tau_i) \geq 2\Gamma(\gamma) \cdot \frac{n_\gamma}{n} \cdot \int_{-x_{\max}}^{x_{\max}} |f''(x)| \, h_\gamma(x) dx, \tag{43}$$

where the last inequality holds since the sum of absolute values is not smaller than the absolute value of the sum. As a result, we can lower bound (i) as:

$$(i) = \lambda_{\max} \left( \frac{1}{n} \Phi D \Phi^T \right) \geq \frac{n_\gamma \Gamma(\gamma)}{n} + 2\Gamma(\gamma) \cdot \frac{n_\gamma}{n} \cdot \int_{-x_{\max}}^{x_{\max}} |f''(x)| \, h_\gamma(x) dx. \tag{44}$$

### D.4 Putting Everything Together

Combining equation 31, equation 32 and equation 44, we have for any choice of $\gamma > 0$,

$$\frac{n_\gamma \Gamma(\gamma)}{n} \left( 1 + 2 \int_{-x_{\max}}^{x_{\max}} |f''(x)| \, h_\gamma(x) dx \right) \leq \lambda_{\max} \left( \nabla_\theta^2 \mathcal{L}(\theta) \right) + 2 \max\{x_{\max}, 1\} \cdot \mathcal{L}(f). \quad (45)$$

Or equivalently,

$$1 + 2 \int_{-x_{\max}}^{x_{\max}} |f''(x)| \, h_\gamma(x) dx \leq \frac{n}{n_\gamma \Gamma(\gamma)} \left( \lambda_{\max} \left( \nabla_\theta^2 \mathcal{L}(\theta) \right) + 2 \max\{x_{\max}, 1\} \cdot \mathcal{L}(f) \right), \quad (46)$$

which finishes the proof of the first part.

If $f = f_\theta$ is a stable solution of GD with learning rate $\eta$, according to the definition of $\mathcal{F}(\eta, \mathcal{D})$ equation 2, we can replace the $\lambda_{\max}(\nabla^2 \mathcal{L}(\theta))$ by $\frac{2}{\eta}$.

### D.5 Some discussions about the parameters

For reference, the notations are listed below.

1. $\mathcal{A}_\gamma = \left\{ x_i \in \mathcal{D} \;\middle|\; |f_\theta(x_i)| \leq \gamma \right\}$, $n_\gamma = |\mathcal{A}_\gamma|$.

2. $\Gamma(\gamma) = \frac{e^{-\gamma}}{(1+e^{-\gamma})^2}$.

3. $\eta$ is the learning rate of GD, $\mathcal{L}(f)$ is the empirical logistic loss.

4. $h_\gamma(x) = \min \left\{ h_\gamma^+(x), h_\gamma^-(x) \right\}$, defined in equation 40 and equation 41.

**Some discussions.** For the result to be meaningful, it is required that $h_\gamma(x)$ is nondegenerate and the R.H.S of equation 46 is not very large. Therefore, the result will be meaningful under the following conditions:

1. $\gamma > 0$ is a constant that is not very large, such that $\Gamma(\gamma)$ is not very small.

2. For the chosen $\gamma$, the function $f_\theta$ and the dataset satisfy that $\frac{n_\gamma}{n}$ is not very small, *i.e.* for at least a constant portion of the dataset, the prediction of $f_\theta$ is not very certain.

3. A constant portion of the data points in $\mathcal{A}_\gamma$ is close to the boundary of $[-x_{\max}, x_{\max}]$. In this way, for $x$ in the interior of $\mathcal{A}_\gamma$, the value of $h_\gamma(x)$ can be lower bounded.

**Trade-off between the parameters.** First, as the threshold $\gamma$ increases, the "uncertain set" $\mathcal{A}_\gamma$ will also become larger. As a result, the term $\frac{n}{n_\gamma}$ on the R.H.S will become smaller, while the interior of $\mathcal{A}_\gamma$ where the TV constraint is strong will also be larger. Both effects tend to make the result more significant. However, the term $\frac{1}{\Gamma(\gamma)} \approx e^\gamma$ on the R.H.S will grow exponentially, and thus balance out the aforementioned effects. Therefore, there is a trade-off in the choice of $\gamma$. Meanwhile, since we do not have any guarantee on the scale of $f_\theta$, the portion or even the existence of the uncertain set $\mathcal{A}_\gamma$ for a fair $\gamma$ is not ensured.

## E Existence of Arbitrarily Flat yet Interpolating Solutions

Theorem 3.1 only provides smoothness guarantee between the left-most and right-most "uncertain sets" determined by the output function $f_\theta$, while the existence of such uncertain set is not guaranteed. In this part, we construct an example to show that there exist arbitrarily flat yet interpolating solutions. For "interpolation" here, we mean that the logistic loss is arbitrarily small, and it is clear that the training loss cannot be exactly 0.

**Example.** Let $x_{\max} = 1$, then the interval becomes $[-1, 1]$. The design $\{x_i\}_{i=1}^n$ is chosen to be $n$ equally spaced points in $[-1, 1]$. For any label set $\{y_i\}_{i=1}^n \in \{-1, 1\}^n$, let $\gamma_{\max} > 0$ be some arbitrarily large constant, below we consider the two-layer ReLU NN $f$ where $f(x_i) y_i = \gamma_{\max}$ and $f$ is (almost) linear between any two neighboring design points. We will show that there exists some function $f = f_\theta$ satisfying the conditions above while $\lambda_{\max}(\nabla_\theta^2 \mathcal{L}(\theta))$ is small. By saying "almost linear", we mean that we need to perturb the knots from the design points by a small $\epsilon$ to satisfy the twice differentiable assumption.

**Theorem E.1.** *[Restate Theorem 3.2] For the example above, there exists choice of $\theta$ such that $f_\theta(x_i) = y_i\gamma_{\max}$ for all $i \in [n]$, $\mathcal{L}(\theta)$ is twice differentiable w.r.t. $\theta$ and $\lambda_{\max}\left(\nabla_\theta^2\mathcal{L}(\theta)\right) \leq O\left((n^2\gamma_{\max} + 1)e^{-\gamma_{\max}}\right).$*

*Proof of Theorem E.1.* Recall that if $f = f_\theta$ is twice differentiable with respect to $\theta$ (*i.e.* the knots of $f$ do not coincide with design points), we have

$$\nabla_\theta^2\mathcal{L}(\theta) = \frac{1}{n}\sum_{i=1}^n \frac{-y_ie^{-y_if(x_i)}}{1 + e^{-y_if(x_i)}}\nabla_\theta^2 f(x_i) + \frac{1}{n}\sum_{i=1}^n \frac{e^{-y_if(x_i)}}{\left(1 + e^{-y_if(x_i)}\right)^2}\nabla_\theta f(x_i)\nabla_\theta f(x_i)^T, \quad (47)$$

which further implies that

$$\lambda_{\max}\left(\nabla_\theta^2\mathcal{L}(\theta)\right) \leq \lambda_{\max}\left(\frac{1}{n}\sum_{i=1}^n \frac{-y_ie^{-y_if(x_i)}}{1 + e^{-y_if(x_i)}}\nabla_\theta^2 f(x_i)\right) + \lambda_{\max}\left(\frac{1}{n}\sum_{i=1}^n \frac{e^{-y_if(x_i)}}{\left(1 + e^{-y_if(x_i)}\right)^2}\nabla_\theta f(x_i)\nabla_\theta f(x_i)^T\right)$$

$$\leq 2e^{-\gamma_{\max}} + \lambda_{\max}\left(\frac{1}{n}\sum_{i=1}^n \frac{e^{-y_if(x_i)}}{\left(1 + e^{-y_if(x_i)}\right)^2}\nabla_\theta f(x_i)\nabla_\theta f(x_i)^T\right)$$

$$\leq 2e^{-\gamma_{\max}} + e^{-\gamma_{\max}} \cdot \max_i \lambda_{\max}\left(\nabla_\theta f(x_i)\nabla_\theta f(x_i)^T\right)$$

$$= 2e^{-\gamma_{\max}} + e^{-\gamma_{\max}} \cdot \max_i \|\nabla_\theta f(x_i)\|_2^2,$$

$$(48)$$

where the first and last inequalities hold because all the matrices here are symmetric. The second inequality results from Lemma C.1 and our choice $x_{\max} = 1$.

Let $f = f_\theta$ be a two-layer ReLU NN with $n$ active neurons such that $f_\theta$ interpolates $\{(x_i, y_i\gamma_{\max})\}_{i=1}^n$ while the knots are arbitrarily close to $\{x_i\}_{i=1}^n$. For inactive neurons (indexed by $j$), we simply let $w_j^{(1)} = w_j^{(2)} = b_j^{(1)} = 0$, then the gradient w.r.t. such inactive neuron is $0$. For active neurons (indexed by $i$), we manually choose $w_i^{(2)} = \sqrt{n\gamma_{\max}}$ or $-\sqrt{n\gamma_{\max}}$, which implies that $|w_i^{(1)}| \leq O(\sqrt{n\gamma_{\max}})$ (the activation threshold is inside $[-1, 1]$). Then according to the calculation of $\nabla_\theta f_\theta(x)$ in Appendix C.1, we have for any data point $x_j$,

$$\begin{cases} \left|\nabla_{w_i^{(1)}} f_\theta(x_j)\right| = \left|x_j w_i^{(2)}\mathbb{1}\left(w_i^{(1)}x_j + b_i^{(1)} > 0\right)\right| \leq \sqrt{n\gamma_{\max}} \\ \left|\nabla_{b_i^{(1)}} f_\theta(x_j)\right| = \left|w_i^{(2)}\mathbb{1}\left(w_i^{(1)}x_j + b_i^{(1)} > 0\right)\right| \leq \sqrt{n\gamma_{\max}} \\ \left|\nabla_{w_i^{(2)}} f_\theta(x_j)\right| = \left|\phi\left(w_i^{(1)}x_j + b_i^{(1)}\right)\right| \leq 2|w_i^{(1)}| \leq O(\sqrt{n\gamma_{\max}}). \end{cases} \quad (49)$$

In this way, we have $\max_i \|\nabla_\theta f(x_i)\|_2^2 \leq 1 + 3n \cdot O(n\gamma_{\max}) = O(n^2\gamma_{\max} + 1)$. Plugging this into equation 48, we finally have

$$\lambda_{\max}\left(\nabla_\theta^2\mathcal{L}(\theta)\right) \leq O\left((n^2\gamma_{\max} + 1)e^{-\gamma_{\max}}\right), \quad (50)$$

which finishes the proof. □

**Remark E.2.** Since $\lim_{\gamma\to\infty}\gamma e^{-\gamma} = 0$, Theorem 3.2 shows that as the learned function tends to interpolate (*i.e.* the training loss gets close to 0), the landscape is actually becoming flat. In other words, the solution that predicts $\infty$ for $y = 1$ and $-\infty$ for $y = -1$ is an arbitrarily flat global minimum with $0$-training loss. This is an interesting separation between logistic loss and nonparametric regression with squared loss, where all interpolating solutions are sharp. Unfortunately, our Theorem 3.1 and the constraint equation 5 could not exclude such "interpolating" solutions in logistic regression. This is because we consider the **Below** Edge of Stability regime which only requires an upper bound of the sharpness.

**Remark E.3.** In our experiments (Figure 2), even for very small learning rates (*e.g.* $\eta = 0.01$), the learned function does not actually converge to infinity. We remark that this is because with finite learning rate, the optimization enters the "edge of stability" regime before it could get to the flat global minima at infinity (see Figure 12). Therefore, the learned function would stabilize around some finite function instead of converging to infinity. Motivated by such phenomenon, in the following sections (Theorem 3.5), we assume that the optimization effectively minimizes the excess risk. In this way, we show the closeness of $f_\theta$ and $f_0$, thus excluding the "interpolating solutions".

**Remark E.4.** We point out that such flat interpolating solutions do not lead to any contradictions with our Theorem 3.1. In the example above, although $\lambda_{\max}(\nabla^2 \mathcal{L}(\theta)) + 2x_{\max}\mathcal{L}(f)$ scales as $n^2\gamma_{\max}e^{-\gamma_{\max}}$, the threshold $\gamma$ must be at least $\gamma_{\max}$ for a nondegenerate $\mathcal{A}_\gamma$ and $h_\gamma$. Therefore, the $e^{-\gamma_{\max}}$ is canceled out by $\Gamma(\gamma)$ and the R.H.S of equation 5 scales as $n^2\gamma_{\max}$. In comparison, if ignoring the constant weight function $h_\gamma$, the L.H.S of equation 5 can be the same order $n^2\gamma_{\max}$ if the ground-truth function is random enough. Therefore, the example shows the tightness of our Theorem 3.1 and the necessity of the logarithmic dependence on $\gamma$ in $\Gamma(\gamma)$.

# F PROOF FOR RESULTS IN SECTION 3.3

## F.1 BOUNDING THE GENERALIZATION GAP BY THE SCALE OF PARAMETERS

We begin with the following lemma, which shows that given the $\ell_2$ norm of the parameter, both the TV(1) norm and the value at $x = 0$ of the function can be upper bounded.

**Lemma F.1.** *For* $f(x) = f_\theta(x) = \sum_{i=1}^k w_i^{(2)}\phi\left(w_i^{(1)}x + b_i^{(1)}\right) + b^{(2)}$ *and* $\theta = \left[w_{1:k}^{(1)}, b_{1:k}^{(1)}, w_{1:k}^{(2)}, b^{(2)}\right]$, *we have*

$$\int_{-x_{\max}}^{x_{\max}} |f''(x)|\,dx \leq \frac{\|\theta\|_2^2}{2}. \tag{51}$$

*Meanwhile, we have the following upper bounds on* $|f(0)|$, $|f'(0)|$:

$$|f(0)| \leq \frac{\|\theta\|_2^2}{2} + \|\theta\|_2, \quad |f'(0)| \leq \frac{\|\theta\|_2^2}{2}. \tag{52}$$

*Furthermore, if* $\|\theta\|_2 \geq 1$, *we have* $|f(0)| \leq 2\|\theta\|_2^2$.

*Proof of Lemma F.1.* For the first part, due to the same analysis as equation 43, it holds that

$$\int_{-x_{\max}}^{x_{\max}} |f''(x)|\,dx \leq \sum_{i=1}^k \left|w_i^{(1)}w_i^{(2)}\right| \leq \sqrt{\left(\sum_{i=1}^k \left(w_i^{(1)}\right)^2\right)\cdot\left(\sum_{i=1}^k \left(w_i^{(2)}\right)^2\right)} \leq \frac{\|\theta\|_2^2}{2}, \tag{53}$$

where the second inequality holds because of Cauchy-Schwarz inequality. The last inequality results from the fact that $x(a - x) \leq \frac{a^2}{4}$.

For the second conclusion, we have

$$|f(0)| = \left|\sum_{i=1}^k w_i^{(2)}\phi\left(b_i^{(1)}\right) + b^{(2)}\right| \leq \sum_{i=1}^k \left|w_i^{(2)}b_i^{(1)}\right| + \left|b^{(2)}\right| \leq \frac{\|\theta\|_2^2}{2} + \|\theta\|_2, \tag{54}$$

$$|f'(0)| \leq \sum_{i=1}^k \left|w_i^{(2)}w_i^{(1)}\right| \leq \frac{\|\theta\|_2^2}{2}, \tag{55}$$

where the last step in both inequalities result from the same analysis as equation 53. The proof is complete. $\square$

Define the set $\mathbb{T} = \left\{f : [-x_{\max}, x_{\max}] \to \mathbb{R} \,\middle|\, |f(0)| \leq 4C_2,\ |f'(0)| \leq C_2,\ \int_{-x_{\max}}^{x_{\max}} |f''(x)|dx \leq C_2\right\}$, where $C_2 > 0$ is any constant. Then according to Lemma F.1, the function $f = f_\theta$ belongs to $\mathbb{T}$ with $C_2 = \frac{\|\theta\|_2^2}{2}$ (if $\|\theta\|_2 \geq 1$). Therefore, we would like to analyze the complexity of $\mathbb{T}$. We first bound the metric entropy of the following subset of $\mathbb{T}$ as an intermediate result.

**Lemma F.2.** *Assume the set* $\mathbb{T}_2 = \left\{f : [-x_{\max}, x_{\max}] \to \mathbb{R} \,\middle|\, f(0) = f'(0) = 0,\ \int_{-x_{\max}}^{x_{\max}} |f''(x)|dx \leq C_2\right\}$

*for some constant* $C_2 > 0$, *and the metric is* $\ell_\infty$ *distance* $\|\cdot\|_\infty$, *then there exists a universal constant* $C_1 > 0$ *such that for any* $\epsilon > 0$, *the metric entropy of* $(\mathbb{T}_2, \|\cdot\|_\infty)$ *satisfies*

$$\log N(\epsilon, \mathbb{T}_2, \|\cdot\|_\infty) \leq C_1\sqrt{\frac{C_2 x_{\max}}{\epsilon}}, \tag{56}$$

*where* $C_1$ *can be chosen as the same* $C_1$ *in Lemma B.7.*

*Proof of Lemma F.2.* Let the set $\mathbb{T}_1 = \left\{ f : [-1,1] \to \mathbb{R} \ \middle| \ \int_{-1}^1 |f''(x)| dx \le 1, \ |f(x)| \le 1 \right\}$ (as in Lemma B.7). For a fixed $\epsilon > 0$, according to Lemma B.7, there exists a $\frac{\epsilon}{C_2 x_{\max}}$-covering set of $\mathbb{T}_1$ with respect to $\| \cdot \|_\infty$, denoted as $\{h_i(x)\}_{i \in [N]}$, whose cardinality $N$ satisfies

$$\log N \le C_1 \sqrt{\frac{C_2 x_{\max}}{\epsilon}}. \tag{57}$$

We define $g_i(x) = C_2 x_{\max} h_i(\frac{x}{x_{\max}})$ for all $i \in [N]$. Then $g_i$'s are all defined on $[-x_{\max}, x_{\max}]$. Obviously, we have $\{g_i(x)\}_{i \in [N]}$ also has cardinality $N$.

For any $f(x) \in \mathbb{T}_2$, we define $g(x) = \frac{1}{C_2 x_{\max}} f(x \cdot x_{\max})$ which is defined on $[-1, 1]$. We now show that $g(x) \in \mathbb{T}_1$. First of all, for any $x \in [-x_{\max}, x_{\max}]$, we have $|f'(x)| \le \int_{-x_{\max}}^{x_{\max}} |f''(x)| dx \le C_2$. Therefore, for any $x \in [-x_{\max}, x_{\max}]$, $|f(x)| \le C_2 x_{\max}$, which implies that $|g(x)| \le 1$ for any $x \in [-1, 1]$. Meanwhile, it holds that

$$\begin{aligned}
\int_{-1}^1 |g''(x)| dx &= \int_{-1}^1 \frac{1}{C_2 x_{\max}} \cdot x_{\max}^2 |f''(x \cdot x_{\max})| dx \\
&\le \frac{1}{C_2} \int_{-x_{\max}}^{x_{\max}} |f''(x)| dx \le 1.
\end{aligned} \tag{58}$$

Combining the two results, we have $g \in \mathbb{T}_1$. Therefore, there exists some $h_i$ such that $\|g - h_i\|_\infty \le \frac{\epsilon}{C_2 x_{\max}}$. Since $f(x) = C_2 x_{\max} g(\frac{x}{x_{\max}})$, $\|g_i - f\|_\infty = C_2 x_{\max} \|h_i - g\|_\infty \le \epsilon$.

In conclusion, $\{g_i\}_{i \in [N]}$ is an $\epsilon$-covering of $\mathbb{T}_2$ with respect to $\| \cdot \|_\infty$. Moreover, the cardinality of $\{g_i\}_{i \in [N]}$ is $N$, which finishes the proof. □

With Lemma F.2, we are ready to bound the metric entropy of $\mathbb{T}$.

**Lemma F.3.** *Assume the metric is $\ell_\infty$ distance $\| \cdot \|_\infty$, then the metric entropy of $(\mathbb{T}, \| \cdot \|_\infty)$ satisfies*

$$\log N(\epsilon, \mathbb{T}, \| \cdot \|_\infty) \le O\left( \sqrt{\frac{C_2 x_{\max}}{\epsilon}} \right), \tag{59}$$

*where the regime is $C_2 \ge 1$, $x_{\max} \ge 1$, $\epsilon \le 1$.*

*Proof of Lemma F.3.* For any function $f \in \mathbb{T}$, it can be written as below:

$$f(x) = f(0) + f'(0)x + g(x), \tag{60}$$

where $g(x) = f(x) - f(0) - f'(0)x$ satisfies that $g(0) = g'(0) = 0$ and $g''(x) = f''(x)$. Therefore, to cover $\mathbb{T}$ to $\epsilon$ accuracy, it suffices to cover the three parts to $\frac{\epsilon}{3}$ accuracy with respect to $\| \cdot \|_\infty$, respectively.

For $f(0)$, since $|f(0)| \le 4C_2$, the covering number is bounded by

$$N_1 \le \frac{8C_2}{\epsilon/3} = \frac{24C_2}{\epsilon}. \tag{61}$$

For $f'(0)x$, since $|f'(0)| \le C_2$, the covering number is bounded by

$$N_2 \le \frac{6C_2 x_{\max}}{\epsilon}. \tag{62}$$

Finally, for $g(x)$, since $g(x) \in \mathbb{T}_2$ with constant $C_2$, the covering number is bounded according to Lemma F.2 above:

$$\log N_3 \le C_1 \sqrt{\frac{3C_2 x_{\max}}{\epsilon}}. \tag{63}$$

Combining the three parts, the metric entropy is bounded by

$$\log N(\epsilon, \mathbb{T}, \| \cdot \|_\infty) \le \log N_1 + \log N_2 + \log N_3 \le O\left( \sqrt{\frac{C_2 x_{\max}}{\epsilon}} \right), \tag{64}$$

where the last inequality holds since we assume $\frac{C_2 x_{\max}}{\epsilon} \ge 1$. □

W.l.o.g, we also assume that the learned function $f = f_\theta$ satisfies that $\|f_\theta\|_\infty \le B$ for some constant $B > 0$. Therefore, the function set to consider is $\widetilde{\mathbb{T}} = \mathbb{T} \cap \{f : [-x_{\max}, x_{\max}] \to \mathbb{R} \mid \|f\|_\infty \le B\}$. Now we prove a generalization gap upper bound over the function set $\widetilde{\mathbb{T}}$. We consider a general class of loss functions, which includes logistic loss as a special case. Note that we still consider the fixed design setting where $\{x_i\}_{i=1}^n$ are fixed and $x_i \in [-x_{\max}, x_{\max}]$. Each $y_i$ is sampled independently from some distribution conditional on $x_i$. Then the empirical loss $\mathcal{L}(f) = \frac{1}{n} \sum_{i=1}^n \ell(f, (x_i, y_i))$, while the populational loss $\bar{\mathcal{L}}(f) = \frac{1}{n} \sum_{i=1}^n \mathbb{E}_{y_i'|x_i} \ell(f, (x_i, y_i'))$. The following lemma shows that if the loss function is Lipschitz continuous (w.r.t $f$) and bounded, then we can derive a high-probability upper bound of $\left|\mathcal{L}(f) - \bar{\mathcal{L}}(f)\right|$.

**Lemma F.4.** *Assume* $\widetilde{\mathbb{T}} = \left\{f : [-x_{\max}, x_{\max}] \to \mathbb{R} \mid |f(0)| \le 4C_2, |f'(0)| \le C_2, \int_{-x_{\max}}^{x_{\max}} |f''(x)| dx \le C_2\right\}$ $\cap \{f : [-x_{\max}, x_{\max}] \to \mathbb{R} \mid \|f\|_\infty \le B\}$, *where* $B > 0$, $C_2 \ge 1$. *If for constant* $L_1 > 0$ *and monotonically increasing function* $L_2(\cdot) > 0$, $\left|\ell(f, (x, y)) - \ell(\widetilde{f}, (x, y))\right| \le L_1 \left|f(x) - \widetilde{f}(x)\right|$ *and* $|\ell(f, (x, y))| \le L_2(|f(x)|)$ *hold for any possible data point* $(x, y)$ *and any possible function pairs* $(f, \widetilde{f})$, *then with probability* $1 - \delta$, *for any* $f \in \widetilde{\mathbb{T}}$,

$$\left|\mathcal{L}(f) - \bar{\mathcal{L}}(f)\right| \le O\left(\left[\frac{L_1 L_2(B)^4 C_2 x_{\max} \log(1/\delta)^2}{n^2}\right]^{\frac{1}{5}}\right). \tag{65}$$

*Proof of Lemma F.4.* Since $\widetilde{\mathbb{T}}$ is a subset of $\mathbb{T}$, the metric entropy of $\widetilde{\mathbb{T}}$ is smaller than the metric entropy of $\mathbb{T}$. Therefore, for a fixed $\epsilon > 0$, according to Lemma F.3, there exists an $\epsilon$-covering set of $\widetilde{\mathbb{T}}$ (with respect to $\|\cdot\|_\infty$) whose cardinality $N$ satisfies that

$$\log N \le O\left(\sqrt{\frac{C_2 x_{\max}}{\epsilon}}\right). \tag{66}$$

For a fixed function $\bar{f}$ in the covering set, according to our assumption on $\ell$, for any possible $(x, y)$,
$$\left|\ell(\bar{f}, (x, y))\right| \le L_2(|\bar{f}(x)|) \le L_2(B).$$

Meanwhile, note that the features $y_i$'s are sampled independently from the data distribution. According to Hoeffding's inequality, with probability $1 - \delta$, it holds that

$$\left|\mathcal{L}(\bar{f}) - \bar{\mathcal{L}}(\bar{f})\right| \le L_2(B) \cdot \sqrt{\frac{2\log(2/\delta)}{n}}. \tag{67}$$

Together with a union bound over the covering set, we have with probability $1 - \delta$, for all $\bar{f}$ in the covering set,

$$\begin{aligned}
\left|\mathcal{L}(\bar{f}) - \bar{\mathcal{L}}(\bar{f})\right| &\le L_2(B) \cdot \sqrt{\frac{2\log(2N/\delta)}{n}} \\
&\le O\left(L_2(B) \cdot \frac{(C_2 x_{\max})^{\frac{1}{4}} \log(1/\delta)^{\frac{1}{2}}}{n^{\frac{1}{2}} \epsilon^{\frac{1}{4}}}\right).
\end{aligned} \tag{68}$$

Under such high probability event, for any $f \in \widetilde{\mathbb{T}}$, let $\bar{f}$ be a function in the covering set such that $\|f - \bar{f}\|_\infty \le \epsilon$. Then it holds that

$$\begin{aligned}
&\left|\mathcal{L}(f) - \bar{\mathcal{L}}(f)\right| \\
&\le \left|\mathcal{L}(\bar{f}) - \bar{\mathcal{L}}(\bar{f})\right| + O(L_1 \epsilon) \\
&\le O(L_1 \epsilon) + O\left(L_2(B) \cdot \frac{(C_2 x_{\max})^{\frac{1}{4}} \log(1/\delta)^{\frac{1}{2}}}{n^{\frac{1}{2}} \epsilon^{\frac{1}{4}}}\right) \\
&\le O\left(\left[\frac{L_1 L_2(B)^4 C_2 x_{\max} \log(1/\delta)^2}{n^2}\right]^{\frac{1}{5}}\right),
\end{aligned} \tag{69}$$

where the first inequality holds because $\left|\ell(f, (x, y)) - \ell(\widetilde{f}, (x, y))\right| \le L_1 \left|f(x) - \widetilde{f}(x)\right|$. The last inequality results from selecting the $\epsilon$ that minimizes the objective. $\qquad\square$

With the result above, we are ready to prove a generalization gap bound based on $\|\theta\|_2$.

**Lemma F.5.** *If for constant $L_1 > 0$ and monotonically increasing function $L_2(\cdot) > 0$, $\left|\ell(f, (x, y)) - \ell(\widetilde{f}, (x, y))\right| \leq L_1 \left|f(x) - \widetilde{f}(x)\right|$ and $|\ell(f, (x, y))| \leq L_2(|f(x)|)$ hold for any possible data point $(x, y)$ and any possible function pairs $(f, \widetilde{f})$, then with probability $1 - \delta$, for any $f = f_\theta$ such that $\|f\|_\infty \leq B$, it holds that*

$$\left|\mathcal{L}(f) - \bar{\mathcal{L}}(f)\right| \leq O\left(\left[\frac{L_1 L_2(B)^4 \cdot \max\{\|\theta\|_2^2, 1\} \cdot x_{\max} \log\left(\frac{2\max\{\|\theta\|_2^2, 1\}}{\delta}\right)^2}{n^2}\right]^{\frac{1}{5}}\right). \quad (70)$$

*Proof of Lemma F.5.* According to Lemma F.4, we have for any $i \in \mathbb{Z}^+$, with probability $1 - \frac{\delta}{2^i}$,

$$\left|\mathcal{L}(f) - \bar{\mathcal{L}}(f)\right| \leq O\left(\left[\frac{L_1 L_2(B)^4 2^{i-1} x_{\max} \log(2^i/\delta)^2}{n^2}\right]^{\frac{1}{5}}\right) \quad (71)$$

holds for all $f \in \left\{f : [-x_{\max}, x_{\max}] \to \mathbb{R} \;\middle|\; |f(0)| \leq 4 \cdot 2^{i-1}, \; |f'(0)| \leq 2^{i-1}, \; \int_{-x_{\max}}^{x_{\max}} |f''(x)|dx \leq 2^{i-1}, \|f\|_\infty \leq B\right\}$. The total failure probability for the events above is $\sum_{i=1}^\infty \frac{\delta}{2^i} = \delta$. Therefore, we consider the high probability event where equation 71 holds for all $i \in \mathbb{Z}^+$, which happens with probability at least $1 - \delta$.

For $f = f_\theta$ such that $\|f\|_\infty \leq B$, if $\|\theta\|_2 \leq \sqrt{2}$, according to Lemma F.1, it holds that

$$f \in \left\{f : [-x_{\max}, x_{\max}] \to \mathbb{R} \;\middle|\; |f(0)| \leq 4, \; |f'(0)| \leq 1, \; \int_{-x_{\max}}^{x_{\max}} |f''(x)|dx \leq 1, \|f\|_\infty \leq B\right\}.$$

According to the high probability event above $(i = 1)$, we further have:

$$\left|\mathcal{L}(f) - \bar{\mathcal{L}}(f)\right| \leq O\left(\left[\frac{L_1 L_2(B)^4 x_{\max} \log(2/\delta)^2}{n^2}\right]^{\frac{1}{5}}\right). \quad (72)$$

If $\|\theta\|_2 \geq \sqrt{2}$, let $i^\star$ be the smallest integer such that $2^{i^\star - 1} \geq \frac{\|\theta\|_2^2}{2}$. Lemma F.1 implies that

$$f \in \left\{f : [-x_{\max}, x_{\max}] \to \mathbb{R} \;\middle|\; |f(0)| \leq 4 \cdot 2^{i^\star - 1}, \; |f'(0)| \leq 2^{i^\star - 1}, \; \int_{-x_{\max}}^{x_{\max}} |f''(x)|dx \leq 2^{i^\star - 1}, \|f\|_\infty \leq B\right\}.$$

According to the high probability event above $(i = i^\star)$, we further have:

$$\left|\mathcal{L}(f) - \bar{\mathcal{L}}(f)\right| \leq O\left(\left[\frac{L_1 L_2(B)^4 2^{i^\star - 1} x_{\max} \log(2^{i^\star}/\delta)^2}{n^2}\right]^{\frac{1}{5}}\right) \leq O\left(\left[\frac{L_1 L_2(B)^4 \|\theta\|_2^2 x_{\max} \log(2\|\theta\|_2^2/\delta)^2}{n^2}\right]^{\frac{1}{5}}\right), \quad (73)$$

where the last inequality holds since $2^{i^\star - 2} \leq \frac{\|\theta\|_2^2}{2}$.

Combining the two cases above, we have with probability $1 - \delta$, for any $f = f_\theta$ such that $\|f\|_\infty \leq B$, it holds that

$$\left|\mathcal{L}(f) - \bar{\mathcal{L}}(f)\right| \leq O\left(\left[\frac{L_1 L_2(B)^4 \cdot \max\{\|\theta\|_2^2, 1\} \cdot x_{\max} \log\left(\frac{2\max\{\|\theta\|_2^2, 1\}}{\delta}\right)^2}{n^2}\right]^{\frac{1}{5}}\right), \quad (74)$$

which finishes the proof. $\qquad\square$

**The case of logistic loss.** The results above consider general loss functions, while the corollary below plugs in the constant $L_1$ and the function $L_2(\cdot)$ under the special case of logistic loss.

**Corollary F.6** (Restate Theorem 3.3)**.** *If the loss is logistic loss $\ell(f, (x, y)) = \log(1 + e^{-yf(x)})$ and data points $(x, y) \in [-x_{\max}, x_{\max}] \times \{-1, 1\}$, then with probability $1 - \delta$, for any $f = f_\theta$ such that $\|f\|_\infty \leq B$, it holds that*

$$\left|\mathcal{L}(f) - \bar{\mathcal{L}}(f)\right| \leq O\left(\left[\frac{\log(1 + e^B)^4 \cdot \max\{\|\theta\|_2^2, 1\} \cdot x_{\max} \log\left(\frac{2\max\{\|\theta\|_2^2, 1\}}{\delta}\right)^2}{n^2}\right]^{\frac{1}{5}}\right). \quad (75)$$

*Proof of Corollary F.6.* It is easy to check that

$$\left|\ell(f, (x, y)) - \ell(\widetilde{f}, (x, y))\right| \leq \left|y\left(f(x) - \widetilde{f}(x)\right)\right| = \left|f(x) - \widetilde{f}(x)\right|,$$

$$|\ell(f, (x, y))| \leq \log\left(1 + e^{|f(x)|}\right).$$

Therefore, plugging $L_1 = 1$, $L_2(t) = \log(1 + e^t)$ into Lemma F.5 finishes the proof. $\qquad\square$

**Extension to the statistical learning setting.** In the analysis above, we consider the fixed design ($\{x_i\}_{i=1}^n$) setting for technical reasons. The results are readily applicable to the statistical learning setting where the data points (not only the features) are i.i.d. samples. Recall that the empirical loss is $\mathcal{L}(f) = \frac{1}{n}\sum_{i=1}^n \ell(f, (x_i, y_i))$. Now we assume the data $(x_i, y_i)$ are i.i.d. samples from some joint distribution $\mathcal{P}$ defined on $[-x_{\max}, x_{\max}] \times \mathbb{R}$. Define the risk $\mathcal{R}(f) := \mathbb{E}_{(x, y) \sim \mathcal{P}}\ell(f, (x, y))$. Then with identical proof as Lemma F.5, we have the following high probability bound on $|\mathcal{L}(f) - \mathcal{R}(f)|$.

**Corollary F.7.** *If for constant $L_1 > 0$ and monotonically increasing function $L_2(\cdot) > 0$, $\left|\ell(f, (x, y)) - \ell(\widetilde{f}, (x, y))\right| \leq L_1\left|f(x) - \widetilde{f}(x)\right|$ and $|\ell(f, (x, y))| \leq L_2(|f(x)|)$ hold for any possible data point $(x, y)$ and any possible function pairs $(f, \widetilde{f})$, then with probability $1 - \delta$, for any $f = f_\theta$ such that $\|f\|_\infty \leq B$, it holds that*

$$|\mathcal{L}(f) - \mathcal{R}(f)| \leq O\left(\left[\frac{L_1 L_2(B)^4 \cdot \max\{\|\theta\|_2^2, 1\} \cdot x_{\max} \log\left(\frac{2\max\{\|\theta\|_2^2, 1\}}{\delta}\right)^2}{n^2}\right]^{\frac{1}{5}}\right). \quad (76)$$

### F.2 A CRUDE EXCESS RISK BOUND

In this part, we would like to derive an upper bound for the excess risk $\text{ExcessRisk}(f) = \bar{\mathcal{L}}(f) - \bar{\mathcal{L}}(f_0)$, where the loss $\ell$ is the logistic loss. To derive a diminishing upper bound, we make two assumptions. First, we assume that $f = f_\theta$ is "optimized".

**Assumption F.8.** The learned function $f = f_\theta$ is "optimized" such that

$$\mathcal{L}(f) \leq \mathcal{L}(f_0). \quad (77)$$

Meanwhile, we assume that the $\ell_2$ norm of $\theta$ grows sub-linearly w.r.t $n$.

**Assumption F.9.** The learned function $f = f_\theta$ satisfies $\|\theta\|_2 \leq t(n)$, where $t(n) > 1$ and

$$\lim_{n \to \infty} \frac{t(n)\log(t(n))}{n} = 0. \quad (78)$$

Note that Assumption F.9 holds whenever there exists some $\alpha > 0$, $t(n) = O(n^{1-\alpha})$. Based on the two assumptions, we derive the following diminishing ER bound.

**Lemma F.10.** *Let the loss be logistic loss $\ell(f, (x, y)) = \log(1 + e^{-yf(x)})$ and data points $(x, y) \in [-x_{\max}, x_{\max}] \times \{-1, 1\}$. For $f = f_\theta$, assume that Assumption F.8 and Assumption F.9 hold, while $\max\{\|f\|_\infty, \|f_0\|_\infty\} \leq B$ for some constant $B > 0$. Then with probability $1 - \delta$,*

$$\bar{\mathcal{L}}(f) - \bar{\mathcal{L}}(f_0) \leq O\left((1 + B)^{\frac{4}{5}} x_{\max}^{1/5}\left(\frac{t(n) \cdot \log\left(\frac{t(n)}{\delta}\right)}{n}\right)^{\frac{2}{5}} + (1 + B) \cdot \sqrt{\frac{\log(1/\delta)}{n}}\right). \quad (79)$$

*The R.H.S will converge to $0$ if $n$ converges to $\infty$.*

*Proof of Lemma F.10.* For the function $f_0$, note that $|\ell(f_0, (x, y))| \leq \log(1 + e^B) \leq B + 1$. Therefore, according to Hoeffding's inequality, with probability at least $1 - \frac{\delta}{2}$,

$$\left| \mathcal{L}(f_0) - \bar{\mathcal{L}}(f_0) \right| \leq (B + 1) \cdot \sqrt{\frac{2 \log(4/\delta)}{n}}. \tag{80}$$

Meanwhile, due to Corollary F.6, with probability $1 - \frac{\delta}{2}$,

$$\left| \mathcal{L}(f) - \bar{\mathcal{L}}(f) \right| \leq O \left( \left[ \frac{\log(1 + e^B)^4 \cdot \max\{\|\theta\|_2^2, 1\} \cdot x_{\max} \log \left( \frac{4 \max\{\|\theta\|_2^2, 1\}}{\delta} \right)^2}{n^2} \right]^{\frac{1}{5}} \right)$$

$$\leq O \left( \left[ \frac{(1 + B)^4 \cdot t(n)^2 x_{\max} \cdot \log \left( \frac{t(n)}{\delta} \right)^2}{n^2} \right]^{\frac{1}{5}} \right) \tag{81}$$

$$= O \left( (1 + B)^{\frac{4}{5}} x_{\max}^{1/5} \left( \frac{t(n) \cdot \log \left( \frac{t(n)}{\delta} \right)}{n} \right)^{\frac{2}{5}} \right),$$

where the second inequality holds due to Assumption F.9.

Combining the results with Assumption F.8, under the two high probability events above,

$$\bar{\mathcal{L}}(f) - \bar{\mathcal{L}}(f_0) \leq \left| \bar{\mathcal{L}}(f) - \mathcal{L}(f) \right| + \mathcal{L}(f) - \mathcal{L}(f_0) + \left| \mathcal{L}(f_0) - \bar{\mathcal{L}}(f_0) \right|$$

$$\leq O \left( (1 + B)^{\frac{4}{5}} x_{\max}^{1/5} \left( \frac{t(n) \cdot \log \left( \frac{t(n)}{\delta} \right)}{n} \right)^{\frac{2}{5}} \right) + 0 + (B + 1) \cdot \sqrt{\frac{2 \log(4/\delta)}{n}} \tag{82}$$

$$= O \left( (1 + B)^{\frac{4}{5}} x_{\max}^{1/5} \left( \frac{t(n) \cdot \log \left( \frac{t(n)}{\delta} \right)}{n} \right)^{\frac{2}{5}} + (1 + B) \cdot \sqrt{\frac{\log(1/\delta)}{n}} \right),$$

where the event holds with probability at least $1 - \delta$.

Finally, according to Assumption F.9, $\lim_{n \to \infty} \frac{t(n) \log(t(n))}{n} = 0$. Therefore, the R.H.S converges to 0 if $n$ converges to $\infty$. The proof is complete. $\qquad\square$

### F.3 A REFINED ANALYSIS AND TIGHTER RESULT

We remark that using $\|\theta\|_2$ in Theorem 3.3 is mainly for the ease of presentation. According to the same proof as Lemma F.1, we have

$$\int_{-x_{\max}}^{x_{\max}} |f''(x)| \, dx \leq \sum_{i : b_i^{(1)}/w_i^{(1)} \in [-x_{\max}, x_{\max}]} \left| w_i^{(1)} w_i^{(2)} \right|, \tag{83}$$

where the i-th term takes the value 0 if $w_i^{(1)} = 0$. Then according to the same analysis in Lemma F.3, the metric entropy of the possible function set can be tighten to

$$\log N(\epsilon) \leq O \left( \sqrt{\frac{x_{\max} \sum_{i : b_i^{(1)}/w_i^{(1)} \in [-x_{\max}, x_{\max}]} \left| w_i^{(1)} w_i^{(2)} \right|}{\epsilon}} \right) + O \left( \log \left( \frac{x_{\max} \|\theta\|_2^2}{\epsilon} \right) \right). \tag{84}$$

Plugging the metric entropy bound above to Lemma F.4 and applying the "doubling trick" to $\sum_{i : b_i^{(1)}/w_i^{(1)} \in [-x_{\max}, x_{\max}]} \left| w_i^{(1)} w_i^{(2)} \right|$ and $\|\theta\|_2$ simultaneously in Lemma F.5, we directly have the following result.

**Lemma F.11** (Refined version of Lemma F.5). *If for constant $L_1 > 0$ and monotonically increasing function $L_2(\cdot) > 0$, $\left|\ell(f,(x,y)) - \ell(\widetilde{f},(x,y))\right| \le L_1 \left|f(x) - \widetilde{f}(x)\right|$ and $|\ell(f,(x,y))| \le L_2(|f(x)|)$ hold for any possible data point $(x,y)$ and any possible function pairs $(f, \widetilde{f})$, then with probability $1 - \delta$, for any $f = f_\theta$ such that $\|f\|_\infty \le B$, it holds that*

$$
\left|\mathcal{L}(f) - \bar{\mathcal{L}}(f)\right| \le \widetilde{O}\left(\left[\frac{L_1 L_2(B)^4 \cdot \max\left\{\sum_{i:b_i^{(1)}/w_i^{(1)} \in [-x_{\max}, x_{\max}]} \left|w_i^{(1)} w_i^{(2)}\right|, 1\right\} \cdot x_{\max}}{n^2}\right]^{\frac{1}{5}}\right),
$$
(85)

*where the $\widetilde{O}(\cdot)$ absorbs $\log(n)$, $\log(x_{\max})$ and $\log(\|\theta\|_2)$.*

Therefore, as a direct corollary, we get a refined version of Corollary F.6 when applied to the special case of logistic loss.

**Corollary F.12** (Refined version of Corollary F.6). *If the loss is logistic loss $\ell(f,(x,y)) = \log(1 + e^{-yf(x)})$ and data points $(x,y) \in [-x_{\max}, x_{\max}] \times \{-1,1\}$, then with probability $1 - \delta$, for any $f = f_\theta$ such that $\|f\|_\infty \le B$, it holds that*

$$
\left|\mathcal{L}(f) - \bar{\mathcal{L}}(f)\right| \le \widetilde{O}\left(\left[\frac{\log(1 + e^B)^4 \cdot \max\left\{\sum_{i:b_i^{(1)}/w_i^{(1)} \in [-x_{\max}, x_{\max}]} \left|w_i^{(1)} w_i^{(2)}\right|, 1\right\} \cdot x_{\max}}{n^2}\right]^{\frac{1}{5}}\right),
$$
(86)

*where the $\widetilde{O}(\cdot)$ absorbs $\log(n)$, $\log(x_{\max})$ and $\log(\|\theta\|_2)$.*

# G   PROOF OF THEOREM 3.5

Recall that in Theorem 3.1, we have for any $\gamma > 0$, the learned function $f = f_\theta$ satisfies

$$
1 + 2\int_{-x_{\max}}^{x_{\max}} |f''(x)| \, h_\gamma(x)dx \le \frac{n}{n_\gamma \Gamma(\gamma)} \left(\lambda_{\max}(\nabla^2 \mathcal{L}(\theta)) + 2x_{\max}\mathcal{L}(f)\right),
$$
(87)

where both the L.H.S ($h_\gamma$) and the R.H.S ($n_\gamma$) depend on the output function $f_\theta$. Therefore, the result is not stable since the output function $f_\theta$ can be arbitrary functions. Below, we will derive a (weighted) TV bound independent of the output function, based on the assumption that the excess risk is diminishing as the number of data points increases.

**Assumption G.1** (Restate Assumption 3.4). We assume that the learned function $f = f_\theta$ satisfies $\bar{\mathcal{L}}(f) - \bar{\mathcal{L}}(f_0) \le \epsilon(n)$, *i.e.*

$$
\mathbb{E}_{x \sim \mathcal{D}}\mathbb{E}_{y \sim \mathcal{B}(x)} \log\left(1 + e^{-yf(x)}\right) \le \mathbb{E}_{x \sim \mathcal{D}}\mathbb{E}_{y \sim \mathcal{B}(x)} \log\left(1 + e^{-yf_0(x)}\right) + \epsilon(n),
$$

for some function $\epsilon(n)$ such that $\lim_{n \to \infty} \epsilon(n) = 0$.

First of all, similar to the "uncertain set" $\mathcal{A}_\gamma$, we define the uncertain set $\bar{\mathcal{A}}_\gamma$ with respect to $f_0$. The set $\bar{\mathcal{A}}_\gamma$ contains the region where the prediction of $f_0$ is not very certain.

**Uncertain set of $f_0$.** For any $\gamma > 0$, we define the uncertain set $\bar{\mathcal{A}}_\gamma = \left\{x \in [-x_{\max}, x_{\max}] \, \middle| \, |f_0(x)| \le \gamma\right\}$, which includes the data points whose labels have more randomness. Below we will show that if $\epsilon(n)$ is small, then $f_\theta$ and $f_0$ will be close to each other. As a result, the two "uncertain sets" $\mathcal{A}_\gamma$ and $\bar{\mathcal{A}}_\gamma$ will also be similar.

**Closeness of two functions.** In the following lemma, we prove that if we set $\zeta > 0$ to be a relatively small constant compared to $\gamma$ (the threshold of the uncertain set $\bar{\mathcal{A}}$), then the following two probabilities $\mathbb{P}_{x \sim \mathcal{D}}\left(|f_0(x)| \le \gamma - \zeta, |f_\theta(x)| > \gamma\right)$ and $\mathbb{P}_{x \sim \mathcal{D}}\left(|f_0(x)| > \gamma + \zeta, |f_\theta(x)| \le \gamma\right)$ are bounded, where $x \sim \mathcal{D}$ means that $x$ is a uniform sample from the dataset $\{x_i\}_{i=1}^n$.

**Lemma G.2.** *Assume that Assumption 3.4 holds for $\epsilon(n)$, then for any $\gamma > \zeta > 0$,*

$$
\mathbb{P}_{x \sim \mathcal{D}}\left(|f_0(x)| \le \gamma - \zeta, |f_\theta(x)| > \gamma\right) + \mathbb{P}_{x \sim \mathcal{D}}\left(|f_0(x)| > \gamma + \zeta, |f_\theta(x)| \le \gamma\right) \le O\left(\epsilon(n) \cdot e^{\gamma+\zeta} \cdot \zeta^{-2}\right).
$$
(88)

*Proof of Lemma G.2.* Recall that $\bar{l}_x(f) := p\log(1 + e^{-f}) + (1-p)\log(1 + e^f)$, where $p = \sigma(f_0(x)) = \frac{e^{f_0(x)}}{1+e^{f_0(x)}}$. Then under the case where $|f_0(x)| \leq \gamma - \zeta$, $|f_\theta(x)| > \gamma$ holds, it holds that

$$
\left|\bar{\ell}_x(f_\theta(x)) - \bar{\ell}_x(f_0(x))\right| \geq \left|\bar{\ell}_x(\gamma) - \bar{\ell}_x(\gamma - \zeta)\right|
$$
$$
\geq \left|\bar{\ell}_x'\left(\gamma - \frac{\zeta}{2}\right)\right| \cdot \frac{\zeta}{2}
$$
$$
\geq \left|\bar{\ell}_x''\left(\gamma - \frac{\zeta}{2}\right)\right| \cdot \left(\frac{\zeta}{2}\right)^2 \tag{89}
$$
$$
\geq \frac{e^\gamma}{(1 + e^\gamma)^2} \cdot \left(\frac{\zeta}{2}\right)^2,
$$

where the first inequality holds because the L.H.S takes the minimal value when $f_0(x) = \gamma - \zeta$, $f_\theta(x) = \gamma$. Note that after the first inequality, the $\bar{\ell}_x$ satisfies $f_0(x) = \gamma - \zeta$. Then the second to the last inequalities result from the monotonicity of $\bar{\ell}_x'$ and $\bar{\ell}_x''$. Detailed calculations of $\bar{\ell}_x'$ and $\bar{\ell}_x''$ can be found in Appendix C.4.

Similarly, if $|f_0(x)| > \gamma + \zeta$, $|f_\theta(x)| \leq \gamma$ holds, we have

$$
\left|\bar{\ell}_x(f_\theta(x)) - \bar{\ell}_x(f_0(x))\right| \geq \left|\bar{\ell}_x(\gamma + \zeta) - \bar{\ell}_x(\gamma)\right|
$$
$$
\geq \left|\bar{\ell}_x'\left(\gamma + \frac{\zeta}{2}\right)\right| \cdot \frac{\zeta}{2}
$$
$$
\geq \left|\bar{\ell}_x''(\gamma + \zeta)\right| \cdot \left(\frac{\zeta}{2}\right)^2 \tag{90}
$$
$$
= \frac{e^{\gamma+\zeta}}{(1 + e^{\gamma+\zeta})^2} \cdot \left(\frac{\zeta}{2}\right)^2,
$$

where the first inequality holds because the L.H.S takes the minimal value when $f_0(x) = \gamma + \zeta$, $f_\theta(x) = \gamma$. Note that after the first inequality, the $\bar{\ell}_x$ satisfies $f_0(x) = \gamma + \zeta$. Then the second to the last inequalities result from the monotonicity of $\bar{\ell}_x'$ and $\bar{\ell}_x''$.

Combining with Assumption 3.4, we have that

$$
\epsilon(n) \geq \mathbb{E}_{x\sim\mathcal{D}}\left|\bar{\ell}_x(f_\theta(x)) - \bar{\ell}_x(f_0(x))\right|
$$
$$
\geq \left(\mathbb{P}_{x\sim\mathcal{D}}\left(|f_0(x)| \leq \gamma - \zeta, |f_\theta(x)| > \gamma\right) + \mathbb{P}_{x\sim\mathcal{D}}\left(|f_0(x)| > \gamma + \zeta, |f_\theta(x)| \leq \gamma\right)\right) \cdot \frac{e^{\gamma+\zeta}}{(1 + e^{\gamma+\zeta})^2} \cdot \left(\frac{\zeta}{2}\right)^2.
$$
$$\tag{91}$$

Reformulating the inequality, we have

$$
\mathbb{P}_{x\sim\mathcal{D}}\left(|f_0(x)| \leq \gamma - \zeta, |f_\theta(x)| > \gamma\right) + \mathbb{P}_{x\sim\mathcal{D}}\left(|f_0(x)| > \gamma + \zeta, |f_\theta(x)| \leq \gamma\right) \leq O\left(\epsilon(n) \cdot e^{\gamma+\zeta} \cdot \zeta^{-2}\right),
$$
$$\tag{92}$$

which finishes the proof. $\square$

To transfer equation 5 to an inequality independent of $f_\theta$, what remains is to upper bound the R.H.S and lower bound the L.H.S. In another word, we need to lower bound the weight function $h_\gamma(x)$ and the cardinality $n_\gamma$. We will first derive a result for any finite $n$ that depends only on $f_0$ and the feature set. When considering the asymptotic performance (*i.e.* $n \to \infty$), we assume that the features $\{x_i\}_{i=1}^n$ are i.i.d. samples from some underlying distribution $\mathcal{P}_x$ supported on $[-x_{\max}, x_{\max}]$. We begin with the lower bound of $n_\gamma$.

**Lower bound of $n_\gamma/n$.** Recall that $n_\gamma$ is the cardinality of $\mathcal{A}_\gamma$ in Theorem 3.1. Below we incorporate Lemma G.2 to replace the term with the number of data points in $\bar{\mathcal{A}}$.

**Lemma G.3.** *For any fixed constants $\gamma > \zeta > 0$, if Assumption 3.4 holds with $\epsilon(n)$, then*

$$
\frac{n_\gamma}{n} = \frac{|\mathcal{A}_\gamma|}{n} \geq \mathbb{P}_{X\sim\mathcal{D}}\left(X \in \bar{\mathcal{A}}_{\gamma-\zeta}\right) - O\left(\epsilon(n) \cdot e^{\gamma+\zeta} \cdot \zeta^{-2}\right). \tag{93}
$$

*Furthermore, when $n$ converges to infinity, the R.H.S will converge in probability:*

$$\mathbb{P}_{X\sim\mathcal{D}}\left(X\in\bar{\mathcal{A}}_{\gamma-\zeta}\right)-O\left(\epsilon(n)\cdot e^{\gamma+\zeta}\cdot\zeta^{-2}\right)\xrightarrow{p}\mathbb{P}_{x\sim\mathcal{P}_x}\left(x\in\bar{\mathcal{A}}_{\gamma-\zeta}\right). \tag{94}$$

*Proof of Lemma G.3.* If Assumption 3.4 holds with $\epsilon(n)$, it holds that

$$
\begin{aligned}
\frac{n_\gamma}{n}=\frac{|\mathcal{A}_\gamma|}{n}&=\frac{\text{number of features in }\mathcal{A}_\gamma}{n}\\
&\geq\frac{\text{number of features in }\mathcal{A}_\gamma\cap\bar{\mathcal{A}}_{\gamma-\zeta}}{n}\\
&=\frac{\text{number of features in }\bar{\mathcal{A}}_{\gamma-\zeta}}{n}-\frac{\text{number of features in }\bar{\mathcal{A}}_{\gamma-\zeta}\cap\mathcal{A}_\gamma^{\complement}}{n}\\
&=\frac{\left|\bar{\mathcal{A}}_{\gamma-\zeta}\cap\mathcal{D}\right|}{n}-\mathbb{P}_{x\sim\mathcal{D}}\left(|f_0(x)|\leq\gamma-\zeta,\ |f_\theta(x)|>\gamma\right)\\
&\geq\frac{\left|\bar{\mathcal{A}}_{\gamma-\zeta}\cap\mathcal{D}\right|}{n}-O\left(\epsilon(n)\cdot e^{\gamma+\zeta}\cdot\zeta^{-2}\right)\\
&=\mathbb{P}_{X\sim\mathcal{D}}\left(X\in\bar{\mathcal{A}}_{\gamma-\zeta}\right)-O\left(\epsilon(n)\cdot e^{\gamma+\zeta}\cdot\zeta^{-2}\right),
\end{aligned}
\tag{95}
$$

where the last inequality holds because of Lemma G.2.

According to Assumption 3.4, $\lim_{n\to\infty}\epsilon(n)=0$. Therefore, $\lim_{n\to\infty}O\left(\epsilon(n)\cdot e^{\gamma+\zeta}\cdot\zeta^{-2}\right)=0$. Meanwhile, note that $\left|\bar{\mathcal{A}}_{\gamma-\zeta}\cap\mathcal{D}\right|$ follows the Binomial distribution Binomial $\left(n,\mathbb{P}_{x\sim\mathcal{P}_x}\left(x\in\bar{\mathcal{A}}_{\gamma-\zeta}\right)\right)$, which implies that $\left|\bar{\mathcal{A}}_{\gamma-\zeta}\cap\mathcal{D}\right|/n$ will converge (in probability) to $\mathbb{P}_{x\sim\mathcal{P}_x}\left(x\in\bar{\mathcal{A}}_{\gamma-\zeta}\right)$ as $n\to\infty$. Combining the results, the proof for the last conclusion is complete. $\square$

**Lower bound of $h_\gamma(x)$.** Recall that $h_\gamma(x)=\min\left\{h_\gamma^+(x),h_\gamma^-(x)\right\}$ and

$$h_\gamma^+(x)=\mathbb{P}^2\left(X>x\mid X\in\mathcal{A}_\gamma\right)\cdot\sqrt{1+\left(\mathbb{E}\left[X\mid X\in\mathcal{A}_\gamma,\ X>x\right]\right)^2}\cdot\mathbb{E}\left[X-x\,\middle|\,X\in\mathcal{A}_\gamma,\ X>x\right],$$

$$h_\gamma^-(x)=\mathbb{P}^2\left(X<x\mid X\in\mathcal{A}_\gamma\right)\cdot\sqrt{1+\left(\mathbb{E}\left[X\mid X\in\mathcal{A}_\gamma,\ X<x\right]\right)^2}\cdot\mathbb{E}\left[x-X\,\middle|\,X\in\mathcal{A}_\gamma,\ X<x\right],$$

where for both functions, $X$ is a random sample from the dataset under the uniform distribution. Similar to the analysis above, we derive a lower bound for $h_\gamma(x)$ which only depends on $f_0$.

**Lemma G.4.** *For any fixed constants $\gamma>\zeta>0$, if Assumption 3.4 holds with $\epsilon(n)$, then we have $h_\gamma(x)=\min\left\{h_\gamma^+(x),h_\gamma^-(x)\right\}\geq\min\left\{\bar{h}_{\gamma,\zeta}^+(x,n),\bar{h}_{\gamma,\zeta}^-(x,n)\right\}=\bar{h}_{\gamma,\zeta}(x,n)$, where*

$$\bar{h}_{\gamma,\zeta}^+(x,n)=\left(\frac{\mathbb{P}_{X\sim\mathcal{D}}(X>x,\ X\in\bar{\mathcal{A}}_{\gamma-\zeta})-p(n)}{\mathbb{P}_{X\sim\mathcal{D}}(X\in\bar{\mathcal{A}}_{\gamma+\zeta})+p(n)}\right)^2\cdot\frac{\mathbb{E}_{X\sim\mathcal{D}}\left[(X-x)\mathbb{1}\left(X>x,\ X\in\bar{\mathcal{A}}_{\gamma-\zeta}^{p(n)}\right)\right]}{\mathbb{P}_{X\sim\mathcal{D}}\left[X>x,\ X\in\bar{\mathcal{A}}_{\gamma+\zeta}\right]+p(n)},$$

$$\bar{h}_{\gamma,\zeta}^-(x,n)=\left(\frac{\mathbb{P}_{X\sim\mathcal{D}}(X<x,\ X\in\bar{\mathcal{A}}_{\gamma-\zeta})-p(n)}{\mathbb{P}_{X\sim\mathcal{D}}(X\in\bar{\mathcal{A}}_{\gamma+\zeta})+p(n)}\right)^2\cdot\frac{\mathbb{E}_{X\sim\mathcal{D}}\left[(x-X)\mathbb{1}\left(X<x,\ X\in\bar{\mathcal{A}}_{\gamma-\zeta}^{p(n)}\right)\right]}{\mathbb{P}_{X\sim\mathcal{D}}\left[X<x,\ X\in\bar{\mathcal{A}}_{\gamma+\zeta}\right]+p(n)}.$$

*For the two functions above, $p(n)$ is the $O\left(\epsilon(n)\cdot e^{\gamma+\zeta}\cdot\zeta^{-2}\right)$ term in Lemma G.2, while*

$$\bar{\mathcal{A}}_\gamma^p:=\left\{x\in\bar{\mathcal{A}}_\gamma\,\middle|\,\min\left(\mathbb{P}_{X\sim\mathcal{D}}(X\in\bar{\mathcal{A}}_\gamma,\ X>x),\ \mathbb{P}_{X\sim\mathcal{D}}(X\in\bar{\mathcal{A}}_\gamma,\ X<x)\right)\geq p\right\}.$$

*Furthermore, when $n$ converges to infinity, $\bar{h}_{\gamma,\zeta}(x,n)$ will converge (in probability) to $\bar{h}_{\gamma,\zeta}(x)=\min\left\{\bar{h}_{\gamma,\zeta}^+(x),\bar{h}_{\gamma,\zeta}^-(x)\right\}$ where*

$$\bar{h}_{\gamma,\zeta}^+(x)=\left(\frac{\mathbb{P}_{X\sim\mathcal{P}_x}(X>x,\ X\in\bar{\mathcal{A}}_{\gamma-\zeta})}{\mathbb{P}_{X\sim\mathcal{P}_x}(X\in\bar{\mathcal{A}}_{\gamma+\zeta})}\right)^2\cdot\frac{\mathbb{E}_{X\sim\mathcal{P}_x}\left[(X-x)\mathbb{1}\left(X>x,\ X\in\bar{\mathcal{A}}_{\gamma-\zeta}\right)\right]}{\mathbb{P}_{X\sim\mathcal{P}_x}\left[X>x,\ X\in\bar{\mathcal{A}}_{\gamma+\zeta}\right]},$$

$$\bar{h}_{\gamma,\zeta}^-(x)=\left(\frac{\mathbb{P}_{X\sim\mathcal{P}_x}(X<x,\ X\in\bar{\mathcal{A}}_{\gamma-\zeta})}{\mathbb{P}_{X\sim\mathcal{P}_x}(X\in\bar{\mathcal{A}}_{\gamma+\zeta})}\right)^2\cdot\frac{\mathbb{E}_{X\sim\mathcal{P}_x}\left[(x-X)\mathbb{1}\left(X<x,\ X\in\bar{\mathcal{A}}_{\gamma-\zeta}\right)\right]}{\mathbb{P}_{X\sim\mathcal{P}_x}\left[X<x,\ X\in\bar{\mathcal{A}}_{\gamma+\zeta}\right]}.$$

*Proof of Lemma G.4.* We first provide a lower bound for the function $h_\gamma^+(x)$ via lower bounding the three parts separately. For the term $\mathbb{P}_{X\sim\mathcal{D}}(X > x \mid X \in \mathcal{A}_\gamma)$, we connect the probability to the probability w.r.t $f_0$. Denote the R.H.S in Lemma G.2: $O\left(\epsilon(n) \cdot e^{\gamma+\zeta} \cdot \zeta^{-2}\right)$ by $p(n)$, it holds that

$$
\begin{aligned}
\mathbb{P}_{X\sim\mathcal{D}}(X > x \mid X \in \mathcal{A}_\gamma) &= \frac{\mathbb{P}_{X\sim\mathcal{D}}(X > x,\, X \in \mathcal{A}_\gamma)}{\mathbb{P}_{X\sim\mathcal{D}}(X \in \mathcal{A}_\gamma)} \\
&\geq \frac{\mathbb{P}_{X\sim\mathcal{D}}(X > x,\, X \in \bar{\mathcal{A}}_{\gamma-\zeta}) - \mathbb{P}_{X\sim\mathcal{D}}(X > x,\, X \in \bar{\mathcal{A}}_{\gamma-\zeta},\, X \in \mathcal{A}_\gamma^{\complement})}{\mathbb{P}_{X\sim\mathcal{D}}(X \in \bar{\mathcal{A}}_{\gamma+\zeta}) + \mathbb{P}_{X\sim\mathcal{D}}(X \in \mathcal{A}_\gamma,\, X \in \bar{\mathcal{A}}_{\gamma+\zeta}^{\complement})} \\
&\geq \frac{\mathbb{P}_{X\sim\mathcal{D}}(X > x,\, X \in \bar{\mathcal{A}}_{\gamma-\zeta}) - O\left(\epsilon(n) \cdot e^{\gamma+\zeta} \cdot \zeta^{-2}\right)}{\mathbb{P}_{X\sim\mathcal{D}}(X \in \bar{\mathcal{A}}_{\gamma+\zeta}) + O\left(\epsilon(n) \cdot e^{\gamma+\zeta} \cdot \zeta^{-2}\right)} \\
&= \frac{\mathbb{P}_{X\sim\mathcal{D}}(X > x,\, X \in \bar{\mathcal{A}}_{\gamma-\zeta}) - p(n)}{\mathbb{P}_{X\sim\mathcal{D}}(X \in \bar{\mathcal{A}}_{\gamma+\zeta}) + p(n)},
\end{aligned}
\tag{96}
$$

where the last inequality results from applying Lemma G.2 to both the numerator and denominator.

Meanwhile, it is straightforward that $\sqrt{1 + \left(\mathbb{E}\left[X \mid X \in \mathcal{A}_\gamma,\, X > x\right]\right)^2} \geq 1$.

To handle the last term $\mathbb{E}_{X\sim\mathcal{D}}\left[X - x \,\middle|\, X > x,\, X \in \mathcal{A}_\gamma\right]$, we have the following:

$$
\begin{aligned}
\mathbb{E}_{X\sim\mathcal{D}}\left[X - x \,\middle|\, X > x,\, X \in \mathcal{A}_\gamma\right] &= \frac{\mathbb{E}_{X\sim\mathcal{D}}\left[(X-x)\mathbb{1}\left(X > x,\, X \in \mathcal{A}_\gamma\right)\right]}{\mathbb{P}_{X\sim\mathcal{D}}\left[X > x,\, X \in \mathcal{A}_\gamma\right]} \\
&\geq \frac{\mathbb{E}_{X\sim\mathcal{D}}\left[(X-x)\mathbb{1}\left(X > x,\, X \in \mathcal{A}_\gamma\right)\right]}{\mathbb{P}_{X\sim\mathcal{D}}\left[X > x,\, X \in \bar{\mathcal{A}}_{\gamma+\zeta}\right] + p(n)},
\end{aligned}
\tag{97}
$$

where the inequality holds because of Lemma G.2 and our definition of $p(n)$. To deal with the numerator above, we define the truncated version of the uncertain set with respect to $f_0$, where the truncation here means removing a constant portion of features close to the boundary of the interval. For a constant $p > 0$, we define the truncated uncertain set $\bar{\mathcal{A}}_\gamma^p$ as:

$$
\bar{\mathcal{A}}_\gamma^p := \left\{ x \in \bar{\mathcal{A}}_\gamma \,\middle|\, \min\left(\mathbb{P}_{X\sim\mathcal{D}}(X \in \bar{\mathcal{A}}_\gamma,\, X > x),\, \mathbb{P}_{X\sim\mathcal{D}}(X \in \bar{\mathcal{A}}_\gamma,\, X < x)\right) \geq p \right\}.
\tag{98}
$$

With the definition above, we directly have

$$
\mathbb{E}_{X\sim\mathcal{D}}\left[X - x \,\middle|\, X > x,\, X \in \mathcal{A}_\gamma\right] \geq \frac{\mathbb{E}_{X\sim\mathcal{D}}\left[(X-x)\mathbb{1}\left(X > x,\, X \in \bar{\mathcal{A}}_{\gamma-\zeta}^{p(n)}\right)\right]}{\mathbb{P}_{X\sim\mathcal{D}}\left[X > x,\, X \in \bar{\mathcal{A}}_{\gamma+\zeta}\right] + p(n)},
\tag{99}
$$

where the inequality holds because of the fact that $\mathbb{P}_{x\sim\mathcal{D}}\left(|f_0(x)| \leq \gamma - \zeta,\, |f_\theta(x)| > \gamma\right) \leq p(n)$ (Lemma G.2).

Combining the three inequalities for the three parts, we have

$$
h_\gamma^+(x) \geq \left(\frac{\mathbb{P}_{X\sim\mathcal{D}}(X > x,\, X \in \bar{\mathcal{A}}_{\gamma-\zeta}) - p(n)}{\mathbb{P}_{X\sim\mathcal{D}}(X \in \bar{\mathcal{A}}_{\gamma+\zeta}) + p(n)}\right)^2 \cdot \frac{\mathbb{E}_{X\sim\mathcal{D}}\left[(X-x)\mathbb{1}\left(X > x,\, X \in \bar{\mathcal{A}}_{\gamma-\zeta}^{p(n)}\right)\right]}{\mathbb{P}_{X\sim\mathcal{D}}\left[X > x,\, X \in \bar{\mathcal{A}}_{\gamma+\zeta}\right] + p(n)} = \bar{h}_{\gamma,\zeta}^+(x,n).
\tag{100}
$$

In addition, according to Assumption 3.4, $\lim_{n\to\infty} p(n) = 0$. Therefore, as $n$ converges to infinity, the R.H.S will converge (in probability) to

$$
\bar{h}_{\gamma,\zeta}^+(x,n) \xrightarrow{p} \left(\frac{\mathbb{P}_{X\sim\mathcal{P}_x}(X > x,\, X \in \bar{\mathcal{A}}_{\gamma-\zeta})}{\mathbb{P}_{X\sim\mathcal{P}_x}(X \in \bar{\mathcal{A}}_{\gamma+\zeta})}\right)^2 \cdot \frac{\mathbb{E}_{X\sim\mathcal{P}_x}\left[(X-x)\mathbb{1}\left(X > x,\, X \in \bar{\mathcal{A}}_{\gamma-\zeta}\right)\right]}{\mathbb{P}_{X\sim\mathcal{P}_x}\left[X > x,\, X \in \bar{\mathcal{A}}_{\gamma+\zeta}\right]} = \bar{h}_{\gamma,\zeta}^+(x).
\tag{101}
$$

Similarly, we can also prove that

$$
\mathbb{P}_{X\sim\mathcal{D}}(X < x \mid X \in \mathcal{A}_\gamma) \geq \frac{\mathbb{P}_{X\sim\mathcal{D}}(X < x,\, X \in \bar{\mathcal{A}}_{\gamma-\zeta}) - p(n)}{\mathbb{P}_{X\sim\mathcal{D}}(X \in \bar{\mathcal{A}}_{\gamma+\zeta}) + p(n)}.
\tag{102}
$$

$$\sqrt{1 + \left(\mathbb{E}\left[X \mid X \in \mathcal{A}_\gamma, \ X < x\right]\right)^2} \geq 1. \tag{103}$$

$$\mathbb{E}_{X \sim \mathcal{D}}\left[x - X \mid X < x, \ X \in \mathcal{A}_\gamma\right] \geq \frac{\mathbb{E}_{X \sim \mathcal{D}}\left[(x - X)\mathbb{1}\left(X < x, \ X \in \bar{\mathcal{A}}_{\gamma-\zeta}^{p(n)}\right)\right]}{\mathbb{P}_{X \sim \mathcal{D}}\left[X < x, \ X \in \bar{\mathcal{A}}_{\gamma+\zeta}\right] + p(n)}. \tag{104}$$

Therefore, combining the three inequalities above, we have

$$h_\gamma^-(x) \geq \left(\frac{\mathbb{P}_{X \sim \mathcal{D}}(X < x, \ X \in \bar{\mathcal{A}}_{\gamma-\zeta}) - p(n)}{\mathbb{P}_{X \sim \mathcal{D}}(X \in \bar{\mathcal{A}}_{\gamma+\zeta}) + p(n)}\right)^2 \cdot \frac{\mathbb{E}_{X \sim \mathcal{D}}\left[(x - X)\mathbb{1}\left(X < x, \ X \in \bar{\mathcal{A}}_{\gamma-\zeta}^{p(n)}\right)\right]}{\mathbb{P}_{X \sim \mathcal{D}}\left[X < x, \ X \in \bar{\mathcal{A}}_{\gamma+\zeta}\right] + p(n)} = \bar{h}_{\gamma,\zeta}^-(x, n). \tag{105}$$

In addition, as $n$ converges to infinity, the R.H.S will converge (in probability) to

$$\bar{h}_{\gamma,\zeta}^-(x, n) \xrightarrow{p} \left(\frac{\mathbb{P}_{X \sim \mathcal{P}_x}(X < x, \ X \in \bar{\mathcal{A}}_{\gamma-\zeta})}{\mathbb{P}_{X \sim \mathcal{P}_x}(X \in \bar{\mathcal{A}}_{\gamma+\zeta})}\right)^2 \cdot \frac{\mathbb{E}_{X \sim \mathcal{P}_x}\left[(x - X)\mathbb{1}\left(X < x, \ X \in \bar{\mathcal{A}}_{\gamma-\zeta}\right)\right]}{\mathbb{P}_{X \sim \mathcal{P}_x}\left[X < x, \ X \in \bar{\mathcal{A}}_{\gamma+\zeta}\right]} = \bar{h}_{\gamma,\zeta}^-(x). \tag{106}$$

Finally, we have

$$h_\gamma(x) = \min\left\{h_\gamma^+(x), h_\gamma^-(x)\right\} \geq \min\left\{\bar{h}_{\gamma,\zeta}^+(x, n), \bar{h}_{\gamma,\zeta}^-(x, n)\right\} = \bar{h}_{\gamma,\zeta}(x, n). \tag{107}$$

Meanwhile, $\lim_{n \to \infty} \bar{h}_{\gamma,\zeta}(x, n) = \min\left\{\bar{h}_{\gamma,\zeta}^+(x), \bar{h}_{\gamma,\zeta}^-(x)\right\} = \bar{h}_{\gamma,\zeta}(x)$. The proof is complete. $\quad\square$

**Putting everything together.** Now we are ready to plug Lemma G.3 and Lemma G.4 into Theorem 3.1, which leads to a (weighted) TV bound that only depends on the ground-truth function $f_0$.

**Theorem G.5** (Complete version of Theorem 3.5). *For a threshold $\gamma > 0$ and probability $p > 0$, define the uncertain set and the truncated uncertain set (w.r.t. $f_0$) as $\bar{\mathcal{A}}_\gamma := \left\{x \in [-x_{\max}, x_{\max}] \mid |f_0(x)| \leq \gamma\right\}$ and $\bar{\mathcal{A}}_\gamma^p := \left\{x \in \bar{\mathcal{A}}_\gamma \mid \min\left(\mathbb{P}_{X \sim \mathcal{D}}(X \in \bar{\mathcal{A}}_\gamma, \ X > x), \ \mathbb{P}_{X \sim \mathcal{D}}(X \in \bar{\mathcal{A}}_\gamma, \ X < x)\right) \geq p\right\}$. Define $\Gamma(\gamma) = \frac{e^{-\gamma}}{(1 + e^{-\gamma})^2} = \Omega(e^{-\gamma})$. Then for any function $f = f_\theta$ such that the training loss $\mathcal{L}$ is twice differentiable at $\theta$ and any $\gamma > \zeta > 0$, if Assumption 3.4 holds with $\epsilon(n)$, for some function $p(n) = O\left(\epsilon(n) \cdot e^{\gamma+\zeta} \cdot \zeta^{-2}\right)$,*

$$\Gamma(\gamma) \cdot \left(\mathbb{P}_{X \sim \mathcal{D}}\left(X \in \bar{\mathcal{A}}_{\gamma-\zeta}\right) - p(n)\right) \cdot \left(1 + 2\int_{-x_{\max}}^{x_{\max}} |f''(x)| \, \bar{h}_{\gamma,\zeta}(x, n) dx\right) \leq \lambda_{\max}(\nabla^2 \mathcal{L}(\theta)) + 2x_{\max}\mathcal{L}(f), \tag{108}$$

*where $\bar{h}_{\gamma,\zeta}(x, n) = \min\left\{\bar{h}_{\gamma,\zeta}^+(x, n), \bar{h}_{\gamma,\zeta}^-(x, n)\right\}$ and*

$$\bar{h}_{\gamma,\zeta}^+(x, n) = \left(\frac{\mathbb{P}_{X \sim \mathcal{D}}(X > x, \ X \in \bar{\mathcal{A}}_{\gamma-\zeta}) - p(n)}{\mathbb{P}_{X \sim \mathcal{D}}(X \in \bar{\mathcal{A}}_{\gamma+\zeta}) + p(n)}\right)^2 \cdot \frac{\mathbb{E}_{X \sim \mathcal{D}}\left[(X - x)\mathbb{1}\left(X > x, \ X \in \bar{\mathcal{A}}_{\gamma-\zeta}^{p(n)}\right)\right]}{\mathbb{P}_{X \sim \mathcal{D}}\left[X > x, \ X \in \bar{\mathcal{A}}_{\gamma+\zeta}\right] + p(n)},$$

$$\bar{h}_{\gamma,\zeta}^-(x, n) = \left(\frac{\mathbb{P}_{X \sim \mathcal{D}}(X < x, \ X \in \bar{\mathcal{A}}_{\gamma-\zeta}) - p(n)}{\mathbb{P}_{X \sim \mathcal{D}}(X \in \bar{\mathcal{A}}_{\gamma+\zeta}) + p(n)}\right)^2 \cdot \frac{\mathbb{E}_{X \sim \mathcal{D}}\left[(x - X)\mathbb{1}\left(X < x, \ X \in \bar{\mathcal{A}}_{\gamma-\zeta}^{p(n)}\right)\right]}{\mathbb{P}_{X \sim \mathcal{D}}\left[X < x, \ X \in \bar{\mathcal{A}}_{\gamma+\zeta}\right] + p(n)}.$$

*Furthermore, assume that the features $\{x_i\}_{i=1}^n$ are i.i.d. samples from some distribution $\mathcal{P}_x$, then as $n \to \infty$, the asymptotic total variation guarantee will be*

$$\Gamma(\gamma) \cdot \mathbb{P}_{X \sim \mathcal{P}_x}\left(X \in \bar{\mathcal{A}}_{\gamma-\zeta}\right) \cdot \left(1 + 2\int_{-x_{\max}}^{x_{\max}} |f''(x)| \, \bar{h}_{\gamma,\zeta}(x) dx\right) \leq \lambda_{\max}(\nabla^2 \mathcal{L}(\theta)) + 2x_{\max}\mathcal{L}(f), \tag{109}$$

*where $\bar{h}_{\gamma,\zeta}(x) = \min\left\{\bar{h}_{\gamma,\zeta}^+(x), \bar{h}_{\gamma,\zeta}^-(x)\right\}$ and*

$$\bar{h}_{\gamma,\zeta}^+(x) = \left(\frac{\mathbb{P}_{X \sim \mathcal{P}_x}(X > x, \ X \in \bar{\mathcal{A}}_{\gamma-\zeta})}{\mathbb{P}_{X \sim \mathcal{P}_x}(X \in \bar{\mathcal{A}}_{\gamma+\zeta})}\right)^2 \cdot \frac{\mathbb{E}_{X \sim \mathcal{P}_x}\left[(X - x)\mathbb{1}\left(X > x, \ X \in \bar{\mathcal{A}}_{\gamma-\zeta}\right)\right]}{\mathbb{P}_{X \sim \mathcal{P}_x}\left[X > x, \ X \in \bar{\mathcal{A}}_{\gamma+\zeta}\right]},$$

$$\bar{h}_{\gamma,\zeta}^{-}(x) = \left( \frac{\mathbb{P}_{X \sim \mathcal{P}_x}(X < x, \ X \in \bar{\mathcal{A}}_{\gamma-\zeta})}{\mathbb{P}_{X \sim \mathcal{P}_x}(X \in \bar{\mathcal{A}}_{\gamma+\zeta})} \right)^2 \cdot \frac{\mathbb{E}_{X \sim \mathcal{P}_x}\left[(x - X)\mathbb{1}\left(X < x, \ X \in \bar{\mathcal{A}}_{\gamma-\zeta}\right)\right]}{\mathbb{P}_{X \sim \mathcal{P}_x}\left[X < x, \ X \in \bar{\mathcal{A}}_{\gamma+\zeta}\right]}.$$

*Moreover, if $f = f_\theta$ is a stable solution in $\mathcal{F}(\eta, \mathcal{D})$, we can replace $\lambda_{\max}(\nabla^2 \mathcal{L}(\theta))$ in equation 108 and equation 109 by $\frac{2}{\eta}$.*

*Proof of Theorem G.5.* Directly plugging Lemma G.3 and Lemma G.4 into Theorem 3.1, the proof is complete. □

## G.1 SOME DISCUSSIONS

**Trade-off between parameters.** In addition to the same tradeoff in the choice of $\gamma$ as in Theorem 3.1, there is also tradeoff in the choice of $\zeta$. Note that the parameter $\zeta < \gamma$ accounts for the discrepancy of the two uncertain sets $\bar{\mathcal{A}}_{\gamma+\zeta}$ and $\bar{\mathcal{A}}_{\gamma-\zeta}$. As the choice of $\zeta$ becomes smaller, both $\bar{\mathcal{A}}_{\gamma+\zeta}$ and $\bar{\mathcal{A}}_{\gamma-\zeta}$ will converge to $\bar{\mathcal{A}}_\gamma$. As a result, $\mathbb{P}_{X \sim \mathcal{P}_x}\left(X \in \bar{\mathcal{A}}_{\gamma-\zeta}\right)$ will increase and converge to $\mathbb{P}_{X \sim \mathcal{P}_x}\left(X \in \bar{\mathcal{A}}_\gamma\right)$, while both the support and the value of the asymptotic weight function ($\bar{h}_{\gamma,\zeta}(x)$ in Theorem G.5) will become larger. In other words, a smaller $\zeta$ will lead to a better asymptotic guarantee. However, this is not necessarily the case for finite $n$. Note that $p(n)$ scales as $\zeta^{-2}$, and small $\zeta$ could lead to larger $p(n)$, and thus do harm to the finite-$n$ TV bound equation 7. Moreover, although the asymptotic result for smaller $\zeta$ is better, the convergence rate could be worse due to the $p(n)$ term. In conclusion, it is important to select a $\zeta$ that balances the asymptotic result and the convergence speed to the asymptotic result.

## H PROOF OF THEOREM 3.7

We prove the theorem based on the high probability event that $\widetilde{h}_{\gamma,\zeta}(x, n) \geq c$ for any $x \in \mathcal{I}$. In this way, it holds that

$$\Gamma(\gamma) \cdot \left(\mathbb{P}_{X \sim \mathcal{D}}\left(X \in \bar{\mathcal{A}}_{\gamma-\zeta}\right) - p(n)\right) \cdot \left(1 + 2 \int_{-x_{\max}}^{x_{\max}} |f''(x)| \, \bar{h}_{\gamma,\zeta}(x, n) dx\right)$$
$$\geq 2 \int_{-x_{\max}}^{x_{\max}} |f''(x)| \widetilde{h}_{\gamma,\zeta}(x, n) dx \geq 2c \int_{\mathcal{I}} |f''(x)| \, dx. \tag{110}$$

Meanwhile, since $f = f_\theta$ is a stable solution in $\mathcal{F}(\eta, \mathcal{D})$, $\lambda_{\max}(\nabla^2 \mathcal{L}(\theta)) \leq \frac{2}{\eta}$. Note that $\|f\|_\infty \leq B$, which implies that $2x_{\max} \mathcal{L}(f) \leq 2x_{\max}(1 + B)$.

Combining the results with Theorem 3.5, we have

$$\int_{\mathcal{I}} |f''(x)| \, dx \leq \frac{1}{c} \left(\frac{1}{\eta} + x_{\max}(1 + B)\right). \tag{111}$$

Define the set $\mathbb{T}_3$ as $\mathbb{T}_3 := \left\{ f : \mathcal{I} \to \mathbb{R} \ \middle| \ \int_{\mathcal{I}} |f''(x)| \, dx \leq \frac{1}{c}\left(\frac{1}{\eta} + x_{\max}(1 + B)\right), \ \|f\|_\infty \leq B \right\}$. Then we have both $f_\theta, f_0 \in \mathbb{T}_3$. Below we analyze the metric entropy of $\mathbb{T}_3$.

**Lemma H.1.** *Assume the metric is $\ell_\infty$ distance $\|\cdot\|_\infty$, then the metric entropy of $(\mathbb{T}_3, \|\cdot\|_\infty)$ satisfies*

$$\log N(\epsilon, \mathbb{T}_3, \|\cdot\|_\infty) \leq O\left( \sqrt{\frac{\frac{x_{\max}}{\eta} + x_{\max}^2(1 + B)}{\epsilon}} \right), \tag{112}$$

*where the $O(\cdot)$ also absorbs the constant $c$ and $\frac{1}{|\mathcal{I}|}$.*

*Proof of Lemma H.1.* Assume the interval $\mathcal{I}$ is $\mathcal{I} = [x_{left}, x_{right}]$ and $\widetilde{x} = (x_{left} + x_{right})/2$ is the middle point of $\mathcal{I}$. Then for any $f \in \mathbb{T}_3$, it is obvious that $f(\widetilde{x}) \leq B \leq x_{\max}(1 + B)$.

Meanwhile, if $|f'(\widetilde{x})| \geq \frac{1}{c}\left(\frac{1}{\eta} + x_{\max}(1 + B)\right)$, then $f$ is monotonic over the interval $\mathcal{I}$ and $|f'(x)| \geq |f'(\widetilde{x})| - \frac{1}{c}\left(\frac{1}{\eta} + x_{\max}(1 + B)\right)$ for any $x \in \mathcal{I}$. As a result,

$$2B \geq |f(x_{right}) - f(x_{left})| \geq |\mathcal{I}| \cdot \left(|f'(\widetilde{x})| - \frac{1}{c}\left(\frac{1}{\eta} + x_{\max}(1 + B)\right)\right). \tag{113}$$

Solving the inequality above, we have

$$|f'(\widetilde{x})| \leq \frac{2B}{|\mathcal{I}|} + \frac{1}{c}\left(\frac{1}{\eta} + x_{\max}(1+B)\right). \tag{114}$$

Let $C_3 = \frac{2B}{|\mathcal{I}|} + \frac{1}{c}\left(\frac{1}{\eta} + x_{\max}(1+B)\right)$ and $\mathbb{T}_4 = \left\{f : \mathcal{I} \to \mathbb{R} \,\middle|\, |f(\widetilde{x})| \leq C_3, \ |f'(\widetilde{x})| \leq C_3, \ \int_{\mathcal{I}} |f''(x)|dx \leq C_3\right\}$.
Then according to the analysis above, we have $f \in \mathbb{T}_4$. As a result, $\mathbb{T}_3 \subset \mathbb{T}_4$, which further implies that $\log N(\epsilon, \mathbb{T}_3, \|\cdot\|_\infty) \leq \log N(\epsilon, \mathbb{T}_4, \|\cdot\|_\infty)$.

Define $\mathbb{T}_5 = \left\{f : [\widetilde{x} - x_{\max}, \widetilde{x} + x_{\max}] \to \mathbb{R} \,\middle|\, |f(\widetilde{x})| \leq C_3, \ |f'(\widetilde{x})| \leq C_3, \ \int_{\widetilde{x}-x_{\max}}^{\widetilde{x}+x_{\max}} |f''(x)|dx \leq C_3\right\}$.
Note that for any $f \in \mathbb{T}_4$, the linear extension of $f$ to $[\widetilde{x} - x_{\max}, \widetilde{x} + x_{\max}]$ belongs to $\mathbb{T}_5$. Therefore, we have $\log N(\epsilon, \mathbb{T}_4, \|\cdot\|_\infty) \leq \log N(\epsilon, \mathbb{T}_5, \|\cdot\|_\infty)$. At the same time, the metric entropy of $\mathbb{T}_5$ is the same as the set $\left\{f : [-x_{\max}, x_{\max}] \to \mathbb{R} \,\middle|\, |f(0)| \leq C_3, \ |f'(0)| \leq C_3, \ \int_{-x_{\max}}^{x_{\max}} |f''(x)|dx \leq C_3\right\}$,
which is a subset of the set $\mathbb{T}$ (in Lemma F.3) with $C_2 = C_3$. According to Lemma F.3, it holds that
$\log N(\epsilon, \mathbb{T}_5, \|\cdot\|_\infty) \leq \log N(\epsilon, \mathbb{T}, \|\cdot\|_\infty) \leq O\left(\sqrt{\frac{C_3 x_{\max}}{\epsilon}}\right)$.

Combining the results above, we finally have

$$\log N(\epsilon, \mathbb{T}_3, \|\cdot\|_\infty) \leq \log N(\epsilon, \mathbb{T}_4, \|\cdot\|_\infty) \leq \log N(\epsilon, \mathbb{T}_5, \|\cdot\|_\infty)$$

$$\leq \log N(\epsilon, \mathbb{T}, \|\cdot\|_\infty) \leq O\left(\sqrt{\frac{C_3 x_{\max}}{\epsilon}}\right) = O\left(\sqrt{\frac{\frac{x_{\max}}{\eta} + x_{\max}^2(1+B)}{\epsilon}}\right), \tag{115}$$

where the last $O(\cdot)$ also absorbs the constant $c$ and $\frac{1}{|\mathcal{I}|}$. $\qquad \square$

With the metric entropy upper bound of $\mathbb{T}_3$ above, we are ready to provide a high probability bound for the generalization gap over the data points in $\mathcal{I}$. For a fixed dataset $\{(x_i, y_i)\}_{i=1}^n$ and a fixed interval $\mathcal{I}$, define the empirical loss on $\mathcal{I}$ as $\mathcal{L}_{\mathcal{I}}(f) = \frac{1}{n_{\mathcal{I}}} \sum_{x_i \in \mathcal{I}} \ell(f, (x_i, y_i))$, where $\ell$ is the logistic loss and $n_{\mathcal{I}} = \sum_{i=1}^n \mathbb{1}(x_i \in \mathcal{I})$ is the number of features in $\mathcal{I}$. Meanwhile, the populational loss is $\bar{\mathcal{L}}_{\mathcal{I}}(f) = \frac{1}{n_{\mathcal{I}}} \sum_{x_i \in \mathcal{I}} \mathbb{E}_{y_i'|x_i} \ell(f, (x_i, y_i'))$, where $\cdot|x$ is the conditional distribution of the label given $x$. Below we derive a high probability bound for $\left|\mathcal{L}_{\mathcal{I}}(f) - \bar{\mathcal{L}}_{\mathcal{I}}(f)\right|$ over the function set $\mathbb{T}_3$.

**Lemma H.2.** *Recall that* $\mathbb{T}_3 := \left\{f : \mathcal{I} \to \mathbb{R} \,\middle|\, \int_{\mathcal{I}} |f''(x)|\, dx \leq \frac{1}{c}\left(\frac{1}{\eta} + x_{\max}(1+B)\right), \ \|f\|_\infty \leq B\right\}$.
*If the loss $\ell$ is logistic loss, then with probability $1 - \frac{\delta}{2}$, for any $f \in \mathbb{T}_3$,*

$$\left|\mathcal{L}_{\mathcal{I}}(f) - \bar{\mathcal{L}}_{\mathcal{I}}(f)\right| \leq O\left(\left[\frac{(B+1)^4 \left(\frac{x_{\max}}{\eta} + x_{\max}^2(1+B)\right)\log(1/\delta)^2}{n_{\mathcal{I}}^2}\right]^{\frac{1}{5}}\right), \tag{116}$$

*where the $O(\cdot)$ also absorbs the constant $c$ and $\frac{1}{|\mathcal{I}|}$.*

*Proof of Lemma H.2.* For a fixed $\epsilon > 0$, according to Lemma H.1, there exists an $\epsilon$-covering set of $\mathbb{T}_3$ (with respect to $\|\cdot\|_\infty$) whose cardinality $N$ satisfies that

$$\log N \leq O\left(\sqrt{\frac{\frac{x_{\max}}{\eta} + x_{\max}^2(1+B)}{\epsilon}}\right). \tag{117}$$

For a fixed function $\bar{f}$ in the covering set, since the loss is logistic loss, for any possible $(x, y)$,

$$\left|\ell(\bar{f}, (x, y))\right| \leq \log\left(1 + e^{|\bar{f}(x)|}\right) \leq B + 1.$$

Meanwhile, note that the features $y_i$'s are sampled independently from the data distribution. According to Hoeffding's inequality, with probability $1 - \delta$, it holds that

$$\left|\mathcal{L}_\mathcal{I}(\bar{f}) - \bar{\mathcal{L}}_\mathcal{I}(\bar{f})\right| \leq (B+1) \cdot \sqrt{\frac{2\log(2/\delta)}{n_\mathcal{I}}}. \tag{118}$$

Together with a union bound over the covering set, we have with probability $1 - \frac{\delta}{2}$, for all $\bar{f}$ in the covering set,

$$\left|\mathcal{L}_\mathcal{I}(\bar{f}) - \bar{\mathcal{L}}_\mathcal{I}(\bar{f})\right| \leq (B+1) \cdot \sqrt{\frac{2\log(4N/\delta)}{n_\mathcal{I}}}$$

$$\leq O\left((B+1) \cdot \frac{\left(\frac{x_{\max}}{\eta} + x_{\max}^2(1+B)\right)^{\frac{1}{4}} \log(1/\delta)^{\frac{1}{2}}}{n_\mathcal{I}^{\frac{1}{2}} \epsilon^{\frac{1}{4}}}\right). \tag{119}$$

Under such high probability event, for any $f \in \mathbb{T}_3$, let $\bar{f}$ be a function in the covering set such that $\|f - \bar{f}\|_\infty \leq \epsilon$. Then it holds that

$$\left|\mathcal{L}_\mathcal{I}(f) - \bar{\mathcal{L}}_\mathcal{I}(f)\right|$$
$$\leq \left|\mathcal{L}_\mathcal{I}(\bar{f}) - \bar{\mathcal{L}}_\mathcal{I}(\bar{f})\right| + O(\epsilon)$$
$$\leq O(\epsilon) + O\left((B+1) \cdot \frac{\left(\frac{x_{\max}}{\eta} + x_{\max}^2(1+B)\right)^{\frac{1}{4}} \log(1/\delta)^{\frac{1}{2}}}{n_\mathcal{I}^{\frac{1}{2}} \epsilon^{\frac{1}{4}}}\right) \tag{120}$$
$$\leq O\left(\left[\frac{(B+1)^4\left(\frac{x_{\max}}{\eta} + x_{\max}^2(1+B)\right)\log(1/\delta)^2}{n_\mathcal{I}^2}\right]^{\frac{1}{5}}\right),$$

where the first inequality holds because $\left|\ell(f, (x,y)) - \ell(\widetilde{f}, (x,y))\right| \leq \left|f(x) - \widetilde{f}(x)\right|$. The last inequality results from selecting the $\epsilon$ that minimizes the objective. $\qquad\square$

Finally, under the assumption that $f = f_\theta$ is "optimized" over the interval $\mathcal{I}$, we can upper bound the excess risk over $\mathcal{I}$.

**Lemma H.3.** *Under the high probability event in Lemma H.2, if $\mathcal{L}_\mathcal{I}(f) \leq \mathcal{L}_\mathcal{I}(f_0)$, then the excess risk over $\mathcal{I}$ is bounded by*

$$\mathrm{ExcessRisk}_\mathcal{I}(f) = \bar{\mathcal{L}}_\mathcal{I}(f) - \bar{\mathcal{L}}_\mathcal{I}(f_0) \leq O\left(\left[\frac{(B+1)^4\left(\frac{x_{\max}}{\eta} + x_{\max}^2(1+B)\right)\log(1/\delta)^2}{n_\mathcal{I}^2}\right]^{\frac{1}{5}}\right), \tag{121}$$

*where $n_\mathcal{I}$ is the number of features in $\mathcal{I}$ and the $O(\cdot)$ also absorbs the constant $c$ and $\frac{1}{|\mathcal{I}|}$.*

*Proof of Lemma H.3.* According to direct calculation, we have

$$\bar{\mathcal{L}}_\mathcal{I}(f) - \bar{\mathcal{L}}_\mathcal{I}(f_0) \leq \left|\bar{\mathcal{L}}_\mathcal{I}(f) - \mathcal{L}_\mathcal{I}(f)\right| + \mathcal{L}_\mathcal{I}(f) - \mathcal{L}_\mathcal{I}(f_0) + \left|\mathcal{L}_\mathcal{I}(f_0) - \bar{\mathcal{L}}_\mathcal{I}(f_0)\right|$$
$$\leq \left|\bar{\mathcal{L}}_\mathcal{I}(f) - \mathcal{L}_\mathcal{I}(f)\right| + 0 + \left|\mathcal{L}_\mathcal{I}(f_0) - \bar{\mathcal{L}}_\mathcal{I}(f_0)\right|$$
$$\leq O\left(\left[\frac{(B+1)^4\left(\frac{x_{\max}}{\eta} + x_{\max}^2(1+B)\right)\log(1/\delta)^2}{n_\mathcal{I}^2}\right]^{\frac{1}{5}}\right), \tag{122}$$

where the last inequality holds due to Lemma H.2 and the fact that $f, f_0 \in \mathbb{T}_3$. $\qquad\square$

Note that the failure probability is at most $\delta$ ($\frac{\delta}{2}$ for $\widetilde{h}_{\gamma,\zeta}(x,n) \geq c$ and $\frac{\delta}{2}$ for Lemma H.2), combining Lemma H.2 and Lemma H.3, the proof of Theorem 3.7 is complete.

