# OpenReview forum: "Does Flatness imply Generalization for Logistic Loss in Univariate Two-Layer ReLU Network?"
_ICLR.cc/2026/Conference — Submitted to ICLR 2026_

### Official Review · Reviewer_B3nP · 2025-10-23

**Soundness:** 3
**Presentation:** 2
**Contribution:** 2
**Rating:** 4
**Confidence:** 4

**Summary:**

This paper studies, on univariate data input, how the flatness translates into generalization for classification tasks. At first, it provides some inequality relation between a weighted TV norm and the sharpness (given by $\nabla^2 \mathcal{L}(\theta)$.

From there, it provides two results:
 - at first it shows that flatness does not necessarily imply generalisation in classification tasks
 - lastly, they provide (via Theorem 3.7) a bound on the excess risk for flat solutions, with some additional assumptions

**Strengths:**

Relating flatness to generalization is an important topic and tackling this in classification problems is not studied enough in the literature. As such, this type of problems is clearly worth investigating.

**Weaknesses:**

My main concern is that the provided results are very complex/hard to parse, while being somehow poor. In particular, I am not sure to understand what kind of message we can retain from this work, besides the "flatness is not enough to ensure generalization in classification" message.

Besides stating complex results, some words are used in a confusing way in my opinion. In particular, the excess risk is defined with respect to the training data examples, which is totally unusual. In consequence, when the authors claim line 394 that their bound (given by Theorem 3.7) matches the minimax optimal rate of Zhang et al. (2024), I somehow disagree: the rate of Zhang et al. (2024) is for the true definition of excess risk (ie population loss), which is harder to bound.

In Theorem 3.5, the authors suggest that their excess risk bound could be changed into a bound on the population loss (line 348), but I think this work would clearly benefit from a precise bound on this quantity (and the same for Theorem 3.7).

**Questions:**

How your "excess risk" bounds can be translated for bounds as the population loss? Of course it holds "asymptoticall" as mentioned in Theorem 3.5, but I guess precise rates would be needed here.

---

> ### Author Response · Authors · 2025-11-25
> **Rebuttal to Reviewer B3nP**
>
> Thanks for your high-quality review, below we reply to your comments.
>
> **My main concern is that the provided results are very complex/hard to parse, while being somehow poor. In particular, I am not sure to understand what kind of message we can retain from this work, besides the "flatness is not enough to ensure generalization in classification" message.**
>
> Thanks for raising the point, we will revise the paper to increase readability. Below we will interpret our main results. In Section 3.1, we prove a weighted TV(1) bound for flat functions, where the smoothness guarantee depends on the uncertainty of the learned function. Then in Section 3.2, we show that flatness alone does not ensure generalization since the function can ensure flatness by predicting confidently. Motivated by this, we further consider the statistical learning setting where the learned function is a consistent estimator (i.e. excess risk converges to 0) which is a reasonable assumption when the ground-truth is realizable. Based on this assumption, in Section 3.4 we further prove a weighted TV(1) bound where the smoothness guarantee is strong within the interior of ''uncertain regions'' of the ground-truth function. Such ''uncertain region'' is predetermined before the training. Furthermore, in Section 3.5, we show that within the interior of ''uncertain regions'' of the ground-truth, we can prove an excess risk bound with optimal convergence rate. To sum up, in Sections 3.4 and 3.5, we show that as well as the estimator is consistent, no matter how fast the excess risk converges to 0, the implicit bias of GD could enhance the convergence rate to a near optimal one.
>
> **Besides stating complex results, some words are used in a confusing way in my opinion. In particular, the excess risk is defined with respect to the training data examples, which is totally unusual. In consequence, when the authors claim line 394 that their bound (given by Theorem 3.7) matches the minimax optimal rate of Zhang et al. (2024), I somehow disagree: the rate of Zhang et al. (2024) is for the true definition of excess risk (ie population loss), which is harder to bound.
> In Theorem 3.5, the authors suggest that their excess risk bound could be changed into a bound on the population loss (line 348), but I think this work would clearly benefit from a precise bound on this quantity (and the same for Theorem 3.7).
> How your "excess risk" bounds can be translated for bounds as the population loss? Of course it holds "asymptoticall" as mentioned in Theorem 3.5, but I guess precise rates would be needed here.**
>
> This is a very good point. In Theorem 3.5, we link the uncertain set of the learned function to the uncertain region of the ground-truth function. Our result is that the gap between these two uncertain sets must be bounded by order $O(\epsilon(n))$. The asymptotic result in Theorem 3.5 is basically when n converges to infinity, the uncertain set of the learned function will converge to the uncertain region of the ground-truth, where the uncertain region of the ground-truth depends on the populational expectation of the feature. This is not the same as populational excess risk. If we switch the excess risk from fixed design to population level, the bound of the gap between these two uncertain regions may differ by a diminishing term. Therefore, the region where the condition in line 376 holds may also shrink by at most a diminishing term. Within such region, the order of the excess risk bound is exactly the same due to a similar application of Hoeffding's inequality. To sum up, if we consider the population level excess risk, the order of the bound will remain the same, while the region where the bound is valid may shrink by a diminishing term. As n approaches infinity, these two cases will converge to each other. We will formalize such idea and add it in the revision.

---

> > ### Comment · Reviewer_B3nP · 2025-11-28
> >
> > I thank the authors for their answer. The lack of clarity and interpretability of the results seems shared by other reviewers, and I think this work might need substantial rewriting.
> >
> > Additionally, I really think the population loss bound claimed by the authors would improve the significance of the paper, and would clearly encourage the authors to add such a result. However, in such state, I am unfortunately maintaining my score.

---

### Official Review · Reviewer_2ptR · 2025-10-27

**Soundness:** 3
**Presentation:** 2
**Contribution:** 3
**Rating:** 4
**Confidence:** 3

**Summary:**

The paper analyzes whether flatness implies generalization in the logistic loss setting. The authors find that there exists solutions that are arbitrarily flat but overfit, thus in general flatness does not imply generalization with logistic loss. However, using a notion of "uncertain set", they show that a similar analysis to previous work can be performed to demonstrate that flatness implies generalization within a region bounded by the left-most and right-most uncertain sets.

**Strengths:**

The paper furthers work on the question of whether flatness implies generalization by investigating logistic loss. The authors provide a concrete example where flatness is not sufficient for generalization, which is in stark contrast to previous work with square loss in which flatness was sufficient. A new analysis technique involving the uncertain sets, as opposed to considering the entire domain, is developed to understand the conditions under which flatness implies generalization in this classification setting.

The experiments corroborate well the theoretical analysis.

**Weaknesses:**

The presentation is relatively dense, challenging to follow for a reader who is not a specialist in the area.  The new ideas in the paper should have been presented more clearly and discussed further.  Similarly, the significance of the main results could have been justified more and explained further.

Theorem 3.3 demonstrates that the generalisation gap of $f_\theta$ is small when $\theta$ is small, however this is rarely mentioned further in the paper.  It is finally compared with Theorem 3.7 which bounds excess risk, but the relationship between these is not elaborated.

**Questions:**

Around line 261, you say that a fair choice of $\gamma$ is not ensured. What would a fair choice of $\gamma$ be? Is this fair $\gamma$ when the term $\frac{1}{\Gamma(\gamma)}$ balances out the effects mentioned on line 256? Or is it when $\frac{1}{\Gamma(\gamma)}$ is small?

Are there any guarantees on the size of the uncertain sets? Would a larger $\mathcal{A}_\gamma$ mean that flatness is more likely to guarantee generalization?

---

> ### Author Response · Authors · 2025-11-25
> **Rebuttal to Reviewer 2ptR**
>
> Thanks for your high-quality review, below we reply to your comments.
>
> **The presentation is relatively dense, challenging to follow for a reader who is not a specialist in the area ......**
>
> Thanks for raising the point, we will revise the paper to increase readability. Below we will interpret our main results. In Section 3.1, we prove a weighted TV(1) bound for flat functions, where the smoothness guarantee depends on the uncertainty of the learned function. Then in Section 3.2, we show that flatness alone does not ensure generalization since the function can ensure flatness by predicting confidently. Motivated by this, we further consider the statistical learning setting where the learned function is a consistent estimator (i.e. excess risk converges to 0) which is a reasonable assumption when the ground-truth is realizable. Based on this assumption, in Section 3.4 we further prove a weighted TV(1) bound where the smoothness guarantee is strong within the interior of ''uncertain regions'' of the ground-truth function. Such ''uncertain region'' is predetermined before the training. Furthermore, in Section 3.5, we show that within the interior of ''uncertain regions'' of the ground-truth, we can prove an excess risk bound with optimal convergence rate. To sum up, in Sections 3.4 and 3.5, we show that as well as the estimator is consistent, no matter how fast the excess risk converges to 0, the implicit bias of GD could enhance the convergence rate to a near optimal one.
>
> **Theorem 3.3 demonstrates that the generalisation gap of $f_\theta$
>  is small when $\theta$ is small, however this is rarely mentioned further in the paper. It is finally compared with Theorem 3.7 which bounds excess risk, but the relationship between these is not elaborated.**
>
> This is a very good question. Basically, Theorem 3.3 and Theorem 3.7 aim to explain the generalization ability of the learned function from two aspects. Note that there exists arbitrarily flat yet overfitting global minimum that has zero loss and no generalization. However, in experiments the optimization does not actually converge to such infima when we choose a reasonably large learning rate, especially when the number of samples $n$ becomes large (please see Figure 2). The goal of our results is to provide some explanation for such phenomenon. Therefore, based on our observations in the experiments, we try to explain the generalization through the following two views. For Theorem 3.3, when the scale of the parameter is bounded (this is supported by our experiment) or grows slowly, the generalization gap would converge to zero as the number of samples converges to infinity, within the whole data support. For Theorem 3.5+3.7, also motivated by the experiment, we assume that the estimator is consistent (i.e. excess risk converges to 0). Based on such assumption, we show that minima stability could enhance the diminishing rate to a near optimal one within the uncertain regions. To sum up, both theorems start from some assumption supported by experiments and show that under such assumption, the learned function would not converge to the infima and could have generalization results.
>
> **Around line 261, you say that a fair choice of $\gamma$
>  is not ensured. What would a fair choice of $\gamma$ be? Is this fair $\gamma$ when the term $1/\Gamma(\gamma)$ balances out the effects mentioned on line 256? Or is it when $1/\Gamma(\gamma)$ is small?**
>
> A fair $\gamma$ means that $\gamma$ itself is not very large (and therefore $1/\Gamma(\gamma)$ is also small) while the set $A_{\gamma}$ is not very small. In this way, the $1/\Gamma(\gamma)$ in the upper bound does not make the result trivial. For line 256, what I mean is when $\gamma$ becomes larger, the uncertain set $\mathcal{A}_\gamma$ will also be larger, which makes the result stronger. However, $1/\Gamma(\gamma)$ in the upper bound is also larger, which weakens the results. Therefore, the choice of $\gamma$ should not be very large.
>
> **Are there any guarantees on the size of the uncertain sets? Would a larger $\mathcal{A}_\gamma$ mean that flatness is more likely to guarantee generalization?**
>
> Unfortunately, due to the example in Theorem 3.2, the size of the uncertain sets is not guaranteed. For the arbitrarily flat yet overfitting solution, $A_\gamma$ will be empty unless $\gamma$ is chosen to be very large. Meanwhile, you are correct that a larger $A_\gamma$ means that flatness is more likely to guarantee generalization. Actually, motivated by the experimental results, we assume that the estimator is consistent in Section 3.4. In this way, the uncertain set of the learned function must be close to the uncertain region of the ground-truth function, and the size of uncertain set can be guaranteed. Based on such results, we further derive generalization results in Section 3.5.

---

### Official Review · Reviewer_nTGn · 2025-10-31

**Soundness:** 3
**Presentation:** 2
**Contribution:** 3
**Rating:** 4
**Confidence:** 4

**Summary:**

This paper studies the generalization behavior of overparameterized two-layer ReLU networks with univariate input, focusing on the logistic loss. While prior work showed that flat (stable) solutions under the square loss do not overfit, the authors find that this relationship is more subtle for logistic loss. They prove that flat solutions can achieve near-optimal generalization within certain uncertainty regions, but also show that some extremely flat solutions can still overfit. Thus, flatness alone does not guarantee generalization. The theoretical findings are supported by experiments.

**Strengths:**

This paper addresses an interesting topic—how dynamical stability, or the curvature of the loss landscape, influences the learned predictor in classification tasks. The authors present results from several perspectives, including bounds on the total variation of the predictor’s output, excess risk, and generalization performance. The results are clearly formulated, and the assumptions are well-specified. Additionally, the theoretical findings are supported by experiments.

**Weaknesses:**

I believe the main weakness of the paper lies in the lack of interpretation of its results. Unlike in the regression setting, where previous findings were relatively straightforward to interpret, several conclusions in this work remain unclear or insufficiently discussed (see questions below).

Moreover, the theory of dynamical stability referenced in the paper applies to local minima, which are equilibria of the gradient descent mapping. However, as the authors themselves note, in the overparameterized regime considered here, true global minima do not exist—only infima do. Consequently, the standard theory of dynamical systems does not directly apply in this context, which weakens the paper’s theoretical motivation. That said, there is empirical evidence showing that the top eigenvalue of the Hessian tends to be regularized during training [R3]. In this sense, the paper's results still have justification.

Additionally, a few of the paper’s contributions and insights are not entirely novel. For instance, the observation that sharpness (or flatness) alone does not guarantee generalization was already discussed in [R1] and [R2] as early as 2017. Another example is the well-known fact that once a network perfectly fits the training data, the weights diverge to infinity in order to minimize the loss, while the Hessian along this trajectory tends to zero. This phenomenon is precisely why prior work has often avoided studying this setting—under the sharpness metric, all interpolating solutions appear equally good. In other words, since the sharpness of any interpolating solution approaches zero, this measure cannot distinguish among them. Please see my questions at the next part in this regard.

**References:**\
[R1] - Exploring Generalization in Deep Learning\
[R2] - Sharp Minima Can Generalize For Deep Nets\
[R3] - Gradient Descent on Neural Networks Typically Occurs at the Edge of Stability

**Questions:**

1) As discussed in the paper, "fully" trained models have vanishing Hessians, regardless of how their predictor function behaves. This implies that curvature cannot distinguish among different predictor functions. How should we interpret Thm. 3.1 and 3.5 in light of this fact? Specifically, for fully trained models, where the loss and the Hessian are effectively zero, the bounds are zero.

2) A similar question arises for Thm. 3.3 and 3.7. The weights for fully trained models are large, as the minimizers are only infima. In this case, the norm of the weights grows to infinity, and the predictor is not bounded, making the bounds trivial. How should we interpret these results?

---

> ### Author Response · Authors · 2025-11-25
> **Rebuttal to Reviewer nTGn (part 1)**
>
> Thanks for your high-quality review, below we reply to your comments.
>
> **I believe the main weakness of the paper lies in the lack of interpretation of its results. Unlike in the regression setting, where previous findings were relatively straightforward to interpret, several conclusions in this work remain unclear or insufficiently discussed.**
>
> Thanks for raising the point, we will revise the paper to increase readability. Below we will interpret our main results. In Section 3.1, we prove a weighted TV(1) bound for flat functions, where the smoothness guarantee depends on the uncertainty of the learned function. Then in Section 3.2, we show that flatness alone does not ensure generalization since the function can ensure flatness by predicting confidently. Motivated by this, we further consider the statistical learning setting where the learned function is a consistent estimator (i.e. excess risk converges to 0) which is a reasonable assumption when the ground-truth is realizable. Based on this assumption, in Section 3.4 we further prove a weighted TV(1) bound where the smoothness guarantee is strong within the interior of ''uncertain regions'' of the ground-truth function. Such ''uncertain region'' is predetermined before the training. Furthermore, in Section 3.5, we show that within the interior of ''uncertain regions'' of the ground-truth, we can prove an excess risk bound with optimal convergence rate. To sum up, in Sections 3.4 and 3.5, we show that as well as the estimator is consistent, no matter how fast the excess risk converges to 0, the implicit bias of GD could enhance the convergence rate to a near optimal one.
>
> **Moreover, the theory of dynamical stability referenced in the paper applies to local minima, which are equilibria of the gradient descent mapping. However, as the authors themselves note, in the overparameterized regime considered here, true global minima do not exist—only infima do. Consequently, the standard theory of dynamical systems does not directly apply in this context, which weakens the paper’s theoretical motivation. That said, there is empirical evidence showing that the top eigenvalue of the Hessian tends to be regularized during training [R3]. In this sense, the paper's results still have justification.**
>
> Our results do not depend on the theory of dynamic stability. We analyze properties of ''flat solutions'' having bounded $\lambda_{\max}(\text{Hessian})$, rather than just ``flat minima''. You are correct that Edge-of-Stability is a stronger motivation for our results, because it provides evidence that almost the *entire trajectory* of GD satisfies EoS.

---

> ### Author Response · Authors · 2025-11-25
> **Rebuttal to Reviewer nTGn (part 2)**
>
> **Additionally, a few of the paper’s contributions and insights are not entirely novel ......**
>
> Our first observation is that "flat solutions can overfit in logistic loss", which is different from the "square loss" study from NeurIPS'24, which proved that flat solutions cannot overfit. This indeed may confuse readers who are familiar with the (now classical) literature on sharpness and generalization, e.g. [R1] and [R2]. The fact is that this existing work does not give the same conclusion. Let us start with [R2] because it concerns the same notion of sharpness measure that we adopted, the largest eigenvalue of Hessian. Note that this [R2] does not say flat solutions can overfit. Their argument is that all flat solutions can be reparameterized to arbitrarily sharp solutions by multiplying a constant on Layer 1 weights and dividing the same constant on Layer 2 weights.  It follows that if a flat solution generalizes, then the corresponding sharp solution also generalizes. Note that "sharp solution can generalize" does not imply anything about whether its converse: "flat solution can overfit" is true, which is what our first result is about. This reparameterization trick does not work the other way. If you take a sharp solution where the weights on both layers are larger, you cannot make it flatter than when the weights are balanced due to AM-GM inequality. In fact, the reparameterization from [R2] does not change the predictive function described by the neural network. In some sense, we deal with the properties of the predictive function directly. Meanwhile, [R1] defines sharpness differently (closer to a first order perturbation bound, rather than curvature), but it also does not say flat solutions can overfit either.
>
> For the well-known fact in your comment, we believe this is what we formally established for logistic loss on two-layer ReLU networks, and most of the paper is about understanding when flatness still implies generalization in practice despite this fact. Could you advise us on how we should cite this observation? Is there a theorem in a paper that we can cite, or is this folklore?
>
>
> **As discussed in the paper, "fully" trained models have vanishing Hessians, regardless of how their predictor function behaves. This implies that curvature cannot distinguish among different predictor functions. How should we interpret Thm. 3.1 and 3.5 in light of this fact? Specifically, for fully trained models, where the loss and the Hessian are effectively zero, the bounds are zero. A similar question arises for Thm. 3.3 and 3.7. The weights for fully trained models are large, as the minimizers are only infima. In this case, the norm of the weights grows to infinity, and the predictor is not bounded, making the bounds trivial. How should we interpret these results?**
>
> This is a very good question. We agree that there are infima that has zero loss and no generalization. However, in experiments the optimization does not actually converge to such infima when we choose a reasonably large learning rate, especially when the number of samples $n$ becomes large (please see Figure 2). The observations are similar to the case of non-parametric regression, where GD generally could not achieve zero-training loss. The goal of our results is to provide some explanation for such phenomenon. Therefore, based on our observations in the experiments, we try to explain the generalization through the following two views. For Theorem 3.3, when the scale of the parameter is bounded (this is supported by our experiment) or grows slowly, the generalization gap would converge to zero as the number of samples converges to infinity. For Theorem 3.5+3.7, also motivated by the experiment, we assume that the estimator is consistent (i.e. excess risk converges to 0). Based on such assumption, we show that minima stability could enhance the diminishing rate to a near optimal one. To sum up, both theorems start from some assumption supported by experiments and show that under such assumption, the learned function would not converge to the infima and could have generalization results. As a disclaimer, when the learned function is the infima (this usually happens when the number of samples is small), both assumptions do not hold and our bounds are no longer valid. However, when the number of samples becomes larger, the cases in our theorems tend to happen. Our explanation for such separation is that when the number of samples is large, the landscape is very complex. Therefore, starting from small initialization, the optimization tends to be trapped in the ``edge of stability'' regime before it could converge to the infima.

---

### Official Review · Reviewer_cYF7 · 2025-11-01

**Soundness:** 3
**Presentation:** 2
**Contribution:** 3
**Rating:** 6
**Confidence:** 3

**Summary:**

This paper studies the relationship between flatness and generalizability for classification. The models considered are univariate (i.e., 1-dimensional input domain) ReLU networks. The authors prove bounds between loss Hessian max eigenvalue and weighted TV norms. Also the authors prove excess risk bounds.

**Strengths:**

In theorem 3.1 the authors prove an upper bound on the weighted TV norm of a function (from the input data to output predictions) in terms of the max eigenvalue of the loss hessian (which is a function from the parameters).

This result is an interesting link between two fundamentally different types of object.

In theorem 3.2, the authors construct an example function whose max eigenvalue of the loss hessian goes to zero, and therefore the function is flat (measured by TV norm). however, the function can have arbitrarily bad performance.

The authors also give an excess risk bound for bounded variation functions when a certain condition is satisfied (i'm not sure if I understand this condition however, see my question below).

**Weaknesses:**

The paper is very symbol heavy. For instance, the weight hγ in the TV norm in Theorem 3.1 is difficult to interpret to the point i can't really tell if the left hand side of inequality (5) is truly a measure of flatness.

**Questions:**

What is the  hγ,ζ (x,n)≥c inside Theorem 3.7? Again, the symbol heaviness makes this is quite difficult to parse.

Can the author give a clear cut definition of "minima stability"? Without knowing what the authors mean by it, it's hard to tell how to interpret theorem 3.7 under the lense of minima stability.

---

> ### Author Response · Authors · 2025-11-25
> **Rebuttal to Reviewer cYF7**
>
> Thanks for your high-quality review, below we reply to your comments.
>
> **The paper is very symbol heavy. For instance, the weight $h_\gamma$ in the TV norm in Theorem 3.1 is difficult to interpret to the point I can't really tell if the left hand side of inequality (5) is truly a measure of flatness. What is the $h_{\gamma,\zeta} (x,n)\geq c$ inside Theorem 3.7?**
>
> Thanks for raising the point, we will revise the paper to increase readability. Below we will interpret our main results and answer your questions. In Section 3.1, we prove a weighted TV(1) bound for flat functions, where the smoothness guarantee depends on the uncertainty of the learned function. The weight function $h_\gamma$ is supported within the convex hull of the uncertain set of the learned function. Therefore the L.H.S measures the smoothness of the learned function when restricted to the uncertain region. Then in Section 3.2, we show that flatness alone does not ensure generalization since the function can ensure flatness by predicting confidently. Motivated by this, we further consider the statistical learning setting where the learned function is a consistent estimator (i.e. excess risk converges to 0) which is a reasonable assumption when the ground-truth is realizable. Based on this assumption, in Section 3.4 we further prove a weighted TV(1) bound where the smoothness guarantee is strong within the interior of ``uncertain regions'' of the ground-truth function. Such ''uncertain region'' is predetermined before the training. Furthermore, in Section 3.5, we show that within the interior of  ''uncertain regions'' of the ground-truth, we can prove an excess risk bound with optimal convergence rate. For your question, $h_{\gamma,\zeta} (x,n)$ is defined in line 376 and this condition will hold within the convex hull of the uncertain region of the ground-truth function. To sum up, in Sections 3.4 and 3.5, we show that as well as the estimator is consistent, no matter how fast the excess risk converges to 0, the implicit bias of GD could enhance the convergence rate to a near optimal one.
>
> **Can the author give a clear cut definition of "minima stability"? Without knowing what the authors mean by it, it's hard to tell how to interpret theorem 3.7 under the lense of minima stability.**
>
> Generally speaking, in optimization—especially non-convex optimization and machine learning—minima stability refers to how robust a minimum is to perturbations, such as changes in parameters, noise, initialization, or the optimization dynamics. More specifically, in our work we consider GD with a constant learning rate $\eta$, and minima stability refers to what kind of minima can GD stably converge to. To characterize such stable minima, linear stability theory (discussed in Appendix B.1) shows that a local minimum must satisfy $\lambda_{\max}(\nabla^2\mathcal{L}(\theta))\leq \frac{2}{\eta}$ for it to be stable. Since our generalization bound depends on the largest eigenvalue of the Hessian matrix, we can derive concrete bounds by incorporating such minima stability results.

---

### Meta-Review · Area_Chair_BsuQ · 2026-01-17

**Summary:**

This paper analyzes whether flatness implies generalization for overparameterized univariate two-layer ReLU networks under logistic loss. While reviewers find the topic interesting, overall support is lukewarm. The main concerns are that the paper is difficult to parse and interpret, and that the core insight is not sufficiently novel relative to prior work. Although the rebuttal provides some intuition, the rebuttal revision does not meaningfully improve the clarity or presentation of the paper. Overall, the paper would require substantial rewriting to clearly convey its contributions and significance.

**Reviewer Concerns:**

All reviewers independently found the paper "difficult to interpret" and "lack of interpretation of its results". While the authors state that they will improve the writing, the rebuttal revision does not convincingly demonstrate that these issues have been resolved.

Reviewer nTGn raised the concern that the insight "flat solutions can overfit" is conceptually unsurprising. Although the rebuttal argues that this contrasts with prior results under squared loss (Qiao et al., 2024), it is well known that under logistic loss, the Hessian vanishes along diverging optimization trajectories. From this perspective, it is not surprising that flatness alone fails to guarantee generalization.

**Reviewer Scores:**

As the main concerns have not yet been fully addressed, it is likely that Reviewers cYF7, nTGn, 2ptR, and B3nP would maintain their scores 6, 4, 4, and 4, respectively.

---

### Decision · Program_Chairs · 2026-01-26

Reject